# EDF1 coordinates cellular responses to ribosome collisions

**Niladri K Sinha[1†], Alban Ordureau[2†], Katharina Best[3†], James A Saba[1], Boris Zinshteyn[1], Elayanambi Sundaramoorthy[4], Amit Fulzele[4], Danielle M Garshott[4], Timo Denk[3], Matthias Thoms[3], Joao A Paulo[2], J Wade Harper[2], Eric J Bennett[4], Roland Beckmann[3]\*, Rachel Green[1]\***

[1]Department of Molecular Biology and Genetics, Howard Hughes Medical Institute, Johns Hopkins University School of Medicine, Baltimore, United States; [2]Department of Cell Biology, Blavatnik Institute of Harvard Medical School, Boston, United States; [3]Gene Center, Department of Biochemistry, Ludwig-Maximilians-Universität München, Munich, Germany; [4]Section of Cell and Developmental Biology, Division of Biological Sciences, University of California, San Diego, San Diego, United States

**Abstract** Translation of aberrant mRNAs induces ribosomal collisions, thereby triggering pathways for mRNA and nascent peptide degradation and ribosomal rescue. Here we use sucrose gradient fractionation combined with quantitative proteomics to systematically identify proteins associated with collided ribosomes. This approach identified Endothelial differentiation-related factor 1 (EDF1) as a novel protein recruited to collided ribosomes during translational distress. Cryo-electron microscopic analyses of EDF1 and its yeast homolog Mbf1 revealed a conserved 40S ribosomal subunit binding site at the mRNA entry channel near the collision interface. EDF1 recruits the translational repressors GIGYF2 and EIF4E2 to collided ribosomes to initiate a negative-feedback loop that prevents new ribosomes from translating defective mRNAs. Further, EDF1 regulates an immediate-early transcriptional response to ribosomal collisions. Our results uncover mechanisms through which EDF1 coordinates multiple responses of the ribosome-mediated quality control pathway and provide novel insights into the intersection of ribosome-mediated quality control with global transcriptional regulation.

**\*For correspondence:** beckmann@genzentrum.lmu.de (RB); ragreen@jhmi.edu (RG)

†These authors contributed equally to this work

## Introduction

Ribosomes are complex macromolecular machines that coordinate the process of protein synthesis in all cells. Throughout their life-cycle, ribosomes interact with sets of factors, usually proteins, that enable their biogenesis and maturation, impact the initiation and elongation phases of the translational cycle, and coordinate their release and recycling from mRNAs (*Hinnebusch, 2014*; *Klinge and Woolford, 2019*; *Schuller and Green, 2018*). It is well known that general stresses lead to attenuation of global protein synthesis (*Holcik and Sonenberg, 2005*). For example, genetic and environmental insults that generate defective mRNAs and proteins, damage ribosomes, or titrate the levels of accessory factors, perturb translational homeostasis and activate kinases critical for the integrated and ribotoxic stress response pathways (*Iordanov et al., 1997*; *Pakos-Zebrucka et al., 2016*; *Wu et al., 2020*). Once triggered, these pathways alter cell fate decisions by either attenuating protein synthesis to alleviate stress and promote cell survival, or activating pro-apoptotic factors that accelerate cell death.

mRNAs that are damaged or difficult-to-translate cause the lead ribosome to stall irreversibly, resulting in collisions with trailing ribosomes on the same mRNA (*Ikeuchi et al., 2019*; *Juszkiewicz et al., 2018*; *Matsuo et al., 2020*; *Simms et al., 2017*). Previous studies have

converged on the idea these collided ribosomes serve as a hub for recruiting a collection of ribosome-mediated quality control (QC) factors that facilitate degradation of the defective mRNAs and partially synthesized peptides, and release of the trapped ribosomes (*Brandman and Hegde, 2016*; *Inada, 2020*). Prompt resolution of ribosomal stalls and removal of partially synthesized proteins are critical for restoring homeostasis, as failure of these processes can lead to cellular protein aggregation and neurodegeneration at the organismal level (*Chu et al., 2009*; *Hipp et al., 2019*; *Ishimura et al., 2014*).

While the temporal sequence of events that lead to these clearance mechanisms are not well characterized, QC events are thought to be triggered when the composite inter-ribosomal interface between collided ribosomes is recognized by the E3 ubiquitin ligase, ZNF598, which ubiquitylates the ribosomal proteins (r-proteins) RPS10 (eS10) and RPS20 (uS10) (*Garzia et al., 2017*; *Juszkiewicz et al., 2018*; *Juszkiewicz and Hegde, 2017*; *Matsuo et al., 2017*; *Simms et al., 2017*; *Sundaramoorthy et al., 2017*). These regulatory ribosomal ubiquitylation marks are thought to trigger a cascade of events that eventually lead to the resolution of ribosomal stalls (*D'Orazio et al., 2019*; *Ikeuchi et al., 2019*; *Juszkiewicz et al., 2018*; *Juszkiewicz and Hegde, 2017*; *Matsuo et al., 2020*; *Matsuo et al., 2017*; *Simms et al., 2017*; *Sundaramoorthy et al., 2017*).

Genetic screens have been productively employed to identify a host of factors involved in the QC steps targeting the mRNA and peptides associated with problematic mRNAs for degradation. For example, genome wide screens in yeast identified the endonuclease Cue2 (N4BP2 in mammals) that degrades problematic mRNAs through the No-Go decay pathway (*D'Orazio et al., 2019*), QC factors such as Hel2, Asc1, and Slh1 (ZNF598, RACK1, and ASCC3 in mammals, respectively) involved in the recognition and resolution of stalled ribosomes (*Brandman et al., 2012*; *Kuroha et al., 2010*; *Letzring et al., 2013*), and downstream peptide targeting factors such as Rqc2, Ltn1, and Cdc48 (NEMF, Listerin, and VCP in mammals, respectively) (*Bengtson and Joazeiro, 2010*; *Brandman et al., 2012*). Proteomic approaches have also been employed in yeast and mammals to identify ribosome-mediated QC factors, though these have generally relied on candidate-based screens involving affinity purification of known factors (*Garzia et al., 2017*; *Matsuo et al., 2017*; *Sitron et al., 2017*; *Zuzow et al., 2018*).

Here, we used sucrose gradient sedimentation and fractionation coupled with quantitative proteomics to systematically characterize the distribution of factors that co-migrate with ribosomal sub-complexes (40S, 60S, 80S and polysomes). This powerful platform identified a core set of ribosome-associated proteins in HCT116 and HEK293 cells under basal growth conditions, and upon induction of transcriptome-wide ribosome collisions. EDF1 was discovered through this approach as a cellular factor that is robustly recruited to polysomes upon conditions that stimulate collisions. Cryo-electron microscopic (Cryo-EM) structures of EDF1 and its highly conserved yeast homolog Mbf1 reveal its binding site at the interface of colliding ribosomes near the mRNA entry channel. We demonstrate that EDF1 plays a central role in facilitating multiple steps associated with ribosome-mediated QC pathways. EDF1 functions upstream of ribosomal stall recognition as its depletion decreases ZNF598-mediated ubiquitylation of eS10 and uS10. Additionally, we show that recruitment of the translational repressors GIGYF2•EIF4E2 to collided ribosomes is dependent upon EDF1 and initiates a negative feedback loop that prevents new ribosomes from translating defective mRNAs. Finally, we provide evidence that EDF1 connects ribosome collision events in the cytoplasm to transcriptional responses in the nucleus.

## Results

### A polysome-proteomics pipeline to identify factors associated with collided ribosomes

To develop a platform to systematically identify proteins associated with collided ribosomes, we first identified conditions that induce global ribosomal collisions. Previous studies reported that global ribosome-collisions can be induced using low, but not high, doses of translational elongation inhibitors such as cycloheximide, anisomycin and emetine (*Juszkiewicz et al., 2018*; *Simms et al., 2017*); such low doses of elongation inhibitors are expected to stall a subset of elongating ribosomes leading to widespread collisions (*Figure 1A*). We selected emetine because its binding to ribosomes is essentially irreversible (compared to cycloheximide and anisomycin) (*Grollman, 1968*;

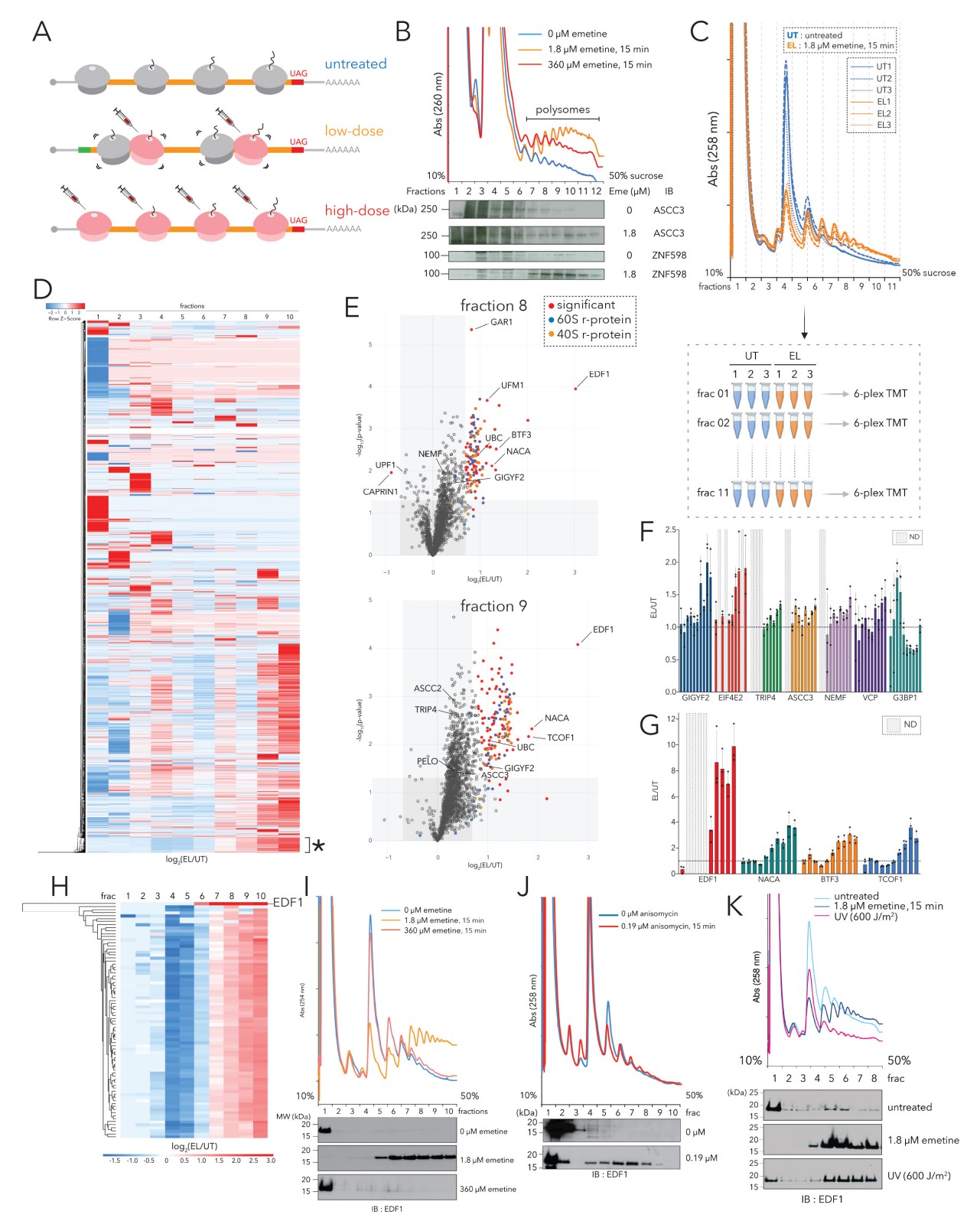

**Figure 1.** Polysome proteomics pipeline identifies EDF1 as a ribosome-mediated quality control factor. (A) Schematic representation of transcriptome-wide ribosomal collisions induced over a range of emetine concentrations; untreated, low-dose (1.8 μM emetine), high-dose (360 μM emetine). (B) Lysates of HCT116 cells treated with no emetine (0 μM, green), 1.8 μM emetine (orange) and 360 μM emetine (red) for 15 min were fractionated over 10-50% sucrose gradients and the resulting UV (A260) absorbance across collected fractions is shown. TCA precipitated proteins were resolved by SDS-

*Figure 1 continued on next page*

*Figure 1 continued*

PAGE and immunoblotted with antibodies against the indicated proteins. (n=2) (**C**) (Top) UV (A258) absorbance across 10-50% sucrose gradients from lysates of HCT116 cells left untreated (untreated, UT x 3, blue traces) or treated with 1.8 µM emetine (low-dose emetine, EL x 3, orange traces) for 15 min. (Bottom) Overview of quantitative polysome proteomics pipeline to monitor fold change in intensity of proteins in low-dose emetine-treated compared to untreated samples. (**D**) Hierarchical clustering of 4288 proteins identified across all fractions. Columns refer to $\log_2$(EL/UT) across density gradient fractions (1-10), rows represent individual proteins. *, cluster of proteins (see text) that exhibits progressive enrichment in $\log_2$(EL/UT) that track with A260 profile. (**E**) Volcano plots of fold change in the abundance of identified proteins in low-dose emetine-treated compared to untreated samples ($\log_2$(EL/UT), x-axis) against its statistical significance (-$\log_{10}$(p-value), y-axis); red: significant, fold change (EL/UT) $\geq$1.6, p-value $\leq$ 0.05. Top panel, fraction 8; bottom panel, fraction 9 (also see *Figure 1—figure supplement 1*). (**F, G**) Mean value of the ratio of relative TMT intensity (EL/UT) for indicated proteins; error bars, SD for n=3. (**H**) Hierarchical clustering of EDF1 and r-proteins. Columns, $\log_2$(EL/UT) across fractions; rows, individual proteins. (**I**) UV (A258) absorbance across 10-50% sucrose gradients from lysates of HCT116 cells untreated (blue) or treated with 1.8 µM (orange, low-dose) or 360 µM (red, high-dose) emetine for 15 min. TCA precipitated fractions were analyzed by immunoblotting (IB) for EDF1. (n=3) (**J**) Same as panel (**I**) except with 0.19 µM anisomycin treatment, 15 min (red). (n=2) (**K**) Same as panel (**I**) except samples were treated with 600J/m$^2$ UV (magenta). (n=2) See also *Figure 1—figure supplement 1* and *Figure 1—source data 1*.

The online version of this article includes the following source data and figure supplement(s) for figure 1:

**Source data 1.** Related to *Figure 1C–H* and *Figure 1—figure supplement 1B–1P*; Sucrose gradient fraction analysis (polysome proteome profiling) with or without low-dose emetine treatment (1.8 µM, 15 min).

**Figure supplement 1.** Polysome proteomics as a platform to identify proteins associated with ribosomes.

---

*Juszkiewicz et al., 2018*) and its effects would thus not be reversed during downstream extract manipulations.

First, we optimized concentrations of emetine to maximize transcriptome-wide ribosomal collisions in HCT116 cells. We monitored collisions by treating cell lysates with ribonuclease A (RNase A) and resolving ribosomal sub-populations across 10–35% sucrose gradients (*Figure 1—figure supplement 1A*). In untreated cells, the majority of polysomes collapsed into monosomes upon RNase A treatment, while in cells treated with low doses of emetine (1.8 µM, 15 min), we observed a significant increase in the fraction of nuclease-resistant disomes (2°) and trisomes (3°).

Using these optimized conditions, we resolved elongating ribosomes across 10–50% sucrose gradients in the absence of RNase A treatment. Compared to untreated HCT116 cells (*Figure 1B*, blue trace) we observed a strong increase of ribosome density in the polysomal fractions for cells treated with low dose emetine (1.8 µM, 15 min) (*Figure 1B*, orange trace) and less so in cells treated with high dose emetine (360 µM, 15 min) (*Figure 1B*, red trace). Individual sucrose gradient fractions were precipitated with trichloroacetic acid (TCA), resolved by SDS-PAGE, and the distribution of ribosome-mediated QC factors were monitored across the gradient. As anticipated, in cells treated with low-dose emetine (EL), compared to untreated cells (UT), ASCC3 and ZNF598 were enriched in denser polysomal fractions (*Figure 1B*).

We next performed quantitative Tandem Mass Tags (TMT)-based proteomics on individual sucrose gradient fractions in untreated (UT) and low-dose emetine (EL, 1.8 µM emetine, 15 min) treated HCT116 cells from three independent biological replicates (*Figure 1C*). Lysates from each sample were quantified (using A260 units) to estimate and optimize reproducibility, and resolved across 10–50% sucrose gradients (*Figure 1C*). For each sample, 11 individual fractions were collected, digested, labeled with TMT (*Figure 1C*, bottom panel) and subjected to mass-spectrometric (MS) analysis.

We quantified a total of 4288 proteins across all fractions (*Figure 1—source data 1*); 79 of the 80 annotated r-proteins were quantified with the exception being RPL41/eL41 which is small (3.5 kilodaltons) and has no tryptic peptides of sufficient length for detection. Principal component analysis (PCA) analyses of individual fractions showed that fractions 1 through 10 could be separated primarily along one principal component with the densest fraction (fraction 11) being an obvious outlier (*Figure 1—figure supplement 1B*). Because fraction 11 is contaminated with large molecular weight complexes comigrating with deep polysomes and nuclear contaminants we eliminated this fraction from all subsequent analyses (*Figure 1—figure supplement 1M*).

For each fraction we plotted the fold change in the intensity of identified proteins in emetine-treated compared to untreated samples ($\log_2$(EL/UT), x-axis) against its statistical significance (-$\log_{10}$(p-value), y-axis; *Figure 1E* and *Figure 1—figure supplement 1C–1M*). Additionally, we performed unbiased clustering to separate the identified proteins into unique subsets based on their

enrichment and distribution along the polysome profile (*Figure 1D*). While we see great diversity in the patterns of emetine-dependent enrichment along the various ribosomal subcomplexes (40S, 60S, 80S and polysomes), we chose to focus on a unique cluster of proteins (discussed below) that exhibits continuous increasing enrichment in the $\log_2$(EL/UT) ratio that tracked with sucrose density (*Figure 1D*, cluster annotated with asterisk).

The emetine-dependent enrichment profile of r-proteins served as a benchmark for annotating differential interacting partners among the ribosomal subcomplexes. Consistent with the profiles of the A260 traces (*Figure 1C*), mapped r-proteins between emetine-treated and untreated samples showed no enrichment in the ribonucleoprotein (RNP) fractions ($\log_2$(EL/UT)~0; fractions 1–2, *Figure 1—figure supplement 1C–1D*, *Figure 1—figure supplement 1N*), de-enrichment along the monosomal fractions ($\log_2$(EL/UT)<0; fractions 4–5; *Figure 1—figure supplement 1F–1G*, *Figure 1— figure supplement 1N*), and progressive enrichment along the light and heavy polysomal fractions ($\log_2$(EL/UT)>0; fractions 6–10; *Figure 1—figure supplement 1H–1L*, *Figure 1—figure supplement 1N*). Known ribosome-binding proteins or those typically associated with ribosome-bound mRNAs followed the overall abundance profile of r-proteins (shown for LARP1, PABPC4, STAU1, and UPF1) (*Figure 1—figure supplement 1O*). Proteins that function on distinct ribosomal subcomplexes are also revealed, such as the late stage initiation factor EIF5B which is preferentially enriched in the monosomal (fractions 4–5) but not polysomal fractions (*Figure 1—figure supplement 1O*).

To report on proteins that specifically associate with collided ribosomes, we hypothesized that these proteins should not simply follow the $\log_2$(EL/UT) trajectory of r-proteins across all fractions, but should instead be enriched in specific fractions relative to r-proteins upon emetine treatment. For example, a few known components of the ribosome-mediated QC machinery such as NEMF and EIF6 were enriched in the monosomal fractions as a function of emetine (*Figure 1—figure supplement 1F–1G*, $\log_2$(EL/UT)>0; *Figure 1—figure supplement 1P*) while cognate ribosome-binding proteins such as LARP1 and UPF1 simply followed the abundance profile of r-proteins (*Figure 1— figure supplement 1F–G*, $\log_2$(EL/UT)<0; *Figure 1—figure supplement 1P*). Next, we showed enrichment across the gradient profile for several known ribosome-mediated QC factors (quantitated in *Figure 1F*). Notably, the cap-dependent translational repressors GIGYF2 and EIF4E2 showed little to no enrichment in the lighter fractions, but were substantially enriched in the heavier polysomal fractions; similar trends were observed for the ribosome-associated protein quality control (RQC) trigger complex (RQT complex), including components TRIP4 and ASCC3. The late stage RQC components, NEMF and VCP, also accumulated along polysomes in an emetine-dependent manner. Interestingly, G3BP1, a component of the USP10 deubiquitylase (DUB) complex that removes ubiquitylation marks on eS10 post ribosomal splitting (*Meyer et al., 2020*), was enriched specifically in the 40S fraction of emetine treated samples (*Figure 1F*).

A subset of proteins was particularly enriched (more than 2-fold in terms of fold change) in polysomal fractions of emetine-treated samples (*Figure 1E*, *Figure 1G*, *Figure 1—figure supplement 1H–1L*). The exemplary candidates in this list were EDF1, NACA, BTF3 and TCOF1. TCOF1 (Treacle protein) is a known substrate of the E3 ubiquitin (Ub) ligase CUL3 that regulates the pseudo-uridylation status of ribosomal RNA (*Werner et al., 2015*). NACA (nascent polypeptide-associated complex subunit alpha) and its binding partner BTF3 (basic transcription factor 3) bind nascent polypeptide chains emerging from the exit tunnel and prevents their mistargeting to the endoplasmic reticulum by the signal recognition particle (SRP) complex (*Gamerdinger et al., 2019*). Additionally, NACA/ BTF3 also function as transcriptional coactivators that regulate the activity of JUN during various differentiation programs (*Addison et al., 2019*).

Overall, our polysome proteomics workflow provided enrichment maps of a set of ribosome-associated proteins that serve as a platform to decipher the ribosome-interactome under basal growth conditions and upon induction of ribotoxic stress.

## Characterization of EDF1 as a novel ribosome-mediated quality control factor

We focused our attention on EDF1 because it was the most enriched protein in heavy polysome fractions of emetine-treated samples (*Figure 1E*, *Figure 1—figure supplement 1H–1L*, *Figure 1—figure supplement 1O*). Importantly, previous genetic studies had implicated its yeast homolog Mbf1 in preventing ribosomal frameshifting on iterated rare codon stretches known to trigger ribosome-mediated QC pathways (*Wang et al., 2018*). We began by looking in more detail at EDF1's

enrichment distribution in our polysome proteomics data set. We were unable to detect EDF1 on 40S, 60S and 80S fractions, but saw strong enrichment ([EL/UT]~7–10) on polysome fractions of emetine-treated samples (*Figure 1G*, *Figure 1—figure supplement 1O*). We compared the enrichment pattern of EDF1 to the cohort of r-proteins (shown in *Figure 1H* and *Figure 1—figure supplement 1O*) – these analyses indicated that EDF1 was barely present in polysomes of untreated cells ([EDF1$_{UT, polysomes}$] << [EDF1$_{EL, polysomes}$] resulting in high EDF1 [EL/UT] ratios).

To validate that EDF1 is recruited to polysomes in a collision-dependent manner, we resolved ribosomes from untreated cells or those treated with low (1.8 µM, 15 min) or high (360 µM, 15 min) doses of emetine across 10–50% sucrose gradients and immunoblotted for EDF1. While little to no EDF1 was detected in polysomal fractions of untreated and high-dose emetine treated cells, EDF1 was strongly recruited to polysomes of cells treated with low-dose emetine (*Figure 1I*). To test if EDF1 recruitment to polysomes was specific to elongation stalls promoted by emetine, we treated cells with low doses of a distinct elongation inhibitor, anisomycin (0.19 µM, 15 min), and again observed strong recruitment of EDF1 to polysomes (*Figure 1J*).

Finally, we wanted to ask whether EDF1 accumulated on collided ribosomes under conditions that mimic a physiological response. Recent observations indicate that pyrimidine dimers accumulate in mRNAs of UV-irradiated cells and impede the decoding step of translational elongation leading to transcriptome-wide ribosome collisions (*Wu et al., 2020*). Indeed, compared to untreated cells, polysomal fractionation of UV-irradiated cells revealed robust recruitment of EDF1 (*Figure 1K*) despite the overall diminished polysome abundance resulting from activation of the Integrated Stress Response (ISR) (*Collier et al., 2015*; *Higgins et al., 2015*).

Taken together, these data identify EDF1 as a highly-enriched factor that is recruited to collided ribosomes under conditions of translational distress.

## Recruitment of EDF1 and GIGYF2 to collided ribosomes is ZNF598 independent

ZNF598 (Hel2 in yeast) is an early-acting ribosome-mediated QC factor that catalyzes site-specific ubiquitylation of eS10 and uS10 at the collided ribosomal interface (*Juszkiewicz et al., 2018*; *Juszkiewicz and Hegde, 2017*; *Matsuo et al., 2017*; *Simms et al., 2017*; *Sundaramoorthy et al., 2017*); while the mechanistic role of eS10 and uS10 ubiquitylation is unclear, it is widely thought to regulate the recruitment other ribosome-mediated QC factors, some of which have well-characterized ubiquitin binding domains. For example, Hel2 in yeast was shown to act upstream of the endonuclease Cue2 (N4BP2 in mammals), which contains four putative ubiquitin binding domains (including the namesake CUE for coupling of ubiquitin binding to ER degradation), and catalyzes the degradation of problematic mRNAs (*D'Orazio et al., 2019*; *Ikeuchi et al., 2019*; *Simms et al., 2017*). CUE domains are also present in Cue3 (ASCC2 in mammals) which is a part of the trimeric RQT complex (along with yKR023W and Slh1; known as TRIP4 and ASCC3 respectively in mammals) that promotes dissociation of collided ribosomes (*Matsuo et al., 2020*; *Matsuo et al., 2017*). While Ltn1-mediated nascent peptide degradation is compromised in yeast carrying ubiquitin binding mutants of Cue3 (*Matsuo et al., 2017*), whether the recruitment of the RQT complex is dependent on Hel2-mediated ubiquitination events is currently unclear (*Juszkiewicz et al., 2020*). Finally, in mammals, ZNF598 was previously shown to associate with the cap-dependent translational initiation repressor EIF4E2 (also called 4EHP) through its binding to the Grb10-interacting GYF protein 2 (GIGYF2) (*Garzia et al., 2017*; *Tollenaere et al., 2019*). Given the strong enrichment of GIGYF2, EIF4E2, EDF1, and other ribosome-mediated QC components on polysomes of emetine-treated cells (*Figure 1F–G*), we wondered whether their recruitment to collided ribosomes might be dependent on ZNF598.

To test ZNF598-dependent recruitment of QC factors, we performed polysome proteomics in HCT116 parental (WT), ΔZNF598, and ZNF598-overexpression (ZNF598-OE) cell lines (*Sundaramoorthy et al., 2017*), without (UT, 0 µM) or with low-dose emetine treatment (EL, 1.8 µM, 15 min); in this case, we pooled fractions from 10–50% sucrose gradients corresponding to monosomes (fraction 5) and polysomes (fractions 6–10) for each sample (*Figure 2A*). Peptides from individual samples were then subjected to TMTpro-MS (*Figure 2B*).

Surprisingly, we noticed that the recruitment of most proteins, including known ribosome-mediated QC factors, to emetine-dependent collided ribosomes occurred in the absence of ZNF598 (log$_2$[ΔZNF598(EL/UT)] > 0, *Figure 2—figure supplement 1B*). Among these, EDF1 was the most

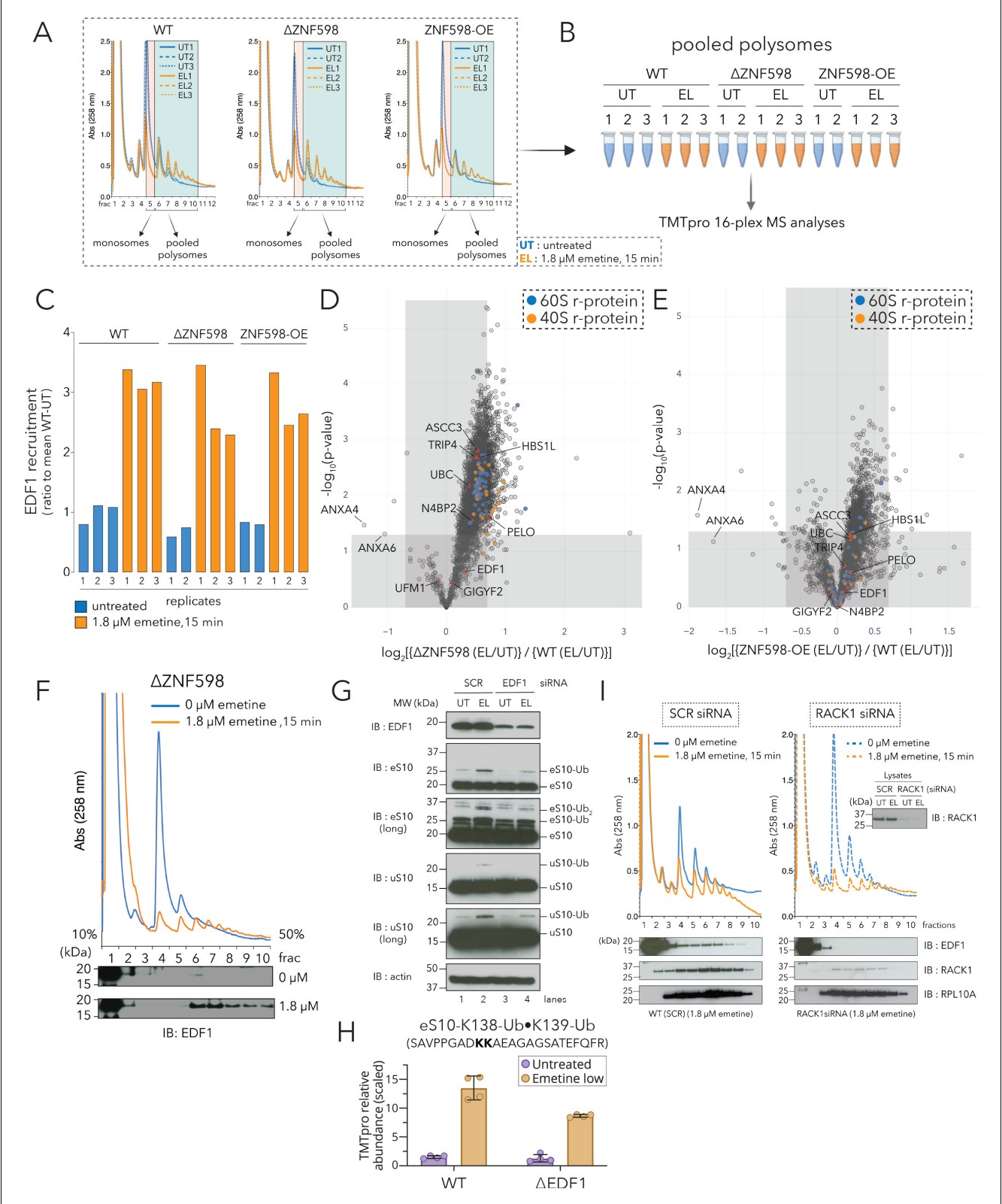

**Figure 2.** Recruitment of EDF1 to collided ribosomes is ZNF598 independent but RACK1 dependent. (**A**) UV (A258) absorbance across 10–50% sucrose gradients from lysates of HCT116 WT, ΔZNF598 and ZNF598-overexpression (OE) cells treated with 0 μM emetine (untreated, UT, blue traces) or 1.8 μM emetine (low-dose emetine, EL, orange traces) for 15 min; fraction five isolated for monosomes; fractions 6–10 pooled for polysomes. (**B**) Schematic of polysome proteomics pipeline to monitor fold change in protein intensity in response to emetine treatment between ΔZNF598 or ZNF598-OE and

*Figure 2 continued on next page*

Figure 2 continued

parental (WT) HCT116 cells. (**C**) Ratio of relative TMT intensity of individual replicates to mean WT-UT intensity for EDF1 in untreated (blue) or low dose (1.8 μM) emetine treated (orange) samples. (**D, E**) Volcano plot of ratio of fold change in protein abundance in response to emetine treatment between ΔZNF598 and WT cells (**D**), or ZNF598-OE and WT cells (**E**). (**F**) UV (A258) absorbance across 10–50% sucrose gradients from lysates of HCT116-Δ ZNF598 cells left untreated (blue) or treated with 1.8 μM emetine for 15 min (orange). TCA precipitated fractions analyzed by immunoblotting with EDF1 antibodies (n = 3). (**G**) Whole cell extracts from untreated (UT) or low-dose of emetine (EL, 1.8 μM, 15 min) treated cells transfected with non-targeting (SCR) or EDF1 siRNAs were analyzed by SDS-PAGE and immunoblotted (IB) with the indicated antibodies. The ubiquitin-modified proteins are indicated. Long, denotes longer exposure. (n = 3) (**H**) Quantification of relative TMT abundance (n = 4) of the doubly modified diGly eS10-K138/139 peptide (SAVPPGAD<u>KK</u>AEAGAGSATEFQFR) normalized to eS10 protein abundance from polysomes of WT and ΔEDF1 cells treated with or without low dose emetine (1.8 μM) for 15 min (also see *Figure 2—figure supplement 2F*). Error bars denote SD for n = 4. (**I**) UV absorbance across 10–50% sucrose gradients from lysates of HCT116 cells treated with non-targeting (SCR) or RACK1 siRNAs with 0 μM emetine (blue) or 1.8 μM emetine treatment for 15 min (orange). TCA precipitated fractions were analyzed by immunoblotting with EDF1, RACK1, or RPL10A antibodies (n = 3). Total lysates (inset) analyzed for RACK1 depletion after siRNA treatment using RACK1 antibody. See also *Figure 2—figure supplement 1*, *Figure 2—figure supplement 2*, *Figure 2—source data 1* and *Figure 2—figure supplement 2—source data 1*.

The online version of this article includes the following source data and figure supplement(s) for figure 2:

**Source data 1.** Related to *Figure 2A-E* and *Figure 2—figure supplement 1A-C*; Polysome sucrose gradient fraction (pooled) analysis with or without low-dose emetine treatment (1.8 μM, 15 min).

**Figure supplement 1.** Recruitment of several known ribosome-mediated QC factors to collided ribosomes is ZNF598 independent.

**Figure supplement 2.** Loss of EDF1 decreases ZNF598-mediated ubiquitylation of eS10 and uS10, and ZAKα-mediated phosphorylation of p38.

**Figure supplement 2—source data 1.** Related to *Figure 2—figure supplement 2A*; Ubiquitin remnant profiling of HEK293T cells treated with 0 μM emetine (UT) or 1.8 μM emetine (EL) for 15 min.

enriched protein accumulating on polysomes of emetine-treated WT, ΔZNF598, and ZNF598-OE cell lines (*Figure 2C*, *Figure 2—figure supplement 1A-C*, *Figure 2—source data 1*). To comprehensively identify factors whose recruitment to collided ribosomes depended on ZNF598, we plotted the ratio of the fold change in protein recruitment in response to emetine treatment between ΔZNF598 and WT cells against their statistical significance (*Figure 2D*, *Figure 2—source data 1*). Proteins whose emetine-dependent polysomal abundance are depleted in ΔZNF598 lines should have an enrichment value less than zero ($\log_2[\{ΔZNF598(EL/UT)\}/\{WT(EL/UT)\}] < 0$); strikingly, under these conditions we see few proteins whose recruitment exhibits strong dependence on ZNF598 (*Figure 2D*, top left quadrant). As a control, we found that r-proteins were enriched in emetine-treated polysome fractions of ΔZNF598 cells (enrichment value > 0), and this effect was largely reversed upon over-expression of ZNF598 (compare r-proteins in *Figure 2D* to *Figure 2E*); these data are consistent with previous results suggesting that clearance of stalled ribosomes is compromised in the absence of ZNF598 (*Juszkiewicz et al., 2018*). Using this analysis, we found that the enrichment of EDF1 to collided ribosomes was unperturbed in the absence of ZNF598, or upon ZNF598 overexpression (*Figure 2D-E*, enrichment value $\geq$ 0). We further validated these results by immunoblot analyses of fractionated lysates from untreated or low dose emetine treated (1.8 μM, 15 min) ΔZNF598 cells, which showed emetine-dependent accumulation of EDF1 on polysomes (*Figure 2F*). Taken together, our results demonstrate that EDF1 recruitment to collided ribosomes is ZNF598 independent.

Interestingly, our enrichment analysis also revealed that the recruitment of known ribosome-mediated QC factors (such as GIGYF2, TRIP4, ASCC3, PELO, HBS1L and N4BP2), some with known Ub-binding domains, to collided ribosomes was largely unperturbed (enrichment value $\geq$ 0) in ΔZNF598 and ZNF598-OE cells (*Figure 2D-E*, *Figure 2—source data 1*). Taken together, these results suggest that while ZNF598 plays an important role in the recognition of ribosomal stalls (*Juszkiewicz et al., 2018*; *Juszkiewicz and Hegde, 2017*; *Simms et al., 2017*; *Sundaramoorthy et al., 2017*), the recruitment of many known QC factors to collided ribosomes is not dependent on ZNF598.

## Loss of EDF1 decreases ZNF598-mediated ubiquitylation of eS10 and uS10

As mentioned previously, a key molecular signature of colliding ribosomes is the ubiquitylation of the r-proteins eS10 and uS10 by ZNF598 (*Juszkiewicz and Hegde, 2017*; *Simms et al., 2017*; *Sundaramoorthy et al., 2017*). Since the recruitment of EDF1 to collided ribosomes does not

depend on ZNF598 (*Figure 2C–F*), we instead wondered whether loss of EDF1 impairs ZNF598-mediated ubiquitylation of r-proteins.

To get a comprehensive look at r-protein ubiquitylation resulting from low dose emetine treatment, we performed ubiquitin-remnant immunoaffinity profiling of untreated (UT) or low-dose emetine (EL, 1.8 µM, 15 min) treated HEK293T cells in triplicate, and plotted the fold change in the intensity of identified peptides in emetine-treated samples compared to untreated samples against their statistical significance (*Figure 2—figure supplement 2A* and *Figure 2—figure supplement 2—source data 1*). In addition to the well characterized ZNF598-dependent eS10 and uS10 Ub-sites at K138/139 and K4/8 respectively (*Juszkiewicz and Hegde, 2017*; *Sundaramoorthy et al., 2017*), we identified a collision-dependent doubly diGly-modified ubiquitylation site on eS31 (RPS27a) at K107/113 (YY<u>K</u>VDENG<u>K</u>ISR; *Figure 2—figure supplement 2A*).

We focused our attention on eS10 and uS10 since these r-proteins were well characterized substrates of ZNF598-mediated ubiquitylation. To test whether loss of EDF1 disrupts ubiquitylation of eS10 and uS10, we treated HEK293T cells with non-targeting (SCR) or EDF1-targeting siRNAs (EDF1i), without (UT, 0 µM) or with low dose emetine (EL, 1.8 µM, 15 min). We observed that depletion of EDF1 decreased emetine-stimulated ubiquitylation of eS10 and uS10 (*Figure 2G*, compare lane 2 to lane 4). To ensure that the effects on eS10 and uS10 ubiquitylation were specific to collisions, we separated cell lysates from non-targeting (SCR) or EDF1 siRNA treated samples, with (EL, 1.8 µM, 15 min) or without (UT, 0 µM) emetine treatment across 10–50% sucrose gradients. Consistent with our results from whole cell extracts, immunoblotting of pooled monosome and polysome fractions showed that EDF1 depletion decreased the amount of mono- and di-ubiquitylated eS10 on polysomes (*Figure 2—figure supplement 2B*).

We performed a similar experiment in ΔEDF1 cell lines (*Figure 2—figure supplement 2C–2D* for characterization of knockout lines) under physiological conditions that induce ribosomal collisions (*Garshott et al., 2020*; *Wu et al., 2020*). Parental and three different clonal populations of ΔEDF1 cells were left untreated or subjected to UV irradiation. Compared to untreated cells, parental (HEK293 Flp-In WT) cell lines showed an increase in eS10 and uS10 ubiquitylation after UV treatment (*Figure 2—figure supplement 2E*, compare lane 1 to lane 2); however, compared to WT cells, all three ΔEDF1 cell clones showed a modest decrease in eS10 and uS10 ubiquitylation (*Figure 2—figure supplement 2E*, compare lane 2 to lanes 4, 6 and 8).

As a final approach to characterize eS10 ubiquitylation, we quantified the levels of the doubly diGly-modified eS10-K138/139 peptide (SAVPPGAD<u>**KK**</u>AEAGAGSATEFQFR) normalized to total eS10 protein from polysomes of WT and ΔEDF1 cells with or without low dose emetine treatment by TMT analyses (*Figure 2H* and *Figure 2—figure supplement 2F*). Consistent with results obtained from immunoblotting, ubiquitylation of eS10 at K138/139 was decreased in ΔEDF1 cells compared to WT cells following low-dose emetine treatment (*Figure 2H* and *Figure 2—figure supplement 2F*). To test whether loss of EDF1 impacted the collision-dependent recruitment of ZNF598 to polysomes, we quantified ZNF598 recruitment to polysomes of WT and ΔEDF1 cells in response to emetine treatment and observed a modest (~10–20%) decrease in the recruitment of ZNF598 in cells lacking EDF1 (*Figure 2—figure supplement 2G*, [{ΔEDF1(EL/UT)}/{WT(EL/UT)}]<1). Based on these results, we conclude that EDF1 facilitates but is not essential for eS10 and uS10 ubiquitylation by ZNF598.

Recent studies have shown that the MAPKKK ZAKα recognizes collided ribosomes to trigger the stress-activated signaling cascade by phosphorylation of p38 and JNKs (*Wu et al., 2020*). We find that compared to WT cells, the emetine-stimulated increase in phospho-p38 level was reduced upon partial depletion of EDF1 with siRNAs (*Figure 2—figure supplement 2H*).

## Recruitment of EDF1 to collided ribosomes is RACK1 dependent

Since recruitment of EDF1 to collided ribosomes is ZNF598 independent, we searched for factors that might enable EDF1 to recognize collided ribosomes. We hypothesized that the unique interface formed between collided disomes may contain binding determinants that enable EDF1 recruitment. Two inter-ribosomal interfaces are formed when ribosomes collide (*Ikeuchi et al., 2019*; *Juszkiewicz et al., 2018*). Interface one is defined by eS1, uS11, eS26 and eS28 of the stalled ribosome and uS4 of the collided ribosome, and interface two is decorated by RACK1 of the stalled ribosome and eS10, uS3, uS10 of the collided ribosome respectively (*Ikeuchi et al., 2019*; *Juszkiewicz et al., 2018*). We focused our attention on RACK1 because proximity-labeling

proteomic analysis of the yeast homolog Asc1 identified Mbf1 as a putative Asc1-interacting protein (*Opitz et al., 2017*). Additionally, Asc1 was shown to cooperate with Mbf1 to prevent +1 frameshifting of ribosomes on problematic codon stretches (*Wang et al., 2018*; *Wolf and Grayhack, 2015*).

HCT116 cells were treated with non-targeting (SCR) or RACK1-specific siRNAs, without or with low-dose emetine (1.8 µM, 15 min), lysed and resolved across 10–50% sucrose gradients. Treatment with RACK1-specific siRNAs did not distort the distribution of polysomes as shown by the A260 traces or RPL10A distribution across the non-targeting (SCR) and RACK1i sucrose gradients (*Figure 2I*). However, emetine-dependent EDF1 accumulation on polysomes was compromised in the RACK1 depleted cells (*Figure 2I*). These results indicate that RACK1 either directly or through its role in stabilizing the collision interface is critical for EDF1 recruitment.

## Structure and recruitment of EDF1 and Mbf1 to collided ribosomes

To understand at a molecular level how EDF1 associates with ribosomes, we performed structural analysis of affinity purified EDF1•ribosome complexes using cryo-EM. To that end, we overexpressed N-terminally FLAG-tagged EDF1 in HEK293 cells and purified EDF1-bound complexes. Notably, despite the enrichment of EDF1 in polysomes following low-dose emetine treatment, we failed to purify significant amounts of EDF1•ribosomal complexes from this fraction, possibly due to masking of the tag. However, we did successfully purify EDF1•80S ribosome complexes which were subjected to cryo-EM analysis (*Figure 3—figure supplement 1A*). After 3D classification (*Figure 3—figure supplement 2A*), the structure of a non-rotated ribosomal complex showing extra density near the mRNA entry tunnel and in the ribosomal A site was determined at an average resolution of 2.9 Å (*Figure 3A*, *Figure 3—figure supplement 1B*, *Figure 3—figure supplement 1D–1E* top panels). Whereas the density in the A site was identified as the cell growth-regulating nucleolar protein LYAR, the density near the mRNA entry channel belonged to EDF1, for which a near complete molecular model was built from residues Ser-24 to Arg-133 (*Figure 3A*).

Since our tagged EDF1 immunoprecipitation did not yield actively elongating polysome-bound EDF1 complexes, we took advantage of the well described collision-inducing SDD1 mRNA in yeast, which allows for affinity purification of stalled polysomes from programmed cell-free extracts (*Ikeuchi et al., 2019*; *Matsuo et al., 2020*). In stalled trisomes isolated from this system, we observed Mbf1 occupying the same position on the ribosome at an average resolution of 3.2 Å (*Figure 3B–D*, *Figure 3—figure supplement 1C*, *Figure 3—figure supplement 1D–1E* bottom panels, *Figure 3—figure supplement 2B*). Although Mbf1 was found associated with the collided ribosome which adopted a rotated state with hybrid tRNAs in the A/P and P/E positions (*Figure 3B*, *Figure 3—figure supplement 2B*), EDF1 and Mbf1 displayed identical structures and modes of binding (*Figure 3C–D*). These observations are consistent with the fact that the conformation of the 40S subunit itself is indistinguishable between the non-rotated state of the EDF1 complex and the rotated state of the Mbf1 complex (*Matsuo et al., 2020*; *Tesina et al., 2020*).

In these structures, the C-terminal helix-turn-helix (HTH)-domain of both proteins is sandwiched between helix 33 (h33) and helix 16 (h16) of the 18S ribosomal RNA (rRNA) with h16 being displaced from its normal position towards EDF1/Mbf1 by about five degrees (*Figure 3D*, *Figure 3E*, bottom left and right). EDF1 and Mbf1 then extend along the mRNA entry channel, interacting prominently with conserved residues of the ribosomal protein uS3 (*Figure 3D*, *Figure 4A*). Interestingly, most of the conserved residues in uS3 interacting with EDF1 and Mbf1 were previously shown to be required for inhibition of frameshifting on stall-inducing iterated CGA codons (*Wang et al., 2018*), underlining the functional significance of the observed architecture in this region in close proximity to the mRNA (*Figure 4A–B*). From uS3, EDF1 and Mbf1 reach down to the base of rRNA h16, where an N-terminal alpha-helix of EDF1/Mbf1 interacts with h16 and the r-proteins uS4 and eS30 (*Figure 3D–E*). Binding of EDF1 and Mbf1 displaces the C-terminus of eS30 which normally interacts with rRNA helix 18 (h18) (*Figure 3E*, top four panels). EDF1 itself interacts with h18 through a GQNKQ-motif, thereby mimicking the interaction of the displaced C-terminus of eS30 (GPNAN). In its canonical conformation, the C-terminus of eS30 forms a loop which would now clash with the repositioned h16 in the EDF1- and Mbf1-bound ribosomes (*Figure 3E*). In addition, lysine 43 (K43) of EDF1 points towards the phosphate backbone of h16 and reaches over the eS30 helix formed by residues extending from glycine 30 to arginine 42, thereby preventing eS30 from maintaining its canonical conformation (*Figure 3E*).

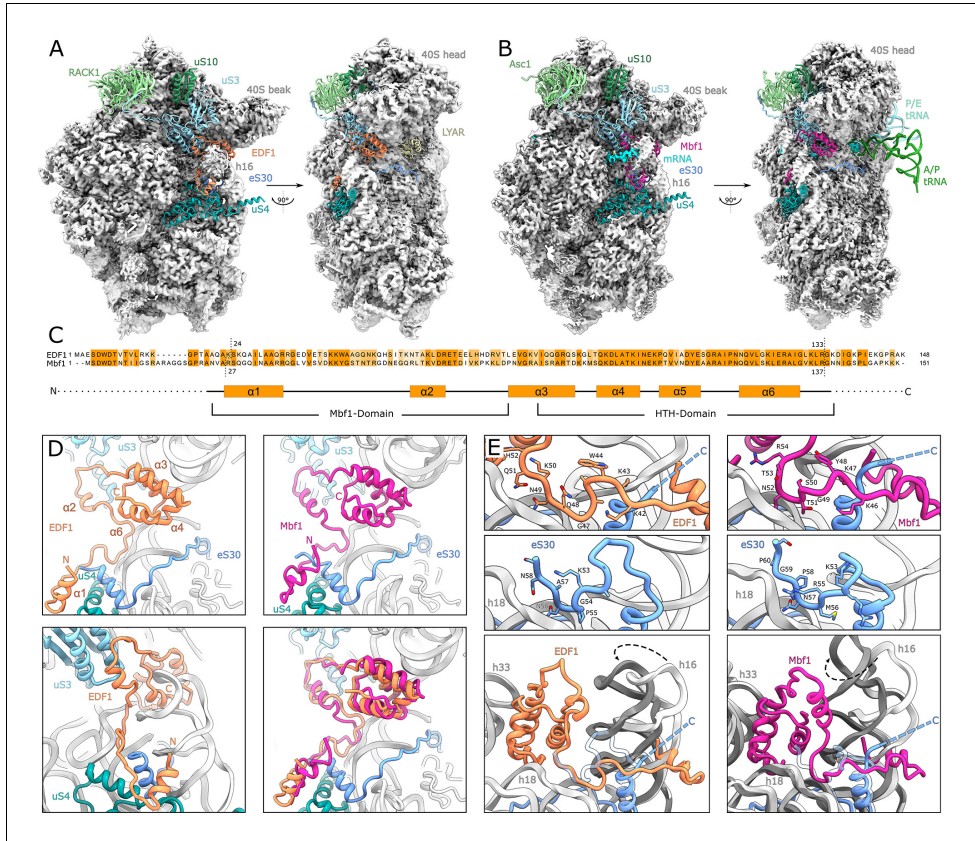

**Figure 3.** Structural analysis of ribosome-bound EDF1 and Mbf1. (**A**) Overview of EM map and models of the 40S subunit of human non-rotated EDF1-bound ribosome. Selected r-proteins and EDF1 (orange) are shown as models in the EDF1•80S map (PDB: 6ZVH). (**B**) Overview of Mbf1 (violet red) bound to the yeast rotated ribosome with hybrid tRNAs. (PDB: 6ZVI). (**C**) Alignment of EDF1 and Mbf1 sequences colored by conservation and domain architecture of EDF1. (**D**) Overall structure of ribosome-bound EDF1 and Mbf1 showing a highly similar fold and binding mode with the C-terminus sandwiched between helix 16 (h16) and helix 33 (h33) of the 18S rRNA and the r-protein uS3 close to the mRNA entry channel, and the N-terminus forming a helix at the base of helix 16. (**E**) EDF1 and Mbf1 interact with rRNA helix 18 (h18), displacing the C-terminus of eS30. Binding of EDF1 and Mbf1 shifts helix 16 towards the ribosome, resulting in a clash of the canonical eS30 position with the new position of helix 16. See also *Figure 3—figure supplement 1*, *Figure 3—figure supplement 2* and *Figure 3—source data 1*.

The online version of this article includes the following source data and figure supplement(s) for figure 3:

**Source data 1.** Cryo-EM data collection, refinement and validation statistics.

**Figure supplement 1.** Sample preparation, cryo-EM maps, local resolution distribution, refinement and model statistics.

**Figure supplement 2.** Cryo-EM data processing scheme for EDF1-ribosome and Mbf1-ribosome datasets.

---

Interestingly, the stretch linking the N-terminal alpha-helix of EDF1/Mbf1 on h16 with the h18/uS3 interaction site contains a conserved KKW$^{42-44}$ motif (KKY$^{46-48}$ in Mbf1), with the aromatic residue ideally positioned to interact with the mRNA in the entry channel. While the mammalian immunopurified EDF1-bound ribosomes lacked visible mRNA, in the Mbf1-bound ribosomes we followed the mRNA density from the E site to the mRNA entry channel near h16 (*Figure 4B*) where Mbf1's KKY$^{46-48}$ motif positions Y48 of Mbf1 to directly interact with the mRNA (*Figure 4C*). The KKY motif is followed immediately by an alpha-helix and together these structural entities clamp the mRNA, resembling a headlock-like arrangement (*Figure 4C*). This notion of an mRNA headlock provided by EDF1/Mbf1 is further supported by the observation that in the presence of Mbf1, the mRNA follows a path which is distinct from the mRNA path in collided ribosomes in the absence of Mbf1 (*Figure 4D*; *Matsuo et al., 2020*). Taken together, we hypothesize that the observed binding mode

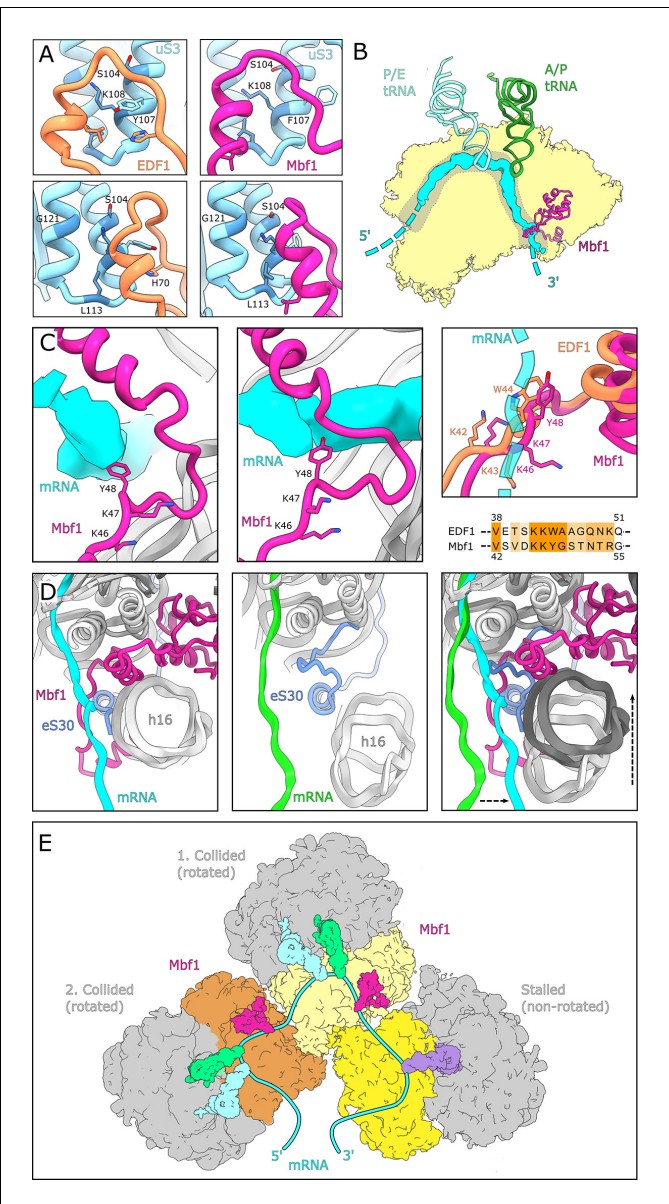

**Figure 4.** Interactions and functional implications of EDF1 and Mbf1. (**A**) EDF1 (orange) and Mbf1 (violet red) interact with ribosomal protein uS3 via a helix-helix interaction. In the human structure, Y107 of uS3 is stacks with H70 of EDF1. Conserved residues required for frameshift inhibition in yeast are colored in steel blue. (**B**) Overview of Mbf1's position with respect to the mRNA path on the 40S ribosomal subunit. (**C**) Mbf1 clamps the mRNA into a headlock, with the aromatic amino acid Y48 exposed to facilitate interaction with the mRNA. The KKY-motif is well conserved between Mbf1 and EDF1 (KKW). (**D**) Comparison of the mRNA path of a Mbf1-bound colliding ribosome with that of a canonical colliding ribosome (PDB: 6I7O). The mRNA and helix 16 are shifted in Mbf1-bound ribosomes. (**E**) Overview of the Mbf1-ribosome interaction in collided polysomes. Mbf1 binds the second and third ribosomes of the trisome unit.

of EDF1/Mbf1 stabilizes the mRNA with respect to the colliding ribosomes, potentially rationalizing how Mbf1 (and likely EDF1) prevents frameshifting.

Notably, when analysing the Mbf1 occupancy of individual ribosomes we observed that Mbf1 was exclusively bound to collided ribosomes found in the rotated state (*Figure 4E*, *Figure 3—figure supplement 2B*). We did not find Mbf1 in the first stalled ribosome, which in the case of the SDD1 mRNA, is found in a non-rotated canonical post state. Moreover, by analysis of neighboring ribosomes, we found that Mbf1 is present in both the first and the second collided ribosome, with a

slight preference for the second one (*Figure 4E*, *Figure 3—figure supplement 2B*). We conclude that EDF1 and Mbf1 are more likely to be recruited to trailing ribosomes following the initial stalling event. The abnormally long-lived rotated state of the collided ribosome, which is unable to complete the tRNA-mRNA translocation step, may serve as a molecular cue for Mbf1/EDF1 recruitment. We speculate that the highly conserved N-terminus of Mbf1/EDF1, which is delocalized in our structures, may play a role in sensing neighboring ribosomes after collision and thus in specific recruitment to colliding ribosomes.

## Defining the EDF1-interactome under normal and ribotoxic-stress conditions

In light of its critical binding site on colliding ribosomes and its general impact on reactions critical to cellular signaling, we wondered whether EDF1 might act as a molecular scaffold to recruit other QC factors to the disome interface. To define the interactome of EDF1 under basal growth conditions and under conditions that induce ribotoxic stress, we performed both traditional immunoaffinity purification and proximity-based labeling (BioID) approaches (*Roux et al., 2013*; *Zuzow et al., 2018*). To avoid artefacts associated with protein overexpression, and in light of the natural high abundance of EDF1 (~$10^5$ copies per cell, *Wiśniewski et al., 2014*), we first immunoaffinity purified endogenous EDF1 from untreated (UT) HEK293T cells or those treated with low dose emetine (EL; 1.8 μM, 15 min) using Protein A coupled EDF1-antibody (*Figure 5A* and *Figure 5—figure supplement 1A*). MS analysis of IP eluates identified TCOF1 along with the translational repressors GIGYF2 and EIF4E2 as strong EDF1-interacting proteins (*Figure 5A*, *Figure 5—source data 1*). Other ribosome-mediated QC factors were also identified including ZNF598 and components of the RQT complex, as well as kinases (mTOR, casein kinase II (CKII), and ribosomal protein S6 kinase (RPS6K)) involved in the phosphorylation of r-proteins (*Figure 5A*, *Figure 5—source data 1*). Interestingly, we find that an interaction between EDF1 and N4BP2 is greatly enhanced under conditions that induce ribosomal collisions (1.8 μM emetine) (*Figure 5A*).

We complemented our AP-MS studies with BioID analyses using stable inducible HEK293 cell lines expressing mutant BirA (R118G) tagged to EDF1 (BirA*-EDF1) (*Samavarchi-Tehrani et al., 2020*). We compared the fold change in the intensity of identified proteins with or without BirA*-EDF1 induction against their statistical significance (*Figure 5B*, *Figure 5—figure supplement 1B–1C*, *Figure 5—source data 2*). Consistent with results obtained from the EDF1 affinity purification, our EDF1-BioID analysis showed that the translational repressors GIGYF2•EIF4E2 along with TCOF1 were among the most significantly enriched proteins (*Figure 5B*). Other notable interactors of EDF1 including components of the ribosome-mediated and mRNA QC machinery are highlighted in *Figure 5—figure supplement 1C*.

Overall, we mapped the EDF1 interactome in cells using complementary approaches which revealed a strong overlap in the top candidates including GIGYF2, EIF4E2, TCOF1, and known components of the ribosome-mediated QC machinery.

## EDF1 recruits GIGYF2•EIF4E2 to collided ribosomes

Our affinity purification and BioID data suggested that EDF1 interacts, directly or indirectly, with the translational repressors GIGYF2 and EIF4E2. To get a comprehensive overview of factors whose recruitment to collided ribosomes depends on EDF1, we set up a polysome proteomics experiment in HEK293 Flp-In T-REx WT and ΔEDF1 cell lines. As previously, cells from WT or ΔEDF1 lines were left untreated (UT x four replicates) or treated with low-dose emetine (EL x four replicates, 1.8 μM, 15 min), lysates were quantified and resolved across 10–50% sucrose gradients (*Figure 6A*). Light (fractions 6–8) and heavy (fractions 9–11) polysomal fractions were pooled separately to distinguish between factors that differentially migrate within these pools. Digested peptides from individual samples from each set were labeled and subjected to TMTpro-MS analyses (*Figure 6A*).

As seen with HCT116 cells (*Figure 1* and *Figure 2*), EDF1 was one of the most enriched proteins in light and heavy polysome fractions of emetine-treated HEK293 WT cells, but was missing in the ΔEDF1 cell line (compare *Figure 6—figure supplement 1A and B* for light polysomes, and *Figure 6—figure supplement 1C and D* for heavy polysomes). EIF5B, LRRFIP2 and FUS were also enriched to varying degrees on light and heavy polysomes of emetine-treated samples (*Figure 6—figure supplement 1A–1D*). As before, we captured significant enrichment of GIGYF2 and EIF4E2 in

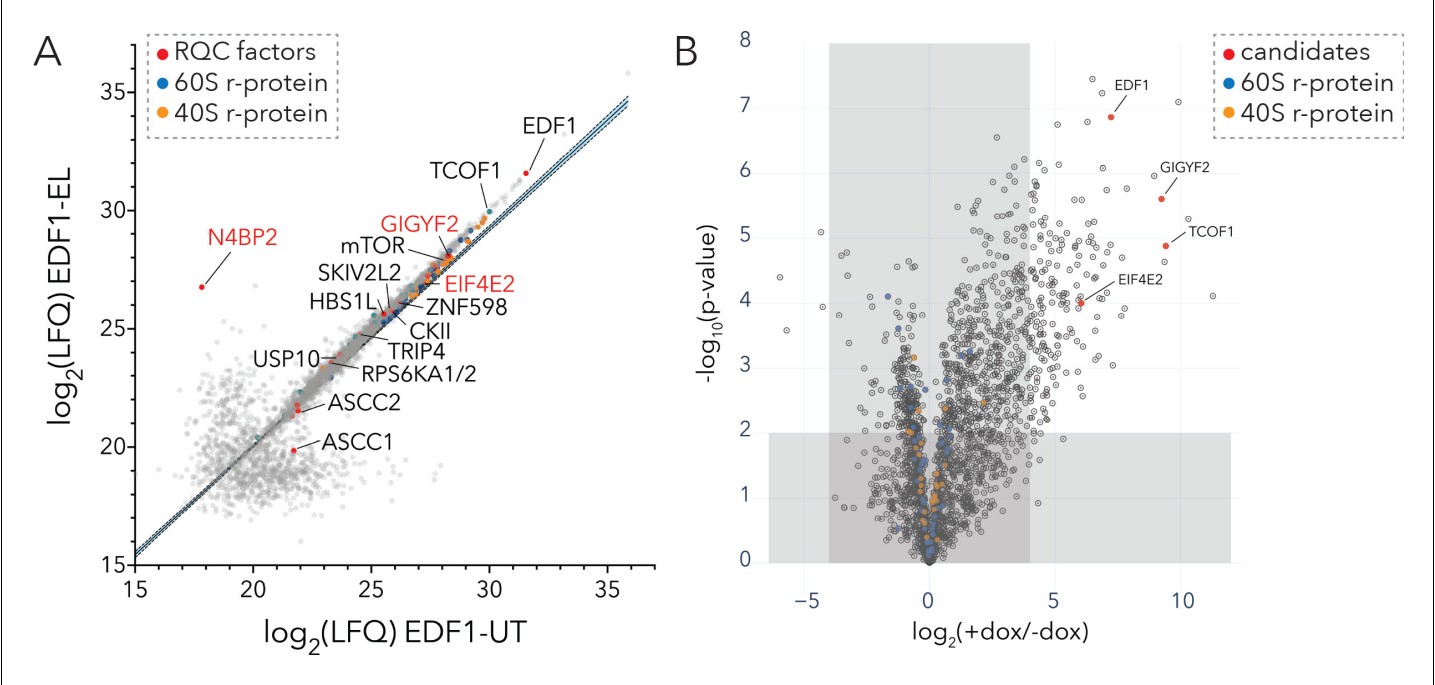

**Figure 5.** Interaction analyses of EDF1 under basal growth and ribotoxic-stress conditions. (**A**) Immunoaffinity purification of endogenous EDF1 from untreated (UT) or low dose emetine treated (EL; 1.8 µM, 15 min) HEK293T cells using Protein A-coupled EDF1-antibody. Scatter plot showing log₂(LFQ) intensity of proteins identified under EL (y-axis) and UT (x-axis) conditions. (**B**) BioID analyses of BirA*-EDF1 with and without doxycycline induction. Volcano plot of fold change in protein LFQ intensity with or without BirA*-EDF1 expression induction by doxycycline (dox). Selected candidates highlighted in red. A cutoff of (+dox/-dox) ≥16 fold and p-value ≤ 0.01 was set to eliminate known BioID contaminants. See also *Figure 5—figure supplement 1*, *Figure 5—source data 1*, and *Figure 5—source data 2*.

The online version of this article includes the following source data and figure supplement(s) for figure 5:

**Source data 1.** Related to *Figure 5A*; Immunoaffinity purification of endogenous EDF1 from untreated (UT) and 1.8 µM emetine treated (EL) HEK293T cells for 15 min.

**Source data 2.** Related to *Figure 5B* and *Figure 5—figure supplement 1B–1C*; BioID analyses of BirA*-EDF1 with or without doxycycline induction for 16 hr.

**Figure supplement 1.** Schematic for AP-MS of endogenous EDF1, and analyses of BirA*-EDF1 interactors identified by BioID.

heavy but not light polysomal fractions of emetine-treated WT samples (compare *Figure 6—figure supplement 1A and C*). These data suggest that translation initiation repressors are predominantly recruited to defective mRNAs with longer queues of collided ribosomes.

Importantly, comparing WT and ΔEDF1 samples, we found a collection of proteins whose recruitment does depend on EDF1 (*Figure 6B–C*; upper left quadrant); these data can be contrasted with our previous observation that few proteins depended on ZNF598 for recruitment to colliding ribosomes (*Figure 2D*, upper left quadrant). Among these, we see that the recruitment of GIGYF2 and EIF4E2 to heavy polysomal fractions of emetine-treated samples strongly depended on EDF1 (*Figure 6C*). The recruitment of LRRFIP2 and FUS to emetine-treated light and heavy polysomes also depended on EDF1, while EIF5B recruitment did not (*Figure 6B–C*).

To further validate our observation that recruitment of GIGYF2 and EIF4E2 to collided ribosomes relied on EDF1, we immunoblotted sucrose gradient fractions from untreated or emetine-treated WT or ΔEDF1 cells (*Figure 6D*). GIGYF2 preferentially accumulated in heavier polysomes of emetine-treated WT cells (*Figure 6D*, left panel); moreover, GIGYF2 was not detected in emetine-treated polysomes of ΔEDF1 cells (*Figure 6D*, right panel). These observations establish that recruitment of the translational repressors GIGYF2 and EIF4E2 to collided ribosomes is dependent upon EDF1.

We next asked whether the EDF1-dependent recruitment of GIGYF2 and EIF4E2 to collided ribosomes could initiate a negative feedback loop preventing new ribosomes from being loaded onto defective mRNAs. To examine this possibility, we used a previously reported dual fluorescence stall

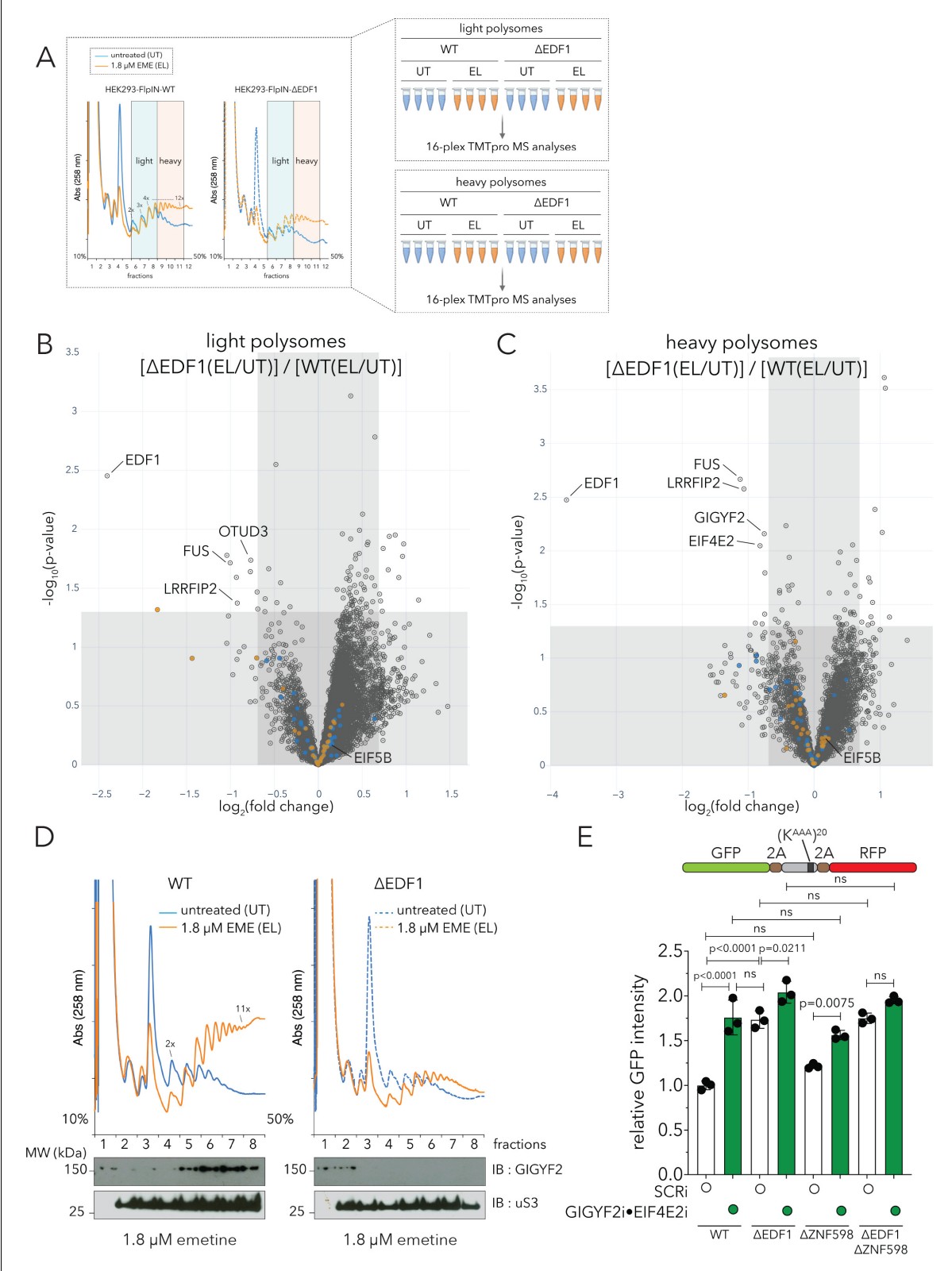

**Figure 6.** EDF1 recruits GIGYF2•EIF4E2 to collided ribosomes. (**A**) (Left) UV (A258) absorbance across 10–50% sucrose gradients from lysates of HEK293 Flp-In TREx WT and ΔEDF1 cells left untreated (UT, blue trace) or treated with 1.8 μM emetine (EL, orange trace) for 15 min; fractions 6–8 pooled for light polysomes; fractions 9–11 pooled for heavy polysomes (n = 4). (Right) Schematic of polysome proteomics pipeline to monitor relative change in protein intensity in response to emetine treatment in light and heavy polysomes between WT and ΔEDF1 cells. (**B, C**) Volcano plot of log₂

*Figure 6 continued on next page*

*Figure 6 continued*

indicated ratio (x-axis) against -log$_{10}$(p-value) (y-axis) for light (**B**) or heavy (**B**) polysomes. (**D**) UV (A258) absorbance across 10–50% sucrose gradients from lysates of HEK293-Flp-In TREx WT and ΔEDF1 cells left untreated (UT, blue trace) or treated with 1.8 μM emetine (EL, orange trace) for 15 min; TCA precipitated proteins from individual fractions were resolved by SDS-PAGE and analyzed by immunoblotting using GIGYF2 and uS3 antibodies. (n = 2) (**E**) Relative GFP intensity from HEK293 Flp-In TREx WT, ΔEDF1, ΔZNF598, and ΔEDF1ΔZNF598 cells transfected with the GFP-(K$^{AAA}$)$_{20}$-RFP stalling reporter without (white bars, non-targeting siRNA, SCRi) or with siRNA-mediated depletion of GIGYF2 and EIF4E2 (green bars; GIGYF2i•EIF4E2i). Error bars denote SD for n = 3. p-values were determined by one-way ANOVA and Tukey's post hoc correction for multiple comparisons. See also *Figure 6—figure supplement 1*, *Figure 6—figure supplement 2* and *Figure 6—source data 1*.

The online version of this article includes the following source data and figure supplement(s) for figure 6:

**Source data 1.** Related to *Figure 6B-C* and *Figure 6—figure supplement 1*; Pooled sucrose gradient fractions (light and heavy polysomes) analysis with or without low-dose emetine treatment (1.8 μM, 15 min).

**Figure supplement 1.** Identification of factors whose recruitment to collided ribosomes depends on EDF1.

**Figure supplement 2.** Loss of EDF1 and GIGYF2 increases the bulk translational output of polyA- and Xbp1u-mediated stalling reporters.

reporter system where GFP and RFP are expressed in frame separated by a ribosome stalling (K$^{AAA}$)$_{20}$ sequence (*Figure 6E*; *Juszkiewicz and Hegde, 2017*; *Sundaramoorthy et al., 2017*). The GFP levels on their own reflect overall reporter mRNA translation while the RFP:GFP ratios reflect the ability of ribosomes to translate through the polyA stall sequence.

If EDF1 recruits GIGYF2•EIF4E2 to collided ribosomes, then its depletion should increase the bulk translational output (as reflected by overall GFP levels) of the polyA-mediated stalling reporter. Indeed, consistent with such a role, loss of EDF1 resulted in a significant increase in GFP fluorescence compared to parental cells (*Figure 6E*). Similarly, GIGYF2•EIF4E2 depletion resulted in the expected increase in GFP fluorescence (*Figure 6E*). Moreover, GIGYF2•EIF4E2 depletion only mildly augmented the already increased GFP levels in ΔEDF1 cells. Restoring EDF1 in ΔEDF1 cells reduced GFP levels whereas expression of truncated versions of EDF1 lacking either the N-terminal MBF1 or the C-terminal HTH domain failed to reduce the elevated GFP levels observed in ΔEDF1 cells (*Figure 6—figure supplement 2A*). The inability of EDF1 truncations to restore translational repression is due to defective ribosome binding (*Figure 6—figure supplement 2B–2C*) consistent with our structural data (*Figures 3–4*) showing that both domains are required for 40S association. By contrast, no significant change in GFP was observed upon loss of ZNF598 (*Figure 6E*). Moreover, the GFP levels observed in ΔEDF1ΔZNF598 lines, without or with GIGYF2•EIF4E2 depletion, reflected those of ΔEDF1 and ΔEDF1•GIGYF2i•EIF4E2i respectively, with no apparent contribution from ZNF598 to the translational repression phenotype (*Figure 6E*). RFP:GFP ratios in these different genetic backgrounds were consistent with prior studies implicating ZNF598 in preventing read-through on problematic mRNA sequences (*Figure 6—figure supplement 2D–2E*; *Juszkiewicz and Hegde, 2017*; *Sundaramoorthy et al., 2017*). Importantly, loss of neither EDF1 nor GIGYF2•EIF4E2 affected the bulk translational output of an identical GFP-RFP reporter without the intervening polyA-stalling sequence (*Figure 6—figure supplement 2F* for relative GFP intensity, *Figure 6—figure supplement 2G* for RFP:GFP ratios).

In an orthogonal set of reporter experiments, loss of EDF1, but not ZNF598, led to an increase in the bulk translation output (as measured by a significant increase in Renilla luciferase (RLuc) activity) on a reporter with an unspliced Xbp1 mRNA (Xbp1u) peptide-pausing sequence (*Shanmuganathan et al., 2019*; *Yanagitani et al., 2011*) previously reported to trigger ribosome collisions (*Han et al., 2020*; *Figure 6—figure supplement 2H* for relative RLuc activity, and *Figure 6—figure supplement 2I* for Fluc:RLuc ratios). Taken together, these data establish that EDF1 recruits the translational repressors GIGYF2•EIF4E2 to multiple types of problematic mRNA sequences.

## EDF1 is critical for JUN-mediated transcriptional response to ribosomal collisions

Previous studies identified EDF1 as an evolutionarily conserved transcriptional coactivator that initiates transcriptional reprogramming in response to cellular stresses (*Baltz et al., 2012*; *Jaimes-Miranda and Chávez Montes, 2020*; *Jindra et al., 2004*; *Kabe et al., 1999*; *Takemaru et al., 1998*; *Takemaru et al., 1997*). For example, in flies and in mammals, EDF1 was shown to bind the protein product of the *jun* locus, JUN/AP-1, to mount a transcriptional response during oxidative

stress (*Jindra et al., 2004*; *Miotto and Struhl, 2006*). Importantly, while there are many JUN transcriptional targets that respond to various JUN/AP-1 complexes, a well-characterized target is *jun* itself which is regulated through a positive feedback loop to mount a sustained cellular response (*Angel et al., 1988*). In light of these connections, we wondered whether an EDF1•JUN nexus is involved in coordinating a transcriptional response in response to collision-inducing stresses.

First, to examine JUN activation in response to collision-inducing stresses, cells were treated with emetine to induce ribosomal collisions. As anticipated, we observed maximal ubiquitylation of eS10 at intermediate but not high concentrations of emetine (*Figure 7A*). Additionally, we observed that the phosphorylation of JUN (at Ser 73) phenocopied the eS10 ubiquitylation dose-response, consistent with a model where JUN is activated in response to ribosomal collisions (*Figure 7A*).

We next evaluated the transcriptional response of cells treated with low doses of emetine, and the dependence of this response on EDF1. RNA-sequencing libraries (in triplicate) were prepared from parental (WT) or ΔEDF1 HEK293-Flp-In TREx cells, untreated (UT) or treated (EL) with low-dose emetine (1.8 µM) for 30 min and 120 min (*Figure 7B*). The resulting libraries were analyzed by differential expression analyses using the DESeq2 pipeline (*Love et al., 2014*). First, we plotted the fold change of normalized transcript reads in emetine-treated compared to untreated samples against its statistical significance for WT cells, and looked for transcripts differentially regulated at 30 and 120 min (*Figure 7C–D*). We observed a striking and progressive increase in the transcript abundance of JUN following low-dose emetine treatment at 30 and 120 min by ~4 and~8 fold respectively (*Figure 7C–D*, *Figure 7F*). We also observed increased expression of other transcripts including FOS, EGR1, ATF3, TXNIP and DUSP1, all transcriptional and signaling regulators that function in diverse stress response pathways (*Figure 7C–D*; *Abraham and Clark, 2006*; *Chen et al., 1996*; p. 3; *Chiu et al., 1988*; *Halazonetis et al., 1988*; *Khachigian et al., 1996*, p. 1; *Slack et al., 2001*). Our initial findings are consistent with a model where ribosomal collisions activate a sustained transcriptional stress response program, with JUN featuring prominently among the upregulated genes.

We next compared ΔEDF1 and WT cells and looked for genes whose emetine-dependent transcript abundance increase is attenuated by loss of EDF1 (*Figures 7E*, 30 min and 120 min). Importantly, transcripts that were differentially regulated between WT and ΔEDF1 lines under basal growth conditions (*Figure 7—figure supplement 1A*) were not among those perturbed by emetine treatment (*Figure 7E*). At both early and late time points, we found that a relatively small collection of genes showed strong dependence on EDF1 (*Figure 7E–F*, $\log_2$(fold change)<0 and -log $p_{adj}$ >2, corresponding to the upper left quadrant of the plots). Strikingly, these included the same cohort of genes (JUN, ATF3, DUSP1 and EGR1) that showed robust transcriptional upregulation in response to ribosomal collisions in WT cells (*Figure 7E–F*); somewhat less strong dependence on EDF1 was also observed for JUND, FOS and FOSB (*Figure 7E–F*). These observations are clarified by the strong overlap (p-value<$10^{-100}$) between transcripts that were upregulated by emetine treatment in WT and ΔEDF1 lines at 30 and 120 min (*Figure 7—figure supplement 1C*); the transcriptional response is similar, but the levels are attenuated in ΔEDF1 cells.

Given previous results describing how EDF1 regulates JUN's transcriptional activity (*Jindra et al., 2004*; *Miotto and Struhl, 2006*), our results are consistent with a model where EDF1 and JUN coordinate a transcriptional program in response to widespread ribosomal collisions.

## Discussion

In this study we combined polysome profiling with quantitative proteomics to generate enrichment maps of a core set of ribosome-associated proteins under basal growth conditions and those recruited during induction of ribosomal collisions (*Figure 1*). Our maps revealed a rich landscape of proteins showing differential association with various ribosomal subcomplexes (40S, 60S, 80S and polysomes) (*Figure 1D–E*, *Figure 1—figure supplement 1C–1L*, *Figure 1—source data 1*,). Here we focused specifically on factors enriched on collided ribosomes that were likely to contribute to ribosome-mediated QC events; among these, EDF1 was an outstanding candidate.

EDF1 is a 16.4 kDa protein in humans with a two-domain architecture comprising of an N-terminal multiprotein bridging factor 1 (MBF1) domain and a conserved C-terminal cro/C1-type helix-turn-helix (HTH) domain (*Figure 3C*). EDF1 is conserved in eukaryotes and archaea, but not found in bacteria (*Aravind and Koonin, 1999*; *de Koning et al., 2009*; *Takemaru et al., 1997*); the archaeal orthologs of EDF1 retain the C-terminal HTH domain, but the N-terminal MBF1 domain is replaced

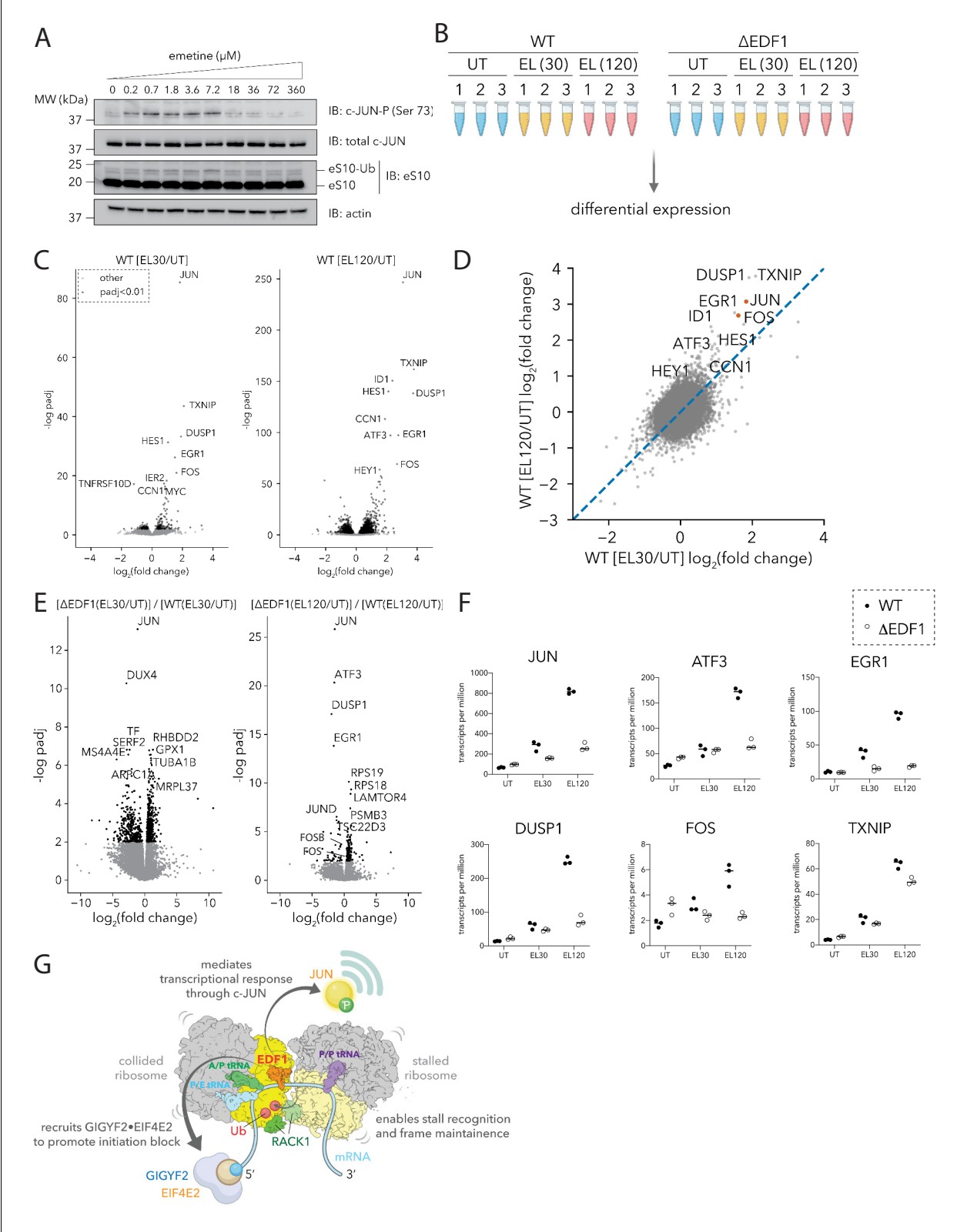

**Figure 7.** EDF1 is critical for JUN-centric transcriptional response to ribosomal collisions. (**A**) Immunoblots of HEK293-Flp-In TREx WT cell extracts showing phosphorylation of JUN at serine 73, and ubiquitylation of eS10 in response to emetine treatment at the indicated concentrations for 15 min. (n = 2) (**B**) Schematic for RNA sequencing analyses of HEK293-Flp-In TREx WT and ΔEDF1 treated with 0 μM (UT) and 1.8 μM emetine for 30 (EL30) and 120 min (EL120). (**C**) Volcano plots of fold change of normalized transcript reads in emetine-treated compared to untreated samples for HEK293-

*Figure 7 continued on next page*

*Figure 7 continued*

Flp-In TREx WT cells at 30 (left) and 120 (right) minutes. (**D**) Scatter plot of log$_2$(EL/UT) fold change of normalized transcript reads from emetine-treated HEK293-Flp-In TREx WT cells after 30 min (x-axis) or 120 min (y-axis). (**E**) Volcano plots of the ratio of fold change in normalized transcript abundance in response to emetine treatment between ΔEDF1 and WT cells at 30 (left) and 120 (right) minutes. (**F**) Normalized transcript reads of selected genes for untreated (UT) and 1.8 μM emetine treated samples at 30 (EL30) and 120 min (EL120) in WT (filled circle) and ΔEDF1 (open circle) cell lines. (**G**) Cartoon showing the multifaceted roles of EDF1 in coordinating different arms of the ribosome-mediated QC pathway and promoting a JUN-centric transcriptional program in response to ribosome collisions. See also *Figure 7—figure supplement 1*.

The online version of this article includes the following figure supplement(s) for figure 7:

**Figure supplement 1.** RNA sequencing analyses of WT and ΔEDF1 cells, and comparison of relative protein expression profiles of ribosome-mediated QC factors.

by a zinc-ribbon domain (*Aravind and Koonin, 1999*). While early characterization of the EDF1 ortholog from *Bombyx mori* revealed that the HTH domain is important for binding the TATA-box binding protein (TBP) (*Liu et al., 2007*; *Takemaru et al., 1997*), later studies of archaeal MBF1 showed that the HTH domain weakly associates with 30S and 70S ribosomes (*Blombach et al., 2014*). Given its evolutionary conservation with the archaeal homolog, we suspected that human EDF1 would also associate with ribosomes. However, unlike archaeal MBF1, human EDF1 behaves differently in that it does not readily associate with 40S and 60S ribosomal subunits. In fact, little to no association of endogenous EDF1 with polysomes is observed in rapidly growing cells (*Figure 1G–I*). This dynamic however changes upon induction of ribosomal collisions, where we observed robust association of EDF1 with polysomal fractions (*Figure 1G–K*). Our experiments showed that recruitment of EDF1 to collided ribosomes requires the presence of RACK1, which is located at the interface of collided ribosomes (*Figure 2I*).

To further characterize how EDF1 associates with collided ribosomes we used cryo-EM to determine structures of EDF1 and its yeast homolog Mbf1 bound to monosomes and polysomes, respectively. EDF1 and Mbf1 bind to a conserved 40S interface strategically positioned at the mRNA entry channel. In addition to its interaction with h16, h18, h33, uS4 and eS30, EDF1/Mbf1 interacts critically with uS3 along the mRNA entry channel. Importantly, genetic screens in yeast have previously identified Mbf1, Asc1 and uS3 as factors critical for suppressing plus one frameshifting (*Culbertson et al., 1982*; *Hendrick et al., 2001*; *Wang et al., 2018*; *Wolf and Grayhack, 2015*). While the exact mechanism by which Mbf1 prevents frameshifting will require further functional characterization, our structural studies suggest that EDF1/Mbf1 may use a conserved KKW motif (KKY in Mbf1) and an alpha-helical segment to clamp the mRNA in a headlock-like arrangement (*Figure 4C*) that prevents frameshifting and stabilizes collided ribosomes. Our functional studies support these observations by showing that loss of EDF1 reduces the efficiency of collision-dependent ZNF598-mediated ubiquitylation of eS10 and uS10 and ZAKα-mediated phosphorylation of the stress-activated protein kinase p38. Based on Mbf1's binding preference for the trailing collided ribosomes, we propose that the abnormally long-lived rotated state of the collided ribosome, and the proximity of the neighboring ribosome, serve as cues for the recruitment of EDF1/Mbf1.

Because, ZNF598 facilitates ribosomal stall recognition through ubiquitylation of eS10 and uS10 (*Juszkiewicz et al., 2018*; *Juszkiewicz and Hegde, 2017*; *Simms et al., 2017*; *Sundaramoorthy et al., 2017*), we predicted that the recruitment of EDF1 and other QC factors would be orchestrated by ZNF598. However, polysome proteomics in ΔZNF598 and ZNF598-OE cell lines surprisingly revealed few proteins whose recruitment to collided ribosomes depended on ZNF598 (*Figure 2D–E*). Of particular interest, we found several ribosome-mediated QC factors, some with known ubiquitin-binding domains (such as N4BP2 and members of the RQT complex), to be equivalently enriched on collided ribosomes in ΔZNF598 cells (*Figure 2D*). How are these ubiquitin-binding proteins recruited to collided ribosomes? Are there alternate ubiquitylation marks and other E3 ubiquitin ligases involved in the recruitment of ribosome-mediated QC components? We showed that in addition to eS10 and uS10, eS31 (RPS27A) is strongly di-ubiquitylated at lysines 107 and 113 (YY**K**VDENG**K**ISR) in response to emetine-dependent collisions (*Figure 2—figure supplement 2A*). Interestingly, eS31 is located adjacent to eS10 on the beak of the 40S subunit in close proximity to the disome interface, and its deubiquitylation by USP16 at lysine 113 has recently been implicated in licensing the terminal steps of cytosolic 40S subunit maturation (*Montellese et al., 2020*). While the E3 ligase that ubiquitylates eS31 in response to collisions is not known, our data

reveal a rich and diverse set of E3 ligases and deubiquitylases (*Figure 1—source data 1*) that warrant further characterization.

Important for this EDF1-focused study, our proteomic experiments showed that EDF1's recruitment to collided ribosomes was ZNF598-independent (*Figure 2C–F*). Our data also revealed that the recruitment of the translational repressors GIGYF2•EIF4E2 to collided ribosomes was ZNF598-independent (*Figure 2D*) though these factors have previously been shown to interact with ZNF598 to mediate the translation repression of ARE-element containing cytokine mRNAs (*Tollenaere et al., 2019*). Instead, using complementary interaction analyses, we discovered that GIGYF2 and EIF4E2 were among the top interacting partners of EDF1 (*Figure 5A–B*), and through polysome proteomics, that recruitment of GIGYF2 and EIF4E2 to collided ribosomes depended strongly on the presence of EDF1 (*Figure 6C–D*). Interestingly, our proteomics data revealed that GIGYF2 and EIF4E2 preferentially co-sedimented with heavier polysomes in response to emetine treatment, distinct from other studies that identified these factors predominantly in lighter polysome fractions (*Timpano and Uniacke, 2016*). We suggest that EDF1 is among the early sensors of ribosomal collisions due to its high copy number (*Wiśniewski et al., 2014*). However, if collisions occur stochastically at low levels under basal growth conditions (*Figure 1—figure supplement 1A*, blue trace) (*Han et al., 2020*) and are transient in nature, perhaps EDF1 fails to stably bind the composite inter-ribosomal interface efficiently, and therefore fails to recruit GIGYF2•EIF4E2. As collisions lead to terminal 'dead-end' stalling of ribosomes, longer ribosomal queues accumulate in the heavier fractions of polysomes, allowing for more efficient recruitment of EDF1 and eventually of GIGYF2•EIF4E2. Such a situation is more likely to occur when the clearance machinery is overwhelmed, as anticipated with cells experiencing abundant collisions.

Our functional assays established that EDF1-dependent recruitment of the translational repressors GIGYF2 and EIF4E2 to collided ribosomes initiates a negative feedback loop that prevents new ribosomes from translating certain problematic mRNAs (*Figure 6E* and *Figure 6—figure supplement 2H*). Currently, we do not have evidence for ZNF598-mediated recruitment of GIGYF2•EIF4E2 to these same stall-inducing mRNAs. However, it remains possible that there are semi-redundant mechanisms for recruiting GIGYF2 and EIF4E2 in other situations, or under different environmental perturbations (*Tollenaere et al., 2019*; *Hickey et al., 2020*).

Recent studies have established that terminally stalled ribosomes on problematic mRNAs are disassembled by the RQT complex (*D'Orazio et al., 2019*; *Juszkiewicz et al., 2020*; *Matsuo et al., 2020*). If stalled ribosomes are cleared by the ribosomal rescue machinery, why and when are translational repression complexes recruited to problematic mRNAs? It is possible that when the number of collisions exceed the threshold capacity of the RQT complex, the high local concentration of EDF1 at stalled ribosomes enables recruitment of GIGYF2•EIF4E2 to selectively inhibit translation initiation on these problematic mRNAs. In this capacity, EDF1 functions as a sensor for monitoring the overall extent of ribosomal collisions in the cell. In considering these balances it seems plausible that cell lines tune the expression levels of the ribosomal rescue and translational repression machineries depending on their translational output and capacity. For example, in cell lines where ribosomal rescue is efficient, there may be more modest demands on translational repression and vice versa. To begin to think about these possibilities, we compared expression levels of the rescue and repression machineries across the proteome of 375 cell lines from the Cancer Cell Line Encyclopedia (*Nusinow et al., 2020*) (https://depmap.org) (*Figure 7—figure supplement 1D*). Reassuringly, the expression levels of the translational repression complex (EDF1 and GIGYF2) and the ribosomal rescue machinery (ASCC2 and ASCC3) show strong correlation across cell lines (*Figure 7—figure supplement 1D*). By contrast, the expression profiles of both EDF1 and GIGYF2 show modest but significant anti-correlation with the ASCC3-complex (*Figure 7—figure supplement 1D*). These data hint at the possibility of cell-specific tuning of these various mechanisms for controlling expression of proteins from problematic mRNAs.

While our study defined a multifaceted role for EDF1 in promoting different arms of the ribosome-mediated QC pathway (*Figure 7G*), previous studies of EDF1 and its orthologs in yeast, plants, flies and mammals had identified a role for this factor as a transcriptional coactivator that enhances the transcriptional response of select bZIP transcription factors (such as JUN and GCN4) in response to environmental stress (*Jaimes-Miranda and Chávez Montes, 2020*; *Jindra et al., 2004*; *Kabe et al., 1999*; *Miotto and Struhl, 2006*; *Takemaru et al., 1998*; *Takemaru et al., 1997*). Our transcriptomics data reveal that ribosome collisions trigger a transcriptional response with a strong

and sustained JUN signature (*Figure 7C–D*). We speculate that this sustained response may in part be driven by JUN autoregulating its own transcription (*Angel et al., 1988*; *Wisdom et al., 1999*). More importantly for this study, we find that the collision-dependent transcriptional response strongly depends on EDF1 (*Figure 7E–F*). The cohort of stress-induced genes that show robust transcriptional upregulation in response to ribosomal collisions are transcriptionally attenuated in the absence of EDF1, with JUN exhibiting the most significant dampening. Given previous results describing how EDF1 stabilizes JUN, and amplifies its transcriptional activity (*Jindra et al., 2004*; *Miotto and Struhl, 2006*), our results are consistent with a model where an EDF1•JUN nexus coordinates a transcriptional response to ribosomal collisions (*Figure 7G*). Though our study established that EDF1 localizes on collided ribosomes, we did not identify JUN itself enriched on colliding ribosomes (*Figure 1—source data 1*, *Figure 2—source data 1*, *Figure 6—source data 1*), and observed only a modest enrichment of JUN (~1.6 fold) in proximity-labeling analyses of EDF1 (*Figure 5—source data 2*). While it is possible that the cross-talk between EDF1 and JUN occurs through direct coupling between these two proteins (*Jindra et al., 2004*; *Miotto and Struhl, 2006*), it could also be regulated through JUN N-terminal kinases (JNKs) which associate with ribosomes through RACK1 (*López-Bergami et al., 2005*) and have recently been shown to be activated by ZAKα through direct interactions with colliding ribosomes (*Wu et al., 2020*).

Overall, these data reveal a new paradigm for ribotoxic stress responses that originate in the cytoplasm on colliding ribosomes, and trigger an EDF1-dependent transcriptional response with a strong JUN signature (*Figure 7G*). It will be interesting moving forward to define the sequence of events, on and off the ribosome, that lead to signal transduction across the nuclear-cytoplasmic boundary.

## Materials and methods

**Key resources table**

| Reagent type (species) or resource | Designation | Source or reference | Identifiers | Additional information |
|---|---|---|---|---|
| Antibody | Anti-phospho-p38 (Thr180/Tyr182) (rabbit monoclonal) | Cell Signaling Technology | Cat #4511S RRID:AB_2139682 | WB (1:1000) |
| Antibody | Anti-p38 (rabbit polyclonal) | Cell Signaling Technology | Cat #9212 RRID:AB_330713 | WB (1:1000) |
| Antibody | Anti-β-actin (13E5)-HRP conjugate (rabbit monoclonal) | Cell Signaling Technology | Cat #5125 RRID:AB_1903890 | WB (1:1000) |
| Antibody | Anti-ASCC3 (rabbit polyclonal) | Bethyl Laboratories | Cat #A304-014A RRID:AB_2620362 | WB (1:1000) |
| Antibody | Anti-ZNF598 (rabbit polyclonal) | Bethyl Laboratories | Cat #A305-108A RRID:AB_2631503 | WB (1:2000) |
| Antibody | Anti-EDF1 (rabbit polyclonal) | Abcam | Cat #ab174651 | WB (1:1000) |
| Antibody | Anti-RACK1 (rabbit monoclonal) | Cell Signaling Technology | Cat #5432 RRID:AB_10705522 | WB (1:1000) |
| Antibody | Anti-FLAG (mouse monoclonal) | Sigma | Cat #A8592 RRID:AB_439702 | WB (1:1000) |
| Antibody | Anti-RPL10A (rabbit polyclonal) | Bethyl Laboratories | Cat #A305-062A RRID:AB_2631457 | WB (1:2000) |
| Antibody | Anti-RPS10 (eS10) (rabbit polyclonal) | LS Bio | Cat #LS-C335612-20 | WB (1:500) |
| Antibody | Anti-RPS20 (uS10) (rabbit monoclonal) | Abcam | Cat #ab133776 RRID:AB_2714148 | WB (1:1000) |
| Antibody | Anti-RPS3 (uS3) (rabbit polyclonal) | Abcam | Cat #ab140688 | WB (1:2000) |

*Continued on next page*

*Continued*

| Reagent type (species) or resource | Designation | Source or reference | Identifiers | Additional information |
|---|---|---|---|---|
| Antibody | Anti-GIGYF2 (mouse monoclonal) | Santa Cruz Biotechnology | Cat #Sc-393918 | WB (1:100) |
| Antibody | Anti-EIF4E2 (mouse monoclonal) | Novus Biological | Cat #H00009470-M01 RRID:AB_1505961 | WB (1:500) |
| Antibody | Anti-phospho-c-Jun (Ser73) (rabbit polyclonal) | Cell Signaling | Cat #9164 RRID:AB_330892 | WB (1:1000) |
| Antibody | Anti-c-Jun (60A8) (rabbit monoclonal) | Cell Signaling | Cat #9165 RRID:AB_2130165 | WB (1:1000) |
| Antibody | Anti-Strep II (rabbit polyclonal) | Novus Biologicals | Cat #NBP2-41075 | WB (1:1000) |
| Antibody | Mouse anti-rabbit IgG-HRP (mouse monoclonal) | Santa Cruz Biotechnology | Cat #sc-2357 RRID:AB_628497 | WB (1:5000) |
| Antibody | Goat anti-mouse IgG2a-HRP (goat polyclonal) | Jackson Immuno Research Laboratories Inc | Cat #115-035-2006 | WB (1:5000) |
| Antibody | anti-Diglycyl Lysine (Clone GX41) antibody (mouse monoclonal) | Millipore Sigma | Cat #MABS27 RRID:AB_10807824 | Ubiquitin remnant immunoaffinity profiling |
| Sequence-based reagent | CRISPR: EDF1 sgRNA exon 1 | GPP, Broad Institute | EDF1 sgRNA exon 1 | 5'-GAGCGACTGGGA CACGGTGA-3' |
| Sequence-based reagent | CRISPR: EDF1 sgRNA exon 3 | GPP, Broad Institute | EDF1 sgRNA exon 3 | 5'-ACATTCTATTACC AAGAACA-3' |
| Sequence-based reagent | siRNA: ON-TARGETplus non-targeting control pool siRNA | Horizon Discovery | Cat #D-001810-01-50 | SMARTPool |
| Sequence-based reagent | siRNA: ON-TARGETplus RACK1 siRNA | Horizon Discovery | Cat #L-006876-00-0020 | SMARTPool |
| Sequence-based reagent | siRNA: ON-TARGETplus EDF1 siRNA | Horizon Discovery | Cat #L-009697-00-0020 | SMARTPool |
| Sequence-based reagent | siRNA: ON-TARGETplus Human GIGYF2 siRNA | Horizon Discovery | Cat #L-013918-01-0020 | SMARTPool |
| Sequence-based reagent | siRNA: ON-TARGETplus Human EIF4E2 siRNA | Horizon Discovery | Cat #L-019870-01-0020 | SMARTPool |
| Recombinant DNA reagent | pCMV-GFP-2A-VHP-(K^{AAA})_{20}-2A-ChFP | *Juszkiewicz and Hegde, 2017* | Addgene Cat #105688 | Stalling reporter based on poly(A) sequence |
| Recombinant DNA reagent | pCMV-GFP-2A-VHP-2A-ChFP (linker control) | *Juszkiewicz and Hegde, 2017* | Addgene Cat #105686 | Linker control without the internal poly(A) stalling sequence |
| Recombinant DNA reagent | psiCHECK2-Renilla-2A-3xFLAG-MsXbp1u-2A-Firefly | *Han et al., 2020* | N/A | Stalling reporter based on Xbp1u peptide stalling sequence |
| Recombinant DNA reagent | psiCHECK2-Renilla-2A-3xFLAG-2A-Firefly (linker control) | This study | N/A | Linker control without the internal Xbp1u stalling sequence |
| Recombinant DNA reagent | pOG44 Flp-recombinase expression vector | Thermo Fisher | Cat #V600520 | Expression of Flp recombinase in mammalian cells when co-transfected with pcDNA5/FRT plasmid. |
| Recombinant DNA reagent | pcDNA5-FRT-tetO-FLAG-BirA*-EDF1 | This study | N/A | Human EDF1 tagged with FLAG-BirA* at the N-terminus for mammalian expression. |

*Continued on next page*

*Continued*

| Reagent type (species) or resource | Designation | Source or reference | Identifiers | Additional information |
|---|---|---|---|---|
| Recombinant DNA reagent | pcDNA5-FRT-tetO-EDF1-FL (full-length) Strep II | This study | N/A | Human EDF1 tagged with Strep-II at the C-terminus for mammalian expression. |
| Recombinant DNA reagent | pcDNA5-FRT-tetO-EDF1-N-term (1-74) Strep II | This study | N/A | Human EDF1 truncation (amino acids 1–74) tagged with Strep-II at the C-terminus for mammalian expression. |
| Recombinant DNA reagent | pcDNA5-FRT-tetO-EDF1-C-term (73-148) Strep II | This study | N/A | Human EDF1 truncation (amino acids 73–148) tagged with Strep-II at the C-terminus for mammalian expression. |
| Recombinant DNA reagent | pcDNA5-FRT-tetO-3xFLAG-3C-EDF1 | This study | N/A | Human EDF1 tagged with 3xFLAG-HRV-3C at the N-terminus for mammalian expression. |
| Recombinant DNA reagent | pLentiCrisprV2 | Addgene | Cat #52961 RRID:Addgene_52961 | |
| Cell line (*H. sapiens*) | HEK293T WT | ATCC | CRL-3216 RRID:CVCL_0063 | |
| Cell line (*H. sapiens*) | HCT116 WT | ATCC | CCL-247 RRID:CVCL_0291 | |
| cell line (*H. sapiens*) | HCT116 ΔZNF598 | *Sundaramoorthy et al., 2017* | N/A | |
| cell line (*H. sapiens*) | HCT116 ZNF598-OE | *Sundaramoorthy et al., 2017* | N/A | |
| cell line (*H. sapiens*) | HEK293-Flp-In T-REx-WT | Thermo Fisher | R78007 RRID:CVCL_U427 | |
| Cell line (*H. sapiens*) | HEK293-Flp-In T-REx-ΔZNF598 | *Garzia et al., 2017* | N/A | |
| Cell line (*H. sapiens*) | HEK293-Flp-In T-REx-ΔEDF1 #3–1 | This study | N/A | CRISPR/Cas9 targeting EDF1, clonal selection |
| Cell line (*H. sapiens*) | HEK293-Flp-In T-REx-ΔEDF1ΔZNF598 #1–5 | This study | N/A | CRISPR/Cas9 targeting EDF1 and ZNF598, clonal selection |
| Cell line (*H. sapiens*) | HEK293-Flp-In T-REx-ΔEDF1: EDF1-FL (full-length) Strep II | This study | N/A | ΔEDF1 cells co-transfected with pcDNA5-FRT-tetO-EDF1-FL (full-length) Strep II and pOG44 Flp-recombinase expression vectors, and selected with hygromycin. |
| Cell line (*H. sapiens*) | HEK293-Flp-In T-REx-ΔEDF1: EDF1-N-term (1-74) Strep II | This study | N/A | ΔEDF1 cells co-transfected with pcDNA5-FRT-tetO-EDF1-N-term (1-74) Strep II and pOG44 Flp-recombinase expression vectors, and selected with hygromycin. |
| Cell line (*H. sapiens*) | HEK293-Flp-In T-REx-ΔEDF1: EDF1-C-term (73-148) Strep II | This study | N/A | ΔEDF1 cells co-transfected with pcDNA5-FRT-tetO-EDF1-C-term (73-148) Strep II and pOG44 Flp-recombinase expression vectors, and selected with hygromycin. |

*Continued on next page*

Continued

| Reagent type (species) or resource | Designation | Source or reference | Identifiers | Additional information |
|---|---|---|---|---|
| Cell line (*H. sapiens*) | HEK293-Flp-In T-REx-3xFLAG-3C-EDF1 | This study | N/A | HEK293-Flp-In T-Rex co-transfected with pcDNA5-FRT-tetO-3xFLAG-3C-EDF1 and pOG44 Flp-recombinase expression vectors, and selected with hygromycin. |
| Chemical compound, drug | Emetine | Cayman Chemical | Cat #21048 | Used to induce ribosome collisions |
| Chemical compound, drug | Anisomycin | Sigma-Aldrich | Cat #A9789 | Used to induce ribosome collisions |
| Chemical compound, drug | Doxycycline | Sigma | D9891-10G | Used to induce gene expression in Flp-In T-Rex cell lines |
| Chemical compound, drug | Hygromycin | ThermoFisher | 10687010 | selection antibiotic |
| Chemical compound, drug | Puromycin | InvivoGen | ant-pr-1 | selection antibiotic |
| Other | Trypsin | Promega | V511C | |
| Other | Lys-C | Wako | 129–02541 | |
| Other | SUPERase•In RNase inhibitor | Ambion | AM2696 | |
| Peptide, recombinant protein | RNAse A | Ambion | AM2270 | |
| Peptide, recombinant protein | Superscript III | Invitrogen | 56575 | |
| Other | Gateway BP Clonase II Enzyme mix | ThermoFisher | 11789020 | |
| Other | Gateway LR Clonase II Enzyme Mix | ThermoFisher | 11791020 | |
| Peptide, recombinant protein | Turbo DNAse | ThermoFisher | AM2239 | |
| Other | Phosphatase inhibitor | Cell Signaling Technology | 5870S | |
| Other | cOmplete, EDTA-free Protease Inhibitor Cocktail | Roche | 5056489001 | |
| Other | Mammalian Protease Inhibitor Cocktail | Sigma | P8340-5mL | |
| Commercial assay, kit | Dual-Glo Luciferase Assay System | Promega | E2940 | |
| Commercial assay, kit | Direct-zol RNA Miniprep kit | Zymo Research | R2051 | |
| Commercial assay, kit | Zymo-Seq RiboFree Total RNA Library Kit | Zymo Research | R3000 | |
| Commercial assay, kit | Dynabeads Protein A | Thermo Fisher | 10008D | |
| Commercial assay, kit | Dynabeads Protein G | Thermo Fisher | 10009D | |
| Commercial assay, kit | ANTI-FLAG M2 Affinity Gel | Sigma-Aldrich | A2220 | |
| Commercial assay, kit | PierceStreptavidin-conjugated agarose beads | Thermo Fisher | 20353 | |

*Continued*

| Reagent type (species) or resource | Designation | Source or reference | Identifiers | Additional information |
|---|---|---|---|---|
| Commercial assay, kit | TMT 6plex Label Reagent | Thermo Fisher | 90068 | |
| Commercial assay, kit | TMTpro 16plex Label Reagent | Thermo Fisher | A44520 | |
| Commercial assay, kit | 3M Empore SPE Disks C18 | Sigma | 66883 U | |
| Commercial assay, kit | Sep-Pak C18 Cartridge | Waters Corporation | WAT054960 and WAT054925 | |
| Commercial assay, kit | High pH Reversed-Phase Peptide Fractionation Kit | Thermo Fisher | 84868 | |
| Commercial assay, kit | Pierce Quantitative Colorimetric Peptide Assay | Thermo Fisher | 23275 | |
| Software, algorithm | Maxquant | *Tyanova et al., 2016a* | Version 1.6.10.43 RRID:SCR_014485 | Data analysis, mass spectrometry-based proteomics |
| Software, algorithm | Comet-based (v2018.01 rev.2) in-house software pipeline | *Eng et al., 2013*; *Huttlin et al., 2010* | N/A | Data analysis, mass spectrometry-based proteomics |
| Software, algorithm | Perseus | *Tyanova et al., 2016b* | Version 1.6.10 RRID:SCR_015753 | Statistical analysis of proteomics data |
| Software, algorithm | Differential Expression Proteomics (DEP) | *Zhang et al., 2018* | N/A | Statistical analysis of proteomics data |
| Software, algorithm | InCyte software for Guava easyCyte | Millipore Sigma | 0500–4120 | Data analysis for single-cell flow-cytometry |
| Software, algorithm | STAR | *Dobin et al., 2013* | STAR_2.5.3a_modified RRID:SCR_015899 | RNA-seq aligner |
| Software, algorithm | Salmon | *Patro et al., 2017* | Version 1.2.1 | Transcript quantification, RNA-seq |
| Software, algorithm | DESeq2 | *Love et al., 2014* | Version 3.11 RRID:SCR_015687 | Differential gene expression analysis, RNA-seq |
| Software, algorithm | Gencode | *Frankish et al., 2019* | v33 RRID:SCR_014966 | Genome annotation, RNA-seq |
| Software, algorithm | Custom software (Python 2.7) for RNA sequencing analysis | https://github.com/greenlabjhmi/EDF1_elife_2020 (*Zinshteyn and Green, 2020*; copy archived at https://github.com/elifesciences-publications/EDF1_elife_2020) | | Analyses and visualization of RNA-seq data |
| Software, algorithm | Relion | *Zivanov et al., 2018* | 3.0 and 3.1 RRID:SCR_016274 | Single particle analyses and reconstruction, cryo-EM |
| Software, algorithm | PHENIX | *Adams et al., 2010* | 1.18 | Tool for automated structure refinement, cryo-EM |
| Software, algorithm | WinCOOT | *Emsley et al., 2010* | 0.8.9.2 | Model building, refinement and validation, Cryo-EM |
| Software, algorithm | ChimeraX | *Goddard et al., 2018* | 1.0 | Visualization of cryo-EM data |
| Software, algorithm | GraphPad Prism | GraphPad Software Inc | Version 8.4.1 | Statistical analysis, graphs |

Further information and requests for resources and reagents should be directed to and will be fulfilled by the Lead Contact Rachel Green (ragreen@jhmi.edu).

## Plasmids

The pCMV-GFP-2A-VHP-2A-RFP (linker control) and pCMV-GFP-2A-VHP-(K$^{AAA}$)$_{20}$-2A-RFP dual-fluorescence translation stall reporter plasmids described here (*Figure 6E*, *Figure 6—figure supplement 2A*, *Figure 6—figure supplement 2D–2G*) were a generous gift from Manu Hegde (MRC, Cambridge, UK) and its use has been described elsewhere (*Juszkiewicz and Hegde, 2017*; *Sundaramoorthy et al., 2017*). The psiCHECK2-Renilla-2A-3xFLAG-MsXbp1u-2A-Firefly dual-luciferase translation stall reporter plasmid (*Figure 6—figure supplement 2H–2I*) was a generous gift from Shintaro Iwasaki (*Han et al., 2020*). The EDF1 codon region was recombined into the pcDNA5-FRT-tetO-Flag-BirA* destination using Gateway (Thermo Fisher) cloning methods.

## Cell lines, maintenance, and transfections,

HEK293T (CRL-3216) and HCT116 (CCL-247) parental cell lines were obtained from ATCC; HEK293-Flp-In T-REx (R78007) parental cell lines were obtained Thermo Fisher. CRISPR knockout and overexpression lines derived from parental backgrounds are listed in the Key Resources Table. Cell lines were tested, and reported to be negative for mycoplasma contamination. All cell lines were thawed and grown for more than two passages prior to any experiment. Unless otherwise stated, HEK293 and HCT116 cell lines and their variants were grown in DMEM (high glucose, pyruvate, and L-Glutamine) supplemented with 10% fetal bovine serum (FBS), maintained in a 5% $CO_2$ humidified incubator and passaged every 2–3 days. siRNA knockdown experiments involved two siRNA transfections on consecutive days at a concentration of 10–25 nM siRNA per day, using Lipofectamine RNAiMax (Thermo Fisher) according to the manufacturer's guidelines. All expression plasmid transfections were performed using Lipofectamine 3000 (Thermo Fisher) according to the manufacturer's guidelines. siRNA and plasmid co-transfections (when applicable) were performed using Lipofectamine 3000 (Thermo Fisher) according to the manufacturer's guidelines.

## Generation of ΔEDF1 knockout lines

EDF1 knockout lines were constructed using a CRISPR/Cas9 approach. The following sgRNAs were cloned into the pLentiCRISPRv2 plasmid (Addgene #52961) (*Sanjana et al., 2014*), corresponding to exon 1 and exon 3, respectively, of the EDF1 locus.

> sgRNA_exon1: GAGCGACTGGGACACGGTGA
> sgRNA_exon3: ACATTCTATTACCAAGAACA

Resultant plasmids were sequenced to confirm the appropriate insertion. HEK293 Flp-In T-REx WT and ΔZNF598 cells were seeded into 12-well plates at a density of $3.8 \times 105$ cells/well. The next day, each well was transfected with one of the plasmids containing an EDF1 sgRNA, or a control vector containing no sgRNA. On the following day, wells were treated with 0.95 µg/ml puromycin. After 4 days of selection, cells were taken off puromycin and replaced with fresh DMEM, and allowed to grow for 1-4 more days, at which point they were trypsinized and seeded into 96-well plates at a density of 1 cell per well. Single colonies (verified by eye) were grown to confluency and transferred to larger plates until they were able to be frozen. Potential ΔEDF1 lines were lysed, tested by immunoblotting against an anti-EDF1 antibody, and confirmed ΔEDF1 lines were frozen. The frameshifts observed in the ΔEDF1#3-1 line used in this paper (*Figures 6* and *7*) were confirmed by next generation sequencing as previously described (*Joung et al., 2017*), demonstrating a frameshift deletion of 2 bp and 14 bp, respectively, within exon 1 of the EDF1 locus at >99% of reads (within sequencing error).

## Cell growth analysis

HEK293 Flp-In T-REx WT, ΔEDF1, ΔZNF598, and ΔZNF598;ΔEDF1, cells were seeded into 24-well plates at a low starting density of 7,500 cells/well in 500 uL DMEM + 10% FBS. At each timepoint, cells from three independent wells were washed, trypsinized in 333 µL trypsin, and pooled into a single tube. Cells were stained with trypan blue, and three cell counts were taken from each tube using the Tc20 Automated Cell Counter (BioRad #145–0103).

## Treatment with elongation inhibitors

Emetine (Cayman Chemical) stock solutions were prepared fresh to 100 mM in water, frozen at −20 ˚C, and used within 2 weeks. If frozen, emetine stocks were thawed and equilibrated to room temperature before use. Anisomycin (Sigma-Aldrich) stock solutions were prepared to 94.2 mM (25 mg/ml) in DMSO and frozen at −20 ˚C. HCT116 or HEK293 cells were seeded at 2–4 million cells per 100 mm TC plate and allowed to grow for 48 hr. At 48 hr (approx. 1–2 hr before emetine treatment), cells were replenished with fresh DMEM supplemented with 10% FBS. Ribosomal collisions were induced by adding emetine to a final concentration of 1.8 µM directly to media, gently swirling the plate, and returning the cells to 37 ˚C for 15 min, after which cells were lysed. Unless indicated, treatment with other elongation inhibitors were also for 15 min at 37 ˚C with following concentrations; emetine low dose (1.8 µM) or high-dose (360 µM); anisomycin low dose (0.19 µM) or high dose (75 µM).

## Sample preparation and cell lysis

Cells were lysed by aspirating media, immediately rinsed with warm PBS (37 ˚C, pH 7.4; Thermo Fisher) supplemented with 360 µM emetine to freeze ribosomes in situ, and lysed by adding 300–400 µl ice cold lysis buffer dropwise to the plate (lysis buffer: 50 mM HEPES pH 7.4, 100 mM KOAc, 15 mM Mg(OAc)$_2$, 5% Glycerol, 0.5% Triton X-100 supplemented with 360 µM emetine (Cayman Chemical), 1x phosphatase inhibitor cocktail (Cell Signaling Technology), 10 mM N-ethylmaleimide (freshly prepared; Sigma Aldrich), 2x cOmplete EDTA-free Protease Inhibitor Cocktail tablets (Roche), 1 mM PMSF (Sigma), 1x mammalian protease inhibitor cocktail (Sigma), 1 mM TCEP (Gold-Bio; TCEP was omitted when samples were prepared for immunoprecipitation) and eight units/ml Turbo DNase (Thermo Fisher)). Plates were swirled to distribute lysis buffer; cells were scraped from the plate using a cell scraper, gently pipetted to homogenize the cell lysate, and transferred to ice for 5–10 min to complete lysis. Lysates were clarified by brief centrifugation at 8000x$g$ (5–7 min, 4 ˚C), and the clarified supernatant was transferred to a fresh tube on ice. Lysates were prepared fresh and used immediately for sucrose gradients and immunoprecipitations to avoid artifacts associated with freeze-thawing. For immunoblots of whole cell lysates, samples were sometimes flash frozen in liquid nitrogen and stored at −80 ˚C.

## Sucrose gradient fractionation

Stock solutions of 10x gradient buffer (250 mM HEPES pH 7.4, 1M KOAc, 50 mM Mg(OAc)$_2$) and 60% (w/v) sucrose in water were prepared, filter-sterilized through a 0.22 µm filter, and stored at room temperature. On the day of the experiment, gradients were prepared from two freshly-made sucrose buffers containing 1x gradient buffer, 1 mM TCEP, 360 µM emetine, 200 units SUPERase•In RNase inhibitor (Thermo Fisher), and sucrose to the appropriate concentration (usually 10% and 50% (w/v) sucrose buffers, unless otherwise stated). To prepare gradients, 6 ml of 10% sucrose buffer was added to a SW41 ultracentrifuge polypropylene tube (Seton Scientific), after which 6 ml of 50% sucrose was added to the bottom of the tube using a 10 ml syringe and cannula; 10–50% sucrose gradients prepared on a Biocomp Gradient Master. Gradients were stored at 4 ˚C until use on the same day. To normalize RNA loading, triplicate A260 measurements from 1:10 dilutions of each clarified sample were read using a NanoDrop UV-Vis spectrophotometer. Background A260 measurement from lysis buffer was subtracted from each reading. Equal RNA load (~100–300 µg, depending on the experiment) was layered on top of each sucrose gradient; gradients were ultra-centrifuged in a Beckman SW41 swinging bucket rotor (40,000 rpm; 105 min). Gradients were fractionated and UV (A260) absorbance across 10–50% sucrose gradients was measured using a top-down Biocomp Piston Gradient Fractionator as per manufacturer's instructions. For polysome proteomics individual or pooled fractions were flash frozen in liquid nitrogen and processed as described in the polysome proteomics section below. For SDS-PAGE and immunoblotting, proteins from individual fractions were TCA-precipitated using standard protocol and stored at −20 ˚C overnight (*Link and LaBaer, 2011*). The following day, TCA-precipitated fractions were centrifuged at 20,000x$g$ (30 min, 4 ˚C), the supernatant aspirated, pellets washed (x 3) in 500 µl acetone and centrifuged at 20,000x$g$ (10 min, 4 ˚C), the supernatant aspirated after each wash; after the final wash step pellets were vacuum-dried briefly (~5 min, 42 ˚C) in a vacuum evaporator, resuspended in Laemmli buffer, pH neutralized with Tris-HCl pH 8.0, boiled (95 ˚C, 5 min) and resolved by SDS-PAGE.

## RNase A treatment

Clarified cell lysates were treated with RNase A (Ambion) using the following condition – 1 µg RNase A was added per 100 µg RNA in a 250 µl reaction volume, shaken at 500 rpm (20 min, 25 ˚C) on a table-top thermo-mixer (Eppendorf); the reaction was quenched by the addition of SUPERase•In RNase inhibitor (~200 units per 100 µg RNA). RNase A digested lysates were layered on top of 10–35% sucrose gradients and processed as described above.

## UV treatment

Cells in DMEM + 10% FBS were removed from 37 ˚C incubator and placed in a Stratalinker UV 1800 Crosslinker with lids removed. Cells were treated with 0.06 J/cm$^2$ (*Figure 1K*) or 0.02 J/cm$^2$ (*Figure 2—figure supplement 2E*), returned to the 37 ˚C incubator and recovered for 30 min (*Figure 1K*) or 1 hr (*Figure 2—figure supplement 2E*), after which samples were lysed and processed as described previously.

## Immunoblotting

Samples for immunoblotting were prepared either directly from clarified cell lysates resuspended in Laemmli buffer to 1X, or from TCA-precipitated sucrose gradient fractions resuspended in 6X Laemmli buffer (see section on 'Sucrose Gradient Fractionation') and boiled at 95 ˚C for 5 min. For immunoblotting of whole cell lysates, triplicate A260 measurements from 1:10 dilutions of each clarified sample were read using a NanoDrop UV-Vis spectrophotometer. Background A260 measurement from lysis buffer was subtracted from each reading. Normalized samples were loaded into 4–12% bis-tris polyacrylamide gels (Criterion Bio-Rad); gel electrophoresis was performed in MES running buffer (150V; 1 hr). Gels were transferred to PVDF membranes using Trans-Blot Turbo Transfer System (Bio-Rad) per the manufacturer's instructions. Membranes were blocked in 5% non-fat milk (Santa Cruz Biotechnology) resuspended in TBST (30 min, 25 ˚C) followed by overnight incubation with primary antibody in 5% non-fat milk in TBST at 4 ˚C, followed by 4 × 10 min washes in TBST at 25 ˚C, followed by incubation with the secondary antibody in 5% non-fat milk in TBST (1 hr, 25 ˚C), followed by 4 × 10 min washes in TBST. All incubation steps were performed with gentle rocking. Primary and secondary antibodies were used at recommended concentrations (key resources table). Western blots were visualized by HRP chemiluminescence using Super Signal West HRP substrate (Thermo Fisher); films were developed in a dark room at multiple exposures.

## Stall reporter assays and flow cytometry

For siRNA mediated knockdown studies, cells were transfected using Lipofectamine RNAiMAX (Thermo Fisher) according to manufacturer guidelines. Cells were then transfected with the pCMV-GFP-2A-VHP-(K$^{AAA}$)$_{20}$-2A-ChFP dual-fluorescence stall reporter plasmid using Lipofectamine 3000 (Thermo Fisher) according to manufacturer guidelines 24 hr after the siRNA transfection, or 24 hr after seeding for non-siRNA studies. Single-cell RFP and GFP fluorescence intensities for 10,000 individual events were measured 48 hr following reporter transfection on a Millipore Sigma Guava easy-Cyte benchtop flow cytometer (Millipore Sigma) using 532 nm and 488 nm excitation lasers respectively. Flow data were analyzed using InCyte software for Guava easyCyte HT systems (Millipore Sigma). For reporter assays using Flp-In cell lines, the reporter was transfected 48 hr prior to analysis by cytometry. Transgene expression was induced using doxycycline (1 µg/ml) 24 hr after reporter transfection. Prism (version 8.4.2) was used for data and statistical analyses (*Figure 6E*, *Figure 6—figure supplement 2A*, *Figure 6—figure supplement 2D–2G*). The ROUT method (Q = 10%) was used to identify outliers. p-values were determined by one-way ANOVA and Tukey's post hoc correction for multiple comparisons.

## Luciferase assay

HEK293 Flp-In TREx WT, ΔEDF1, ΔZNF598 cells, and ΔEDF1ΔZNF598 cells were seeded in a 96-well plate at a density of 15000 cells per well in 200 µl DMEM supplemented with 10% FBS. The following day, wells were transfected with 100 ng of psiCHECK2-Renilla-2A-3xFLAG-MsXbp1u-2A-Firefly stalling reporter (*Han et al., 2020*) or the psiCHECK2-Renilla-2A-3xFLAG-2A-Firefly linker control plasmid using Lipofectamine 3000 according to manufacturer guidelines. Approximately 48 hr later, Renilla and Firefly Luciferase activities were measured using the Dual-Glo Luciferase Assay System

(Promega, #E2940) in a Synergy H1 microplate reader (BioTek). Renilla (RLuc) and Firefly Luciferase (Fluc) values for six biological replicates for each condition were averaged and the Fluc:RLuc ratio was computed.

## Polysome proteomics (related to *Figure 1*, *Figure 2*, *Figure 6* and *Figure 1—source data 1*, *Figure 2—source data 1* and *Figure 6—source data 1*)

### Sample preparation and digestion

Collected sucrose gradient fractions were supplemented with Urea (6 M final) (*Figure 1*) or SDS (1% final) (*Figure 2* and *Figure 6*) and subjected to disulfide bond reduction with 5 mM TCEP (room temperature, 10 min) and alkylation with 25 mM chloroacetamide (room temperature, 20 min) followed by TCA precipitation, prior to protease digestion. Samples were resuspended in 100 mM EPPS, pH 8.5 containing 0.1% RapiGest and digested at 37°C for 2 hr with LysC (*Figure 1*) protease at a 200:1 protein-to-protease ratio. Trypsin was added at a 100:1 protein-to-protease ratio and the reaction was incubated for 6 hr at 37°C. Following incubation, digestion efficiency of a small aliquot was tested. Tandem mass tag labeling of each sample was performed by adding indicated amount of the 20 ng/μl stock of TMT or TMTpro reagent along with acetonitrile to achieve a final acetonitrile concentration of approximately 30% (v/v). 5 μl of TMTpro 16plex reagent was added for *Figures 2* and *6*; 4 μl of TMT 6plex reagent was added for *Figure 1*. Following incubation at room temperature for 1 hr, labeling efficiency of a small aliquot was tested, and the reaction was then quenched with hydroxylamine to a final concentration of 0.5% (v/v) for 15 min. The TMT-labeled samples were pooled together at a 1:1 ratio. The sample was vacuum centrifuged to near dryness, resuspended in 5% formic acid for 15 min, centrifuged at 10000 × g for 5 min at room temperature and subjected to C18 solid-phase extraction (SPE) (Sep-Pak, Waters).

### Off-line basic pH reversed-phase (BPRP) fractionation
#### Relevant to *Figure 2*

Dried peptides were fractionated according to manufacturer's instructions using High pH reversed-phase peptide fractionation kit (Thermo Fisher Scientific) for a final six fractions and subjected to C18 StageTip desalting prior to MS analysis.

#### Relevant to *Figure 6*

Dried TMT-labeled sample was resuspended in 100 μl of 10 mM $NH_4HCO_3$ pH 8.0 and fractionated using BPRP HPLC (*Paulo et al., 2016*). Briefly, samples were offline fractionated over a 90 min run, into 96 fractions by high pH reverse-phase HPLC (Agilent LC1260) through an aeris peptide xb-c18 column (Phenomenex; 250 mm x 3.6 mm) with mobile phase A containing 5% acetonitrile and 10 mM $NH_4HCO_3$ in LC-MS grade $H_2O$, and mobile phase B containing 90% acetonitrile and 10 mM $NH_4HCO_3$ in LC-MS grade $H_2O$ (both pH 8.0). The 96 resulting fractions were then pooled in a non-continuous manner into 24 fractions (as outlined in Supplemental Figure 5 of *Paulo et al., 2016*) and 12 fractions (even numbers) were used for subsequent mass spectrometry analysis. Fractions were vacuum centrifuged to near dryness. Each consolidated fraction was desalted via StageTip, dried again via vacuum centrifugation, and reconstituted in 5% acetonitrile, 1% formic acid for MS analysis.

### Liquid chromatography and tandem mass spectrometry
#### Relevant to *Figure 1*

Mass spectrometry data were collected using an Orbitrap Fusion Lumos mass spectrometer, coupled to a Proxeon EASY-nLC1200 liquid chromatography (LC) pump (Thermo Fisher Scientific). Peptides were separated on a 100 μm inner diameter microcapillary column packed in house with ~35 cm of Accucore150 resin (2.6 μm, 150 Å, Thermo Fisher Scientific, San Jose, CA) with a gradient consisting of 5–22% (0–100 min), 22–28% (100–110 min) (ACN, 0.1% FA) over a total 120 min run at ~550 nL/min. For analysis, we loaded 1/3 of each fraction onto the column. To reduce ion interference compared to $MS^2$ quantification, each analysis used the Multi-Notch $MS^3$-based TMT method (*McAlister et al., 2014*), combined with newly implemented Real Time Search analysis software (*Erickson et al., 2019*; *Schweppe et al., 2020*). The scan sequence began with an $MS^1$ spectrum

(Orbitrap analysis; resolution 120,000 at 200 Th; mass range 350–1400 m/z; automatic gain control (AGC) target $1 \times 10^6$; maximum injection time 240 ms). Precursors for MS$^2$ analysis were selected using a 3 s TopSpeed method. MS$^2$ analysis consisted of collision-induced dissociation (quadrupole ion trap analysis; Rapid scan rate; AGC $2.5 \times 10^4$; isolation window 0.7 Th; normalized collision energy (NCE) 35; maximum injection time 60 ms). Monoisotopic peak assignment was used, previously interrogated precursors were excluded using a dynamic window (120 s ± 7 ppm), and dependent scan was performed on a single charge state per precursor. Following acquisition of each MS$^2$spectrum, a synchronous-precursor-selection (SPS) API-MS$^3$ scan was collected on the top 10 most intense ions b or y-ions matched by the online search algorithm in the associated MS$^2$ spectrum (*Erickson et al., 2019*; *Schweppe et al., 2020*). MS$^3$ precursors were fragmented by high energy collision-induced dissociation (HCD) and analyzed using the Orbitrap (NCE 65; AGC $2.5 \times 10^5$; maximum injection time 200 ms, resolution was 15,000 at 200 Th). The closeout was set at two peptides per protein per fraction, so that MS$^3$s were no longer collected for proteins having two peptide-spectrum matches (PSMs) that passed quality filters (*Schweppe et al., 2020*).

## Relevant to *Figure 2*

Mass spectrometry data were collected using an Orbitrap Fusion Lumos mass spectrometer, coupled to a Proxeon EASY-nLC1200 liquid chromatography (LC) pump (Thermo Fisher Scientific). Peptides were separated on a 100 μm inner diameter microcapillary column packed in house with ~35 cm of Accucore150 resin (2.6 μm, 150 Å, Thermo Fisher Scientific, San Jose, CA) with a gradient consisting of 4–14% (0–70 min), 14–21% (70–80 min) (ACN, 0.1% FA) over a total 90 min run at ~550 nl/min. For analysis, we loaded 1/3 of each fraction onto the column. To reduce ion interference compared to MS$^2$ quantification, each analysis used the Multi-Notch MS$^3$-based TMT method (*McAlister et al., 2014*), combined with newly implemented Real Time Search analysis software (*Erickson et al., 2019*; *Schweppe et al., 2020*). The scan sequence began with an MS$^1$ spectrum (Orbitrap analysis; resolution 120,000 at 200 Th; mass range 350–1400 m/z; automatic gain control (AGC) target $1 \times 10^6$; maximum injection time 50 ms). Precursors for MS$^2$ analysis were selected using a 3 s Top-Speed method. MS$^2$ analysis consisted of collision-induced dissociation (quadrupole ion trap analysis; Rapid scan rate; AGC $2.5 \times 10^4$; isolation window 0.7 Th; normalized collision energy (NCE) 35; maximum injection time 35 ms). Monoisotopic peak assignment was used, previously interrogated precursors were excluded using a dynamic window (120 s ± 10 ppm), and dependent scan was performed on a single charge state per precursor. Following acquisition of each MS$^2$spectrum, a synchronous-precursor-selection (SPS) API-MS$^3$ scan was collected on the top 10 most intense ions b or y-ions matched by the online search algorithm in the associated MS$^2$ spectrum (*Erickson et al., 2019*; *Schweppe et al., 2020*). MS$^3$ precursors were fragmented by high energy collision-induced dissociation (HCD) and analyzed using the Orbitrap (NCE 45; AGC $2.5 \times 10^5$; maximum injection time 200 ms, resolution was 50,000 at 200 Th). The closeout was set at two peptides per protein per fraction, so that MS$^3$s were no longer collected for proteins having two peptide-spectrum matches (PSMs) that passed quality filters (*Schweppe et al., 2020*).

## Relevant to *Figure 2H* and *Figure 6*

Mass spectrometry data were collected using an Orbitrap Fusion Lumos mass spectrometer combined with a high-field asymmetric waveform ion mobility spectrometry (FAIMS) Pro interface, coupled to a Proxeon EASY-nLC1200 liquid chromatography (LC) pump (Thermo Fisher Scientific). Peptides were separated on a 100 μm inner diameter microcapillary column packed in house with ~35 cm of Accucore150 resin (2.6 μm, 150 Å, Thermo Fisher Scientific, San Jose, CA) with a gradient consisting of 5–15% (0–70 min), 15–23% (70–85 min) (ACN, 0.1% FA) over a total 95 min run at ~550 nL/min. For analysis, we loaded 1/3 of each fraction onto the column. To reduce ion interference compared to MS$^2$ quantification, each analysis used the Multi-Notch MS$^3$-based TMT method (*McAlister et al., 2014*), combined with newly implemented Real Time Search analysis software (*Erickson et al., 2019*; *Schweppe et al., 2020*) and the FAIMS Pro Interface. The scan sequence began with an MS$^1$ spectrum (Orbitrap analysis; resolution 120,000 at 200 Th; mass range 400–1600 m/z; automatic gain control (AGC) target $8 \times 10^5$; maximum injection time 50 ms). Precursors for MS$^2$ analysis were selected using a cycle type of 1.25 s/CV method (FAIMS CV=-40/–60/-80 previously optimized for TMT multiplexed samples *Schweppe et al., 2019*). MS$^2$ analysis consisted of collision-induced dissociation (quadrupole ion trap analysis; Rapid scan rate; AGC $1.0 \times 10^4$; isolation

window 0.7 Th; normalized collision energy (NCE) 35; maximum injection time 35 ms). Monoisotopic peak assignment was used, precursor fit filter was used (70% for a fit window of 0.7 Th), previously interrogated precursors were excluded using a dynamic window (150 s ± 10 ppm), and dependent scan was performed on a single charge state per precursor. Following acquisition of each MS$^2$ spectrum, a synchronous-precursor-selection (SPS) API-MS$^3$ scan was collected on the top 10 most intense ions b or y-ions matched by the online search algorithm in the associated MS$^2$ spectrum (*Schweppe et al., 2020*). MS$^3$ precursors were fragmented by high energy collision-induced dissociation (HCD) and analyzed using the Orbitrap (NCE 45; AGC 2.5 × 10$^5$; maximum injection time 200 ms, resolution was 50,000 at 200 Th). The closeout was set at two peptides per protein per fraction, so that MS$^3$s were no longer collected for proteins having two peptide-spectrum matches (PSMs) that passed quality filters (*Schweppe et al., 2020*).

## Data analysis (relevant to *Figures 1*, *2* and *6*)

Mass spectra were processed using a Comet-based (v2018.01 rev.2) in-house software pipeline (*Eng et al., 2013*; *Huttlin et al., 2010*). Spectra were converted to mzXML using a modified version of ReAdW.exe. Database searching included all canonical entries from the human Reference Proteome UniProt database (SwissProt – 2019–01), as well as an in-house curated list of contaminants. This database was concatenated with one composed of all protein sequences in the reversed order. Trypsin was used as the digestion enzyme, two missed cleavages were allowed, and the minimal peptide length was set to seven amino acids. Searches were performed using a 20 ppm precursor ion tolerance for total protein level analysis. The recommended product ion parameters for ion trap ms/ms were used (1.0005 tolerance, 0.4 offset (mono masses), theoretical fragment ions = 1). TMT tags on lysine residues and peptide N termini (+229.163 Da for Amino-TMT or +304.2071 Da for TMTpro) and carbamidomethylation of cysteine residues (+57.021 Da) were set as static modifications, while oxidation of methionine residues (+15.995 Da) was set as a variable modification. Peptide-spectrum matches (PSMs) were adjusted to a 1% false discovery rate (FDR) and PSM filtering was performed using a linear discriminant analysis, as described previously (*Huttlin et al., 2010*), while considering the following parameters: Comet Log Expect and Diff Seq. Delta Log Expect, missed cleavages, peptide length, charge state, and precursor mass accuracy. For protein-level comparisons, PSMs were identified, quantified, and collapsed to a 1% peptide false discovery rate (FDR) and then collapsed further to a final protein-level FDR of 1% using the Picked FDR method (*Savitski et al., 2015*). For TMT-based reporter ion quantitation, we extracted the summed signal-to-noise (S:N) ratio for each TMT channel and found the closest matching centroid to the expected mass of the TMT reporter ion (integration tolerance of 0.003 Da). Reporter ion intensities were adjusted to correct for the isotopic impurities of the different TMT reagents according to the manufacturer's specifications. Moreover, protein assembly was guided by principles of parsimony to produce the smallest set of proteins necessary to account for all observed peptides. Proteins were quantified by summing reporter ion counts across all matching PSMs using in-house software, as described previously (*Huttlin et al., 2010*). PSMs with poor quality, MS$_3$ spectra with more than 4 TMT reporter ion channels missing, or isolation specificity less than 0.5 (0.2 for *Figure 6*), or with TMT reporter summed signal-to-noise ratio that were less than 200 (100 for *Figure 1*) or had no MS$_3$ spectra were excluded from quantification.

Protein quantification values were exported for further analysis in Microsoft Excel and Perseus (*Tyanova et al., 2016b*) and statistical test and parameters used are indicated in the corresponding **Source Data Tables**. Briefly, Welch's t-test analysis was performed to compare two datasets, using s0 parameter (in essence a minimal fold change cut-off) and correction for multiple comparison was achieved by the permutation-based FDR method, both functions that are built-in in Perseus software. Spectrum annotation for RPS10 diGly site (*Figure 2—figure supplement 2F*) was generated using IPSA (*Brademan et al., 2019*).

## EDF1 co-immunoprecipitation (related to *Figure 5A* and *Figure 5— source data 1*)

HEK293 cells were seeded at 4 million cells per plate and allowed to grow for 48 hr. At 48 h cells were replenished with fresh DMEM supplemented with 10% FBS. Ribosomal collisions were induced by adding emetine to a final concentration of 1.8 µM directly to media, gently swirling the plate,

and returning the cells to 37 °C for 15 min, after which cells were lysed in the following lysis buffer (50 mM HEPES pH 7.4, 100 mM KOAc, 15 mM Mg(OAc)$_2$, 5% Glycerol) supplemented with 0.5% Triton-X-100, 360 µM emetine, 1x phosphatase inhibitor cocktail (Cell Signaling Technology), 10 mM N-ethylmaleimide, 2x cOmplete EDTA-free Protease Inhibitor Cocktail tablets (Roche), 1 mM PMSF (Sigma), 1x mammalian protease inhibitor cocktail (Sigma) and eight units/ml Turbo DNase (Thermo Fisher). Lysates were clarified as described previously and equal amounts of lysates for untreated (UT) and low dose emetine treated (EL) samples were incubated with Dynabeads Protein A coupled EDF1 antibody for 2 hr (at 4 °C) with gentle rocking. EDF1 antibody (Abcam # ab174651) was used at a concentration of 3.6 µg/mg of protein in clarified lysate. Dynabeads Protein A used according to manufacturer guidelines and preincubated with EDF1 antibody prior to addition to clarified lysate. Following incubation, the samples bound to Dyna-Mag magnet as per manufacturer guidelines, the flow-through was removed, and the Dynabeads Protein A-Ab-Ag complex was washed with lysis buffer supplemented with 0.1% Triton-X-100, 360 µM emetine (4 × 10 min, 4 °C), followed by 4 × 10 min (4 °C) washes with lysis buffer not containing glycerol or detergent. Proteins were eluted from the beads using 50 mM glycine pH 2.8, pH neutralized and processed for MS.

## Protein digestion

Protein extracts (~10 µg) were diluted up to 300 µl in 10 mM triethyl ammonium bicarbonate (TEAB) and were reduced with 15 µl of 7.5 mg/ml DL-dithiothreitol (DTT) (60°C, 1 hr). After cooling to room temperature, samples were alkylated with 15 µl of 18.5 mg/ml iodoacetamide for 15 min at room temperature in the dark. Reduced and alkylated proteins were buffer-exchanged on a 30 kDa molecular weight spin cartridge (Amicon Ultra 0.5 ml, Millipore Sigma) and washed four times with 400 µl 10 mM TEAB. Proteins were digested overnight at 37°C on the filter with 300 µl Trypsin (20 µg in 3 ml 10 mM TEAB, Promega Sequencing Grade Modified Trypsin). Additional Trypsin (100 µl of 10 mg/ml) was added the next morning (37°C, 1 hr). Peptides were removed from the top of the filter and the filter was washed twice with 300 2% acetonitrile, 0.1% formic acid. All washes were combined and dried.

## Liquid Chromatography and mass spectrometry

Peptides were analyzed by liquid chromatography interfaced with tandem mass spectrometry (LC/MS/MS) using an Easy-LC 1000 UPLC system (Thermo Fisher) interfaced with an Orbitrap Q-Exactive Plus Mass Spectrometer (Thermo Fisher). As part of the desalting step using the Oasis plates (Waters Corporation), the peptides were dissolved in 100 µl 0.1%TFA, washed with 0.1%TFA, but then eluted in a step-wise fashion using the following basic pH buffers: 10 mM TEAB (pH 8.5), followed by 5%, 10%, 25%, and 50% acetonitrile in 10 mM TEAB. The first two fractions were combined due to low complexity, then all fractions were dried. The four fractions were resuspended in 20 µl loading buffer (2% acetonitrile in 0.1% formic acid) and analyzed by reverse phase liquid chromatography coupled to tandem mass spectrometry. Peptides (25%, approx. 0.5 µg) were loaded onto a C18 trap (S-10 µM, 120 Å, 75 µm x 2 cm, YMC, Japan) and subsequently separated on an in-house packed PicoFrit column (75 µm x 200 mm, 15 u, +/- 1 µm tip, New Objective) with C18 phase (ReproSil-Pur C18-AQ, 3 µm, 120 Å, www.dr-maisch.com) using 2–90% acetonitrile gradient at 300 nl/min over 120 min. Eluting peptides were sprayed at 2.0 kV directly into the Q-Exactive Plus.

Survey scans (full MS) were acquired from 350 to 1800 m/z with data-dependent monitoring with a loop count of 15. Each precursor individually isolated in a 1.4 Da window and fragmented using HCD activation collision energy 28 and 15 s dynamic exclusion, first mass being 120 m/z. Precursor and the fragment ions were analyzed at resolutions 70,000 and 35,000, respectively, with automatic gain control (AGC) target values at 3e6 with 50 ms maximum injection time (IT) and 1e5 with 100 ms maximum IT, respectively.

## Data analyses

Raw data were processed and analyzed using the MaxQuant (1.6.7.0) software suite (*Tyanova et al., 2016a*). four fractions (corresponding to individual RAW files; UT_F1-F4; EL_F1-F4) were set for the untreated (UT) and low dose emetine treated (1.8 µM, EL) samples. Default settings were used except that 'Match between runs' was turned on to transfer peptide identification from an LC-MS run, in which the peptide has been identified by MS/MS, to another LC-MS run, in which no MS/MS

data for the same peptide was acquired or no peptide was assigned (*Tyanova et al., 2016a*). Search parameters were as follows: a maximum of two missed cleavages were allowed, cysteine carbamidomethyl was included as a fixed modification, and variable modifications included oxidation of methionine, protein N-terminal acetylation, deamidation of glutamine and asparagine, and K-GG ubiquitin remnant on lysines. Trypsin was used as the digestion enzyme, and the minimal peptide length was set to seven amino acids. Searches were performed using a 20-ppm precursor ion tolerance for total protein level analysis. Database search was performed with Andromeda against Uniprot human database (UP000005640_9606.fasta; downloaded on 09/10/2018) with common serum contaminants and enzyme sequences. False discovery rate (FDR) was set to 1% at peptide spectrum match (PSM) and protein level. Minimum peptide count required for protein quantification was set to two. Protein groups were further analyzed using the Perseus (*Tyanova et al., 2016b*). Common contaminants, reverse proteins and proteins only identified by site were filtered out. LFQ values were transformed to $\log_2$ space and intensity distributions were checked to ensure that data were normally distributed.

## BioID proximity-labeling proteomics (related to *Figure 5B* and *Figure 5—source data 2*)

The EDF1 codon region was recombined into the FRT-tetO-Flag-BirA* destination using Gateway (Thermo Fisher) cloning methods. 293 Flp-In T-REx cell lines were transfected with Flp recombinase vector (pOG44) and FRT-tetO-expression vectors and stable cell lines were selected with hygromycin. Protein expression was either uninduced or induced by adding 1 µg/ml doxycycline to the growth media for 16–20 hr prior to cell harvesting. Both induced and uninduced conditions were supplemented with 50 µM biotin 16 hr before harvesting cells. Frozen cell pellets were lysed in mammalian cell lysis buffer (0.5% NP-40, 150 mM NaCl, 50 mM Tris pH 7.8, protease inhibitors) at 4°C at an approximate 2:1 (v:v) ratio per pellet. Lysates were sonicated and clarified by centrifugation. Total protein was quantified using a BCA protein assay (Pierce). Lysates were mixed with 80 µl (1:1 slurry) of streptavidin-conjugated agarose beads. After overnight incubations, the resin was washed 3X in wash buffer (0.1% NP-40, 150 mM NaCl, 50 mM Tris pH 7.8, protease inhibitors) followed by three washes in cold PBS. Trypsin (400 ng) was added to the washed resin and incubated overnight. Trypsin digested samples were desalted using the C18 stage-tip method. The desalted peptides were vacuum dried and reconstituted with 12 µl of peptide reconstitution buffer (5% Formic acid/5% Acetonitrile) for the LC-MS/MS analysis.

### Liquid chromatography and tandem mass spectrometry

The samples were analyzed by nLC-MS/MS using a Q-Exactive mass spectrometer (Thermo Scientific, San Jose, CA) coupled with an EASY-nLC 1000 (Thermo Scientific) chromatography system. Briefly, peptides were first separated by reverse-phase chromatography using a fused silica microcapillary column (75 µm ID, 15 cm) packed with C18 reverse-phase resin (ReproSil-pur 120 C18-AQ, 1.9 µm, Dr. Maisch GmbH) using an in-line nano-flow EASY-nLC 1000 UHPLC. Peptides were eluted over a 100 min 2–30% ACN gradient, a 5 min 30–60% ACN gradient, a 5 min 60–95% ACN gradient, with a final 10 min step at 0% ACN for a total run time of 120 min at a flow rate of 250 nl/min. All gradient mobile phases contained 0.1% formic acid. MS/MS data were collected in a data-dependent fashion using a top 10 method with a full MS mass range from 400 to 1800 m/z, 70,000 resolution, and an AGC target of 3e6. MS2 scans were triggered when an ion intensity threshold of 1e5 was reached with a maximum injection time of 60 ms. Peptides were fragmented using a normalized collision energy setting of 25. A dynamic exclusion time of 40 s was used, and the peptide match setting was disabled. Singly charged ions, charge states above eight and unassigned charge states were excluded. The RAW files were searched on the Maxquant software (version 1.6.10.43) against the UniProt Human reference proteome database (downloaded in year 2017). For the Maxquant analysis, default parameters were used except the following changes- The 'Label Free Quantification' (LFQ) and 're-quantification' options were enabled in the group specific parameters settings and the 'match between runs' option was enabled in the global parameters settings. The statistical analysis was done on the Maxquant output file 'proteinGroups.txt' by the interactive analysis using the Differential Expression Proteomics (DEP) Shiny apps in the R-studio environment (*Zhang et al., 2018*). Briefly, proteins were filtered that were identified in 2 out of 3 replicates of at least one condition.

Filtered protein intensity values were normalized using the Variance Stabilizing Normalization. Missing values were imputed using the MiniProb method by randomly selecting values from a Gaussian distribution centered on a minimal value of the dataset. Protein-wise linear models combined with empirical Bayes statistics were used for the differential enrichment analysis. Fold change ratio and the p-values were calculated. Proteins with $p<0.05$ and log2 fold change $>1$ in comparison with at least one of the controls were considered as significant.

## Ubiquitin remnant immunoaffinity profiling (related to *Figure 2—figure supplement 2A* and *Figure 2—figure supplement 2—source data 1*)

HEK293 Flp-In T-REx WT were seeded at 8 million cells per 15 cm dish and allowed to grow for 48 h. At 48 h cells were replenished with fresh DMEM supplemented with 10% FBS. Cells were either left untreated (UT), or treated with low dose emetine (1.8 µM, EL) for 15 min. Cells from 3 x 15 cm dishes were combined for each replicate. Three biological replicates were used for each condition (UT x 3 or EL x 3; that is total of 9 plates per condition). Following emetine treatment, cells were quickly rinsed with warm PBS (37°C, pH 7.4; Thermo Fisher) and lysed immediately in denaturing lysis buffer (8 M Urea, 50 mM HEPES pH 7.4, 100 mM KOAc, 1 mM sodium fluoride (NaF), 1 mM β-glycerophosphate (β-Gly), 1 mM sodium orthovanadate, and 5 mM N-ethylmaleimide (NEM)). The lysate was clarified by centrifugation at 20,000xg for 15 min at 25°C to pellet insoluble material. Protein concentration of the clarified supernatants was determined by standard BCA assay; equal amount of protein per sample (~20 mg) was used for each replicate. The clarified supernatant was reduced (5 mM DTT, 55°C, 30 min) and alkylated (2 µg/ml iodoacetamide, 25°C, 15 min in the dark), diluted to 4M urea with 50 mM HEPES pH 7.4 lysis buffer (not supplemented with urea), digested with LysC (1:100 enzyme: substrate (w/w)) for 2 h at 37°C, further diluted to 1M urea with 50 mM HEPES pH 7.4 lysis buffer (not supplemented with urea), supplemented with 1 mM $CaCl_2$, followed by overnight digestion with trypsin (TPCK treated, Sigma) at 37°C (1:100 enzyme: substrate (w/w)). Overnight digestion was stopped by addition of 0.4% TFA followed by brief centrifugation at 300xg (15 min, 25°C) to remove insoluble aggregates; peptides from each sample were desalted and purified using Sep-Pak C18 columns, and eluted sequentially with 3 ml (x 2) 50% ACN and 0.5% HAcO. The eluate was flash frozen in liquid nitrogen, stored at -80°C for 4 days, and lyophilized for ~2 days to remove residual TFA. The lyophilized peptides were resuspended in immunoprecipitation (IP) buffer (10 mM $Na_2HPO_4$, 50 mM NaCl, 50 mM MOPS pH 7.2). Peptides were immunoprecipitated with mouse anti-Diglycyl Lysine (Clone GX41) antibody (Millipore Sigma MABS27) coupled to Dynabeads Protein G for 2 h at 4°C. Following incubation, the samples bound to Dyna-Mag magnet, flow-through was removed, and the Dynabeads Protein A-Ab-Ag complex was washed (4x) with IP buffer, followed by 2 x 1 ml washes with PBS (10 min). Peptides were eluted with 0.1% TFA in water, concentrated and desalted by stage-tip chromatography and analyzed by liquid chromatography interfaced with tandem mass spectrometry (LC/MS/MS) using an Easy-LC 1000 UPLC system (Thermo Fisher) interfaced with an Orbitrap Q-Exactive Plus Mass Spectrometer (Thermo Fisher) as described previously in the section for "EDF1 Co-Immunoprecipitation".

## Data analyses

Raw data were processed and analyzed using the MaxQuant (1.6.7.0) software suite (*Tyanova et al., 2016a*) as described previously. Default settings were used except that 'Match between runs' was turned on to transfer peptide identification from an LC-MS run, in which the peptide has been identified by MS/MS, to another LC-MS run, in which no MS/MS data for the same peptide was acquired or no peptide was assigned (*Tyanova et al., 2016a*). Search parameters were as follows: Trypsin/LysC were selected as the digestion enzymes, a maximum of two missed cleavages were allowed, cysteine carbamidomethyl and K-$\varepsilon$-GG ubiquitin remnant on lysines were included as a fixed modification, and variable modifications included oxidation of methionine, protein N-terminal acetylation, deamidation of glutamine and asparagine; the minimal peptide length was set to 7 amino acids. Searches were performed using a 20-ppm precursor ion tolerance. Database search was performed with Andromeda against Uniprot human database (UP000005640_9606.fasta; downloaded on 09/10/2018) with common serum contaminants and enzyme sequences. False discovery rate (FDR) was set to 1% at peptide spectrum match (PSM). The statistical analysis was done on the Maxquant output file "peptides.txt" using Perseus (*Tyanova et al., 2016b*). Common contaminants, reverse

proteins and proteins only identified by site were filtered out. LFQ values were transformed to $\log_2$ space and intensity distributions were checked to ensure that data was normally distributed. Peptides were filtered that were identified in 2 out of 3 replicates of at least one condition. Missing values were imputed using functions that are built-in in Perseus (1.6.7) software (*Tyanova et al., 2016b*). Two-sided t-test analysis was performed to compare the UT and EL datasets, using s0 parameter (in essence a minimal fold change cut-off) and a truncation based on permutation-based FDR method (default: 0.05), both functions that are built-in in Perseus software.

## Cryo-EM (related to *Figure 3*, *Figure 3—figure supplement 1*, *Figure 3—figure supplement 2*, and *Figure 4*)

### EDF1 affinity purification

HEK293 Flp-In T-Rex cells expressing EDF1 with an N-terminal 3xFLAG-3C Protease cleavage site tag were lysed in lysis buffer (20 mM HEPES pH 7.5, 150 mM KOAc, 5 mM $MgCl_2$, 0.5% IGEPAL CA-630 (Sigma), 0.1 mM $Na_3VO_4$, 0.5 mM NaF, 1 mM DTT, 1x cOmplete EDTA-free Protease Inhibitor Cocktail tablets (Roche)). The crude lysate was consecutively sonicated four times for 10 s followed by 30 s on ice each. The lysate was clarified by two subsequent centrifugation steps at 2960 x g and 4°C for 15 min and 36,500 x g and 4°C for 25 min. The resulting supernatant was incubated with ANTI-FLAG M2 Affinity Gel (Sigma) at 4°C for 120 min. The affinity beads were washed twice with NP-40 washing buffer (20 mM HEPES pH 7.5, 150 mM KOAc, 5 mM $MgCl_2$, 0.01% IGEPAL CA-630 (Sigma), 0.1 mM $Na_3VO_4$, 0.5 mM NaF, 1 mM DTT) and once with Nikkol washing buffer (20 mM HEPES pH 7.5, 150 mM KOAc, 5 mM $MgCl_2$, 0.05% octaethylene glycol monododecyl ether, 1 mM DTT). After transferring the beads to a 1 mL Mobicol spin-column (MoBiTech) they were washed once with Nikkol washing buffer. For elution, the beads were incubated in elution buffer (20 mM HEPES pH 7.5, 150 mM KOAc, 5 mM MgCl2, 0.05% octaethylene glycol monododecyl ether, 1 mM DTT, 0.352 mg/mL 3C Protease (homemade)) at 4°C for 60 min. The eluate was collected by centrifugation and subjected to cryo-EM.

### In vitro translation of SDD1 mRNA and purification of RNCs

SDD1 stalled ribosomes were generated and purified as described previously (*Matsuo et al., 2020*). The purified RNCs were applied to a 10–50% sucrose gradient, and ribosomal fractions were separated via centrifugation for 3 hr at 202,048 x g at 4°C in a SW40 rotor.

### Cryo-EM analysis of 3x-FLAG-EDF-ribosome and SDD1 trisome complexes

Freshly prepared samples of the EDF1-80S or SDD1 trisome fraction were applied to holey carbon support grids (R3/3 with 2 nm continuous carbon support, Quantifoil), which had been glow discharged at $2.1 \times 10^{-1}$ mbar for 20 s. Grids were incubated for 45 s at 4°C and subsequently plunge frozen in liquid ethane using a Vitrobot Mark IV (FEI Company). Data were collected on a Titan Krios at 300 kV using a K2 Summit direct electron detector (Gatan) with a nominal pixel size of 1.059 Å and a defocus range from 0.5 to 2.5 µm at low-dose conditions. For each movie, 40 frames with approximately 1.12 e- $Å^{-2}$ exposure were gain corrected and aligned using MotionCor2 (*Zheng et al., 2017*). Contrast-transfer function (CTF) parameters of the summed micrographs were estimated with Gctf (*Zhang, 2016*), before micrographs were manually screened for quality.

### Data processing of the EDF1 data set

The EDF1-80S data set was processed using Relion 3.1 (*Zivanov et al., 2018*). After two-dimensional (2D) classification, 95,832 particles from 4260 micrographs were subjected to a 3D classification. First, 80S states and low-resolution particles of the ribosome were separated in five 3D classes. Approximately 85% of the particles represented post-state ribosomes with high EDF1 occupancy which were refined to an overall resolution of 3.1 Å. Post-processing, CTF corrections and a focused refinement with a soft mask around the 40S subunit yielded an overall resolution of 2.9 Å and improved the density of EDF1 for interpretation. This map was filtered according to local resolution with a negative B-factor of 20 and used for model building.

## Data processing of the SSD1 trisome data set

The trisome data set was processed as an 80S dataset in Relion 3.0 and Relion 3.1 (*Zivanov et al., 2018*). In brief, individual 80S particles were picked using the Laplacian of Gaussian mode of Relion Autopicker and subjected to 2D classification. A total 398,371 particles from 4109 micrographs were selected after 2D classification. Initial refinement followed by masked 3D classification into five classes were performed, with the 3D classification focusing on differentiating between tRNA states. Of the observed ribosomes, 23% were occupied by P/P tRNA, 64% by A/P, P/E tRNA (in three classes) and 13% had A-site tRNA. The three classes representing rotated ribosomes with A/P and P/E tRNAs were merged for further processing. Sub-classification of these ribosomes into three classes with a mask around the 40S beak and rRNA helix 16 gave one class (31%) with extra density where helix 16 was shifted compared to the other classes. After CTF-refinement and subsequent sub-classification in two classes, one well resolved class with 77% of the particles was observed. Focused 3D refinement with a soft mask around the 40S subunit yielded a map with an overall resolution of 3.0 Å which was filtered according to local resolution with a negative B-factor of 30 and used for model building.

## Model building

To generate molecular models, we used our previously refined models of stalled yeast and human 80S ribosome (*Thoms et al., 2020*) (PDB 6ZMI) and disome (*Ikeuchi et al., 2019*) (PDB 6I7O). First, individual subunits and tRNAs were fitted as rigid bodies into the densities. These models were then remodeled in COOT (*Emsley et al., 2010*) and refined in Phenix (*Adams et al., 2010*). Cryo-EM structures and models were displayed using UCSF ChimeraX (*Goddard et al., 2018*). Detailed statistics of model refinements and validations are listed in *Figure 3—source data 1*.

## RNA sequencing (related to *Figure 7*)

HEK293 Flp-In T-REx WT and ΔEDF1 cells were seeded in 6-well plates at $2 \times 10^5$ cells/well and allowed to grow for 48 hr. At 48 h cells were replenished with fresh DMEM supplemented with 10% FBS. Cells were either left untreated (UT), or treated with low dose (1.8 µM) emetine for 30 min (EL30) or 120 min (EL120) to induce ribosomal collisions, after which samples were harvested by aspiration of media and addition of 1 ml TRIzol reagent (Thermo Fisher) directly to each well of the plate. Samples were homogenized by pipetting up and down several times. RNA was extracted with Direct-zol RNA Miniprep kit (R2051) following the manufacturer's protocol. RNA sequencing libraries were prepared from 1 µg of total RNA using the Zymo-Seq RiboFree Total RNA Library Kit following the manufacturer's instructions and sequenced by GENEWIZ on an Illumina Hiseq 2500 using 150 nt paired-end reads. Raw sequencing data were deposited in the GEO database under the accession number GSE149565. Secure token for reviewers: uzajoeeultgrpsr.

## RNA-sequencing data analysis

Genome and transcript sequences and annotations were downloaded from Gencode v31 (*Frankish et al., 2019*). Transcript levels were quantified using Salmon (*Patro et al., 2017*) with optional parameters –libType A –gcBias –seqBias –validateMappings. Reads or TPMs (transcripts per million) were summed across all transcripts for a given gene for downstream analysis. To compute fold changes and statistical significance, total counts for each gene were rounded to the nearest integer and processed with DESeq2 (*Love et al., 2014*). To determine the effect of EDF1Δ on the emetine response of each gene, we used and experimental design with and interaction term in DESeq2: design = ~ genotype + condition + genotype:condition, where genotype grouped samples by their EDF1 status, and condition grouped samples by emetine treatment (untreated (UT), 1.8 µM emetine 30 min (EL30), 1.8 µM emetine 120 min (EL120)). Statistically significant genes were chosen based on a Benjamini-Hochberg adjusted p-value of 0.01.

## Acknowledgements

We would like to thank Colin Wu for thoughtful discussions and critical reading of this manuscript; members of the Green lab, especially Jamie Wangen, for advice and helpful feedback throughout; Dr. Aitor Garzia and Dr. Thomas Tuschl for their kind gift of the HEK293 Flp-In T-Rex ΔZNF598 cell

line (*Garzia et al., 2017*); Dr. Andrew Holland and Gina Lomastro for sharing the pLentiCRISPRv2 plasmid and assistance generating the ΔEDF1 and ΔEDF1ΔZNF598 cell lines; Dr. Shintaro Iwasaki for sharing the psiCHECK2-Renilla-2A-3xFLAG-MsXbp1u-2A-Firefly plasmid; Dr. Elizabeth J Grayhack for initial reagents for Mbf1; Dr. Otto Berninghausen for cryo-EM data collection; Lauren DeVine and Dr. Robert N Cole at the JHMI Mass Spectrometry and Proteomics Facility for mass-spec support.

# Additional information

## Competing interests

Rachel Green: eLife, reviewing editor; advisory board member for Moderna, Inc, FL63, and the Cystic Fibrosis Foundation. J Wade Harper: J.W.H. is a reviewing editor, eLife, a founder and advisory board member for Caraway Therapeutics, Inc, and an advisor for X-Chem Inc. Eric J Bennett: E.J.B is an advisor and scientific advisory board member for Plexium. The other authors declare that no competing interests exist.

## Funding

| Funder | Grant reference number | Author |
|---|---|---|
| Howard Hughes Medical Institute | | Rachel Green |
| National Institute of General Medical Sciences | R37GM059425 | Rachel Green |
| National Institute of Neurological Disorders and Stroke | R37NS083524 | Wade Harper |
| National Institute on Aging | AG011085 | Wade Harper |
| National Institute of General Medical Sciences | DP2GM119132 | Eric J Bennett |
| National Institute of General Medical Sciences | 5K99GM135450-02 | Boris Zinshteyn |
| Jane Coffin Childs Memorial Fund for Medical Research | | Niladri K Sinha |
| National Institute of General Medical Sciences | T32GM007240 | Danielle M Garshott |
| National Science Foundation | DGE-1650112 | Danielle M Garshott |
| Deutsche Forschungsgemeinschaft | GRK 1721 | Roland Beckmann |
| Deutsche Forschungsgemeinschaft | QBM (Quantitative Biosciences Munich) Graduate School Fellowships | Katharina Best Timo Denk |
| National Institutes of Health | RO1GM132129 | Joao A Paulo |

The funders had no role in study design, data collection and interpretation, or the decision to submit the work for publication.

## Author contributions

Niladri K Sinha, Conceptualization, Data curation, Formal analysis, Supervision, Funding acquisition, Investigation, Methodology, Writing - original draft, Project administration, Writing - review and editing; Alban Ordureau, Conceptualization, Data curation, Formal analysis, Investigation, Methodology, Writing - original draft, Writing - review and editing; Katharina Best, Conceptualization, Data curation, Formal analysis, Investigation, Methodology, Writing - review and editing; James A Saba, Investigation, Methodology, Writing - review and editing; Boris Zinshteyn, Formal analysis, Investigation, Methodology, Writing - review and editing; Elayanambi Sundaramoorthy, Amit Fulzele, Danielle M Garshott, Investigation, Methodology; Timo Denk, Formal analysis, Investigation, Writing - review

and editing; Matthias Thoms, Conceptualization, Supervision, Funding acquisition, Methodology, Project administration, Writing - review and editing; Joao A Paulo, Conceptualization, Data curation, Formal analysis, Supervision, Funding acquisition, Methodology, Project administration, Writing - review and editing; J Wade Harper, Eric J Bennett, Conceptualization, Data curation, Formal analysis, Supervision, Funding acquisition, Investigation, Methodology, Project administration, Writing - review and editing; Roland Beckmann, Conceptualization, Data curation, Supervision, Funding acquisition, Methodology, Project administration, Writing - review and editing; Rachel Green, Conceptualization, Data curation, Supervision, Funding acquisition, Writing - original draft, Project administration, Writing - review and editing

### Author ORCIDs
Niladri K Sinha (iD) https://orcid.org/0000-0002-9143-495X
Alban Ordureau (iD) https://orcid.org/0000-0002-4924-8520
James A Saba (iD) http://orcid.org/0000-0003-3453-8151
Boris Zinshteyn (iD) http://orcid.org/0000-0003-0103-3501
Elayanambi Sundaramoorthy (iD) http://orcid.org/0000-0003-1256-9758
Danielle M Garshott (iD) http://orcid.org/0000-0002-4357-1781
J Wade Harper (iD) https://orcid.org/0000-0002-6944-7236
Eric J Bennett (iD) http://orcid.org/0000-0002-1201-3314
Roland Beckmann (iD) https://orcid.org/0000-0003-4291-3898
Rachel Green (iD) https://orcid.org/0000-0001-9337-2003

### Decision letter and Author response
Decision letter https://doi.org/10.7554/eLife.58828.sa1
Author response https://doi.org/10.7554/eLife.58828.sa2

## Additional files

### Supplementary files
• Transparent reporting form

### Data availability

Raw mass spectrometry data associated with the following Figures have been deposited in MassIVE repository: Source data for all proteomics-based plots are provided in Source data tables.Figure 1, Figure 1-figure supplement 1: MSV000085423; Figure 2, Figure 2-figure supplement 1: MSV000085419; Figure 2-figure supplement 2A: MSV000085422; Figure 2H, Figure 2-figure supplement 2F: MSV000085425; Figure 5A: MSV000085424; Figure 5B, Figure 5-figure supplement 1B-1C: MSV000085421; Figure 6B-6C, Figure 6-figure supplement 1: MSV000085420. Raw sequencing data were deposited in the GEO database under the accession number GSE149565. The cryo-EM structures reported here have been deposited in the Protein Data Bank under the accession codes 6ZVH (EDF1•ribosome) and 6ZVI (Mbf1•ribosome), and in the Electron Microscopy Data Bank under the accession codes EMD-11456 (EDF1•ribosome) and EMD-11457 (Mbf1•ribosome).

The following datasets were generated:

| Author(s) | Year | Dataset title | Dataset URL | Database and Identifier |
|---|---|---|---|---|
| Sinha NK, Ordureau A, Harper JW, Bennett EJ, Green R | 2020 | EDF1 coordinates cellular responses to ribosome collisions, related to Figure 1, Figure 1-figure supplement 1 | https://doi.org/10.25345/C5T70C | MassIVE, 10.25345/C5T70C |
| Sinha NK, Ordureau A, Harper JW, Bennett EJ, Green R | 2020 | EDF1 coordinates cellular responses to ribosome collisions, related to Figure 2, Figure 2-figure supplement 1 | https://doi.org/10.25345/C5B71D | MassIVE, 10.25345/C5B71D |
| Sinha NK, Ordureau A, Harper JW, | 2020 | EDF1 coordinates cellular responses to ribosome collisions, | https://doi.org/10.25345/C5Z11X | MassIVE, 10.25345/C5Z11X |

| | | | | | |
|---|---|---|---|---|---|
| Bennett EJ, Green R | | | Figure 2-figure supplement 2A | | |
| Sinha NK, Ordureau A, Harper JW, Bennett EJ, Green R | | 2020 | EDF1 coordinates cellular responses to ribosome collisions, related to Figure 2H, Figure 2-figure supplement 2F | https://doi.org/10.25345/C5JQ47 | MassIVE, 10.25345/C5JQ47 |
| Sinha NK, Ordureau A, Harper JW, Bennett EJ, Green R | | 2020 | EDF1 coordinates cellular responses to ribosome collisions, related to Figure 5A | https://doi.org/10.25345/C5PH7J | MassIVE, 10.25345/C5PH7J |
| Sinha NK, Ordureau A, Harper JW, Bennett EJ, Green R | | 2020 | EDF1 coordinates cellular responses to ribosome collisions, related to Figure 5B, Figure 5-figure supplement 1B-1C | https://doi.org/10.25345/C52Q58 | MassIVE, 10.25345/C52Q58 |
| Sinha NK, Ordureau A, Harper JW, Bennett EJ, Green R | | 2020 | EDF1 coordinates cellular responses to ribosome collisions, related to Figure 6B-6C, Figure 6-figure supplement 1 | https://doi.org/10.25345/C56H6T | MassIVEm, 10.25345/C56H6T |
| Sinha NK, Zinshteyn B, Green | | 2020 | EDF1 binds collided ribosomes and facilitates recruitment of translational repressors GIGYF2/EIF4E2 and initiates JUN-mediated transcriptional response | https://www.ncbi.nlm.nih.gov/geo/query/acc.cgi?acc=GSE149565 | NCBI Gene Expression Omnibus, GSE149565 |
| Best KM, Denk T, Cheng J, Thoms M, Berninghausen O, Beckmann R | | 2020 | EDF1-ribosome complex | https://www.rcsb.org/structure/6ZVH | RCSB Protein Data Bank, 6ZVH |
| Best KM, Denk T, Cheng J, Thoms M, Berninghausen O, Beckmann R | | 2020 | Mbf1-ribosome complex | https://www.rcsb.org/structure/6ZVI | RCSB Protein Data Bank, 6ZVI |
| Best KM, Denk T, Cheng J, Thoms M, Berninghausen O, Beckmann R | | 2020 | EDF1-ribosome complex | https://www.ebi.ac.uk/pdbe/entry/emdb/EMD-11456 | Electron Microscopy Data Bank, EMD-11456 |
| Best KM, Denk T, Cheng J, Thoms M, Berninghausen O, Beckmann R | | 2020 | Mbf1-ribosome complex | https://www.ebi.ac.uk/pdbe/entry/emdb/EMD-11457 | Electron Microscopy Data Bank, EMD-11457 |

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
