## [Decision Letter]

**Acceptance summary:**

The paper shows that human EDF1 (Endothelial differentiation-related factor 1. in other species termed, MBF1; Multiprotein-bridging factor 1), which is known as a transcriptional co-activator that links between TATA-box-binding proteins and sequence-specific transcription factors to activate transcription of downstream genes, is recruited to collided ribosomes during translational distress, and engender a transcriptional response to ribotoxic stress. The paper concludes that EDF1 binds collided ribosomes, recruits translational repressors GIGYF2/EIF4E2, and initiates a transcriptional response. Thus, the transcriptional coactivation is likely to be mediated through ribosome collisions. The paper provides the first structure of a ribosome collision sensor.

**Decision letter after peer review:**

Thank you for submitting your article "EDF1 binds collided ribosomes, recruits translational repressors GIGYF2/EIF4E2 and initiates a transcriptional response" for consideration by *eLife*. Your article has been reviewed by three peer reviewers, including Nahum Sonenberg as the Reviewing Editor and Reviewer #1, and the evaluation has been overseen by James Manley as the Senior Editor.

The reviewers have discussed the reviews with one another and the Reviewing Editor has drafted this decision to help you prepare a revised submission.

Summary:

The authors show that EDF1 as a novel protein recruited to collided ribosomes during translational distress, and report the transcriptional response to ribotoxic stress. They demonstrated that (1) RACK1, but not ZNF598 is required for the recruitment of EDF1 to collided ribosomes. (2) ZNF598-mediated ubiquitylation of eS10 and uS10 and ZAKα-mediated phosphorylation of p38 by the emetine treatment is moderately reduced in the EDF1-KO cells. (3) EDF1 recruits GIGYF2•EIF4E2 to collided ribosomes and is required for the increase of JUN, ATF3 mRNAs by emetine treatment. Based on these results, the authors propose that EDF1 binds collided ribosomes, recruits translational repressors GIGYF2/EIF4E2, and initiates a transcriptional response. EDF1 has been reported to be involved in transcription coactivation of the JUN-mediated response, thus the coactivation is likely to be mediated through ribosome collisions.

Essential revisions:

(1) The reduction of ribosome stalling by K(AAA)20 and XBP1u in ZNF598 KO is dependent on EDF1 (Figure 3 and Figure 5E). In ribosome stalling by K(AAA)20, the ZNF598-independent function of EDF1 is dependent on GIGYF2•EIF4E2 (Figure 5E). It is necessary to confirm that EDF also functions in ribosome stalling on XBP1u mRNA in ZNF598-independent but GIGYF2•EIF4E2-independent manner.

(2) ZNF598-mediated ubiquitylation of eS10 and uS10 and ZAKα-mediated phosphorylation of p38 by emetine treatment is moderately reduced in the EDF1-KO cells. However, EDF1 slightly represses ribosome stalling by K(AAA)20 and XBP1u in the presence of ZNF598 (Figure 3 and Figure 5E). This may be due to the reduced association of ZNF598 in the EDF1-KO cells. This possibility could be examined by the determination of the distribution of ZNF598 in the EDF1-KO cells.

(3) EDF1 recruits GIGYF2•EIF4E2 to collided ribosomes and is required for the increase of JUN, ATF3 mRNAs by the emetine treatment (Figure 5D and Figure 6). However, it is unclear whether the increase of JUN, ATF3 mRNAs by the emetine treatment depends on the EDF1-mediated recruitment of GIGYF2•EIF4E2 to collided ribosomes. It should be examined whether the JUN, ATF3 mRNAs are increased by emetine treatment when EDF1 is not recruited to the collided ribosomes, like the RACK1-KO background.

4) The authors need to show that EDF1-fusion proteins behave like the native one (i.e. associate with collided ribosomes, activates JUN, and other properties). They also should either remove the data for the frameshifting panel or use another reporter.

5) Some of the claims for the role of EDF1 in recruiting ZNF598 need to be tuned down.

6) The authors need to explain why they used different approaches to inhibit the EDF1 function, as they went back and forth using siRNA and CRISPR knockout, as well as the drugs used to induce ribosome collisions.

We include the unabridged comments of the 3 referees for better explanation of the revisions required, and for those expected in follow up work.

Reviewer #1:

Ribosome quality control is an important mechanism responsible for resolving stalled ribosomes in eukaryotic cells. Important questions, particularly on the mechanisms of detection of the target mRNAs and the fate of the mRNA are unanswered. Particularly, if and how the cell blocks further rounds of translation of the mRNA which hosts the stalled ribosome remains to be revealed. In this manuscript Sinha, et al. report the discovery of Endothelial differentiation-related factor 1 (EDF1) as a key factor in regulation of RQC in response to ribosome stalling. The report that EDF1 is recruited to the stalled ribosome and this event is required for the ZNF598 mediated ubiquitination of the small ribosomal subunit proteins eS10 and uS10. Notably, they also show that EDF1 interacts with and recruits GIGYF2 and EIF4E2 proteins, which have previously been linked to miRNA-induced translational repression of target mRNAs, to the stalled ribosome.

This is an interesting report which reveals new aspects of the RQC mechanism and is particularly important for its implication in a better understanding of the translational repression induced by RQC. An aspect that has been largely neglected in favor of mRNA decay induced by RQC. However, some of the findings are counterintuitive and need to be further explored/explained.

- For instance, the majority of prior studies showed that 4EHP is restricted to very light fractions (e.g. Timpano and Uniacke, 2016 and Garzia, et al., 2017). But in this study, 4EHP and GIGYF2 are heavy polysome fractions. Overall, considering the translational repressor function of these two proteins, it would make more sense to observe them in light fractions. Could the authors explain the possible reasons for this inconsistency? It is also interesting to know why GIGYF2/4EHP stays associated with a stalled ribosome in WT cells, long enough to be detected by polysome profiling. Isn't RQC-dependent mRNA decay supposed to be so fast that it degrades the target mRNA almost instantly? In that case, would not it be expected to see these proteins only in very heavy fractions?

- GIGYF2 and 4EHP recruitment to the stalled ribosome are claimed to be dependent on EDF1 but independent of ZNF598. ZNF598 has been demonstrated to interact with GIGYF2 via its PPGL motifs (Morita et al., 2012). Therefore, ZNF598 would have been the most obvious candidate for direct binding and recruitment of GIGYF2 and via that 4EHP to the stalled ribosome. In fact, the recent findings of Weissman's group (GIGYF2 and 4EHP Inhibit Translation Initiation of Defective Messenger RNAs to Assist Ribosome-Associated Quality Control. bioRxiv) support that hypothesis. The findings in this manuscript are at variance with the latter findings, as they argue for a ZNF598-independent mechanism for the recruitment of GIGYF2 and 4EHP. The authors need to address this discrepancy. How does EDF1 do that? Does it also have a PPGL motif, which directly binds to the GYF motif on GIGYF2?

Reviewer #2:

The manuscript by Sinha and colleagues used quantitative mass-spec approaches to identify EDF1 as a factor that is recruited to collided ribosomes. The authors then went on to explore the role of its recruitment in quality control and transcriptional response to ribotoxic stress. Human EDF1 and its homologues (called MBF1 in most organisms) have been extensively studied for their role in transcription in stress responses; they bind TATA-binding protein and bridge an interaction with transcription factors. Recently, however, yeast Mbf1 has been shown to likely associate with ribosomes through interactions with Asc1. The factor is also required to maintain reading frame of the ribosome during stalling. Furthermore, the archaeal homologue interacts with the 30S and 70S ribosome subunits.

In this study, the authors expand on this ribosome-centric role of the human factor, and provide some insights into how these two seemingly unrelated functions of the factor might be coordinated by collided ribosomes. They found the factor to be required for recruiting the translation-initiation inhibitors GIGYF2/EIF4E2 to problematic mRNAs that stall the ribosome to inhibit initiation on them. Interestingly, the factor has been implicated in transcription coactivation of JUN-mediated response, and the authors in this paper show that this coactivation is likely to be mediated through ribosome collisions.

Overall, I thought the paper provided some important details about the role of EDF1 in quality control and transcriptional regulation. I found the data on its recruitment to collided ribosomes and its requirement for RACK1 to be especially convincing. Having said that, for certain parts I felt the authors overinterpreted the data. This is especially true for the effect of the factor on Znf598 and frameshifting. Below I summarize my concerns that I feel need to be addressed before the paper can be published:

1) I was curious as to why the authors did not carry out MS or western-blotting analyses on sucrose-gradient fractions of ribosomes treated with high concentration of emetine. Did I miss that data? Also does the factor remain on polysomes following RNAse treatment.

2) It is not clear to me how the data in Figure 3A report on frameshifting. Is Fluc out of frame with Rluc? Shouldn't EDF1 deletion promote more readthrough, unless fluc is in frame. In that case, a different reporter is needed. In yeast deletion of Mbf1 on its own is sufficient to promote frameshifting.

3) I found the data on ribosomal protein ubiquitination not convincing. The changes in the levels of ubiquitinated ribosomal proteins seem small. If EDF1 indeed affects ZNF598 function, one would expect its deletion to have a similar effect on the stalling reporter.

4) Does the immunoprecipitation shown in Figure 4 work for ribosome-bound factor; is the antibody capable of recognizing the factor when it associates with the ribosome? More important, does the BirA-fused protein associate with the collided ribosomes? Does it rescue the ∆EDF1 cells?

5) Why did the authors keep changing between using siRNAs and ∆EDF1?

6) Could the authors speculate on why the deletion of ZNF598 results in less translation of the reporter (Figure 2G)? What were the GFP fluorescence values normalized to? The data in Figure 5E shows opposite results. Deletion of the factor results in slight increase of GFP in this case.

Reviewer #3:

The authors identified EDF1 as a novel protein recruited to collided ribosomes during translational distress with proteomics combined with sucrose gradient fractionation. They demonstrated that (1) RACK1, but ZNF598 is required for the recruitment of EDF1 to collided ribosomes. (2) ZNF598-mediated ubiquitylation of eS10 and uS10 and ZAKa-mediated phosphorylation of p38 by the emetine treatment is moderately reduced in the EDF1-KO cells. (3) EDF1 recruits GIGYF2•EIF4E2 to collided ribosomes and is required for the increase of JUN, ATF3 mRNAs by the emetine treatment. Based on these results, the authors propose that EDF1 binds collided ribosomes, recruits translational repressors GIGYF2/EIF4E2, and initiates a transcriptional response.

Major comments:

(1) The reduction of ribosome stalling by K(AAA)20 and XBP1u in ZNF598 KO is dependent on EDF1 (Figure 3 and Figure 5E). In ribosome stalling by K(AAA)20, the ZNF598-independent function of EDF1 is dependent on GIGYF2•EIF4E2 (Figure 5E). It is necessary to confirm that EDF also functions in ribosome stalling on XBP1u mRNA in ZNF598-independent but GIGYF2•EIF4E2-independent manner.

(2) ZNF598-mediated ubiquitylation of eS10 and uS10 and ZAKa-mediated phosphorylation of p38 by the emetine treatment is moderately reduced in the EDF1-KO cells. However, EDF1 slightly represses ribosome stalling by K(AAA)20 and XBP1u in the presence of ZNF598 (Figure 3 and Figure 5E). This may be due to the reduced association of ZNF598 in the EDF1-KO cells. This possibility could be examined by the determination of the distribution of ZNF598 in the EDF1-KO cells.

(3) EDF1 recruits GIGYF2•EIF4E2 to collided ribosomes and is required for the increase of JUN, ATF3 mRNAs by the emetine treatment (Figure 5D and Figure 6). However, it is unclear whether the increase of JUN, ATF3 mRNAs by the emetine treatment depends on the EDF1-mediated recruitment of GIGYF2•EIF4E2 to collided ribosomes. It should be examined whether the JUN, ATF3 mRNAs are increased by the emetine treatment in the condition that EDF1 is not recruited to the collided ribosomes, like the RACK1-KO background.

---

## [Author Response]

Summary:The authors show that EDF1 as a novel protein recruited to collided ribosomes during translational distress, and report the transcriptional response to ribotoxic stress. They demonstrated that (1) RACK1, but not ZNF598 is required for the recruitment of EDF1 to collided ribosomes. (2) ZNF598-mediated ubiquitylation of eS10 and uS10 and ZAKα-mediated phosphorylation of p38 by the emetine treatment is moderately reduced in the EDF1-KO cells. (3) EDF1 recruits GIGYF2•EIF4E2 to collided ribosomes and is required for the increase of JUN, ATF3 mRNAs by emetine treatment. Based on these results, the authors propose that EDF1 binds collided ribosomes, recruits translational repressors GIGYF2/EIF4E2, and initiates a transcriptional response. EDF1 has been reported to be involved in transcription coactivation of the JUN-mediated response, thus the coactivation is likely to be mediated through ribosome collisions.Essential revisions:1) The reduction of ribosome stalling by K(AAA)20 and XBP1u in ZNF598 KO is dependent on EDF1 (Figure 3 and Figure 5E). In ribosome stalling by K(AAA)20, the ZNF598-independent function of EDF1 is dependent on GIGYF2•EIF4E2 (Figure 5E). It is necessary to confirm that EDF also functions in ribosome stalling on XBP1u mRNA in ZNF598-independent but GIGYF2•EIF4E2-independent manner.

We have addressed this concern in (Figure 6—figure supplement 2H-2I) of our revised manuscript. In summary, we have tested the role of EDF1 (and ZNF598) in translation repression of the XBP1u stalling reporter. Consistent with results from the polyA-mediated stalling reporter (Figure 6E of revised manuscript), we show that loss of EDF1, but not ZNF598, leads to an increase in the bulk translation output from the XBP1u-stalling reporter (Figure 6—figure supplement 2H). Together, these data indicate that EDF1-dependent recruitment of the translational repressors GIGYF2•EIF4E2 to problematic mRNAs functions broadly on the multiple ribosome-stalling sequences that we tested.

2) ZNF598-mediated ubiquitylation of eS10 and uS10 and ZAKα-mediated phosphorylation of p38 by emetine treatment is moderately reduced in the EDF1-KO cells. However, EDF1 slightly represses ribosome stalling by K(AAA)20 and XBP1u in the presence of ZNF598 (Figure 3 and Figure 5E). This may be due to the reduced association of ZNF598 in the EDF1-KO cells. This possibility could be examined by the determination of the distribution of ZNF598 in the EDF1-KO cells.

The reviewers raise two interesting points here. The first point demands an explanation for the modest decrease in the readthrough ratio (RFP/GFP or Fluc/RLuc) from our stalling reporters in EDF1ZNF598 compared to ZNF598 lines (Figure 6-figure supplement 2D and Figure 6-figure supplement 2I). We argue that the decrease in readthrough ratio in the double knockout (DKO) lines is due to EDF1-mediated loss of translational repression in addition to ZNF598-mediated defective stall recognition in these lines – the cumulative effect of the two phenotypes is an increase in both upstream GFP and downstream RFP (or RLuc and Fluc) levels in the DKO lines. Depletion of ZNF598 increases RFP (and Fluc) levels (reflecting increased readthrough) with no effect on the upstream GFP (and RLuc) resulting in higher RFP/GFP ratios (Figure 6E and Figure 6—figure supplement 2D and Figure 6—figure supplement 2H-2I). Thus, while ZNF598 prevents readthrough on problematic mRNAs, it does not contribute towards translation repression on these mRNAs.

To address the second point, we include data in Figure 2—figure supplement 2G of the revised manuscript. There we quantify ZNF598 recruitment to polysomes as a function of EDF1 in response to emetine treatment and observed a modest (~10-20%) decrease in ZNF598 levels. These data are consistent with our observations of a moderate reduction in the ability of ZNF598 to ubiquitylate eS10 and uS10 in ∆EDF1 cells (Figure 2G). In light of these overall modest effects, we have stated our conclusions more softly in this section of the revised manuscript: “we conclude that that loss of EDF1 compromises, but is not essential for, ZNF598-mediated ubiquitylation of eS10 and uS10”.

3) EDF1 recruits GIGYF2•EIF4E2 to collided ribosomes and is required for the increase of JUN, ATF3 mRNAs by the emetine treatment (Figure 5D and Figure 6). However, it is unclear whether the increase of JUN, ATF3 mRNAs by the emetine treatment depends on the EDF1-mediated recruitment of GIGYF2•EIF4E2 to collided ribosomes. It should be examined whether the JUN, ATF3 mRNAs are increased by emetine treatment when EDF1 is not recruited to the collided ribosomes, like the RACK1-KO background.

We appreciate this idea. Our data clearly suggest that EDF1 coordinates distinct arms of the ribotoxic stress response pathway; first, we show that EDF1 recruits GIGYF2•EIF4E2 to collided ribosomes to initiate translational repression on problematic mRNAs (Figure 6); second, we show that EDF1 is critical for activating the transcriptional response to ribosomal collisions (Figure 7). The reviewers are interested here in an experiment that asks whether EDF1 functions on the colliding ribosome for both of these roles and suggests that the RACK1-KO background might be useful for this. We are concerned however that RACK1 is too centrally involved in all aspects of these pathways – from sitting at the interface of the collision and regulating ribosome-mediated QC to its known functions in recruiting JNKs and activating downstream signaling cascades. What is needed for this smart experiment is a ribosome binding defective EDF1 variant, and another large set of RNA-sequencing experiments – we would prefer to address this question in a subsequent study.

4) The authors need to show that EDF1-fusion proteins behave like the native one (i.e. associate with collided ribosomes, activates JUN, and other properties).

We are a bit confused by this concern as the only experiment in the original manuscript that depended on an EDF1-fusion protein was the proximity-dependent ligation assay (Figure 5B of the revised manuscript), the results of which strongly correlated with affinity purification/mass spectrometry of endogenous EDF1 reported in Figure 5A.

Nevertheless, we have since performed new experiments in cell lines stably over-expressing C-terminally Strep-II-tagged EDF1 (EDF1-Strep II) and are able to show that EDF1-Strep II is (1) recruited to polysomes following low-dose emetine treatment (Figure 6—figure supplement 2C), and (2) rescues the loss of translational repression phenotype observed in ∆EDF1 cells with the poly A-mediated stall reporter (Figure 6—figure supplement 2A).

For cryo-EM studies reported in the revised manuscript (Figure 3), we overexpressed N-terminal 3X-FLAG tagged EDF1 (FLAG-EDF1) in HEK293 cells. In this case too, FLAG-EDF1 is recruited to polysomes of cells treated with low-dose emetine.

They also should either remove the data for the frameshifting panel or use another reporter.

We agree with the reviewer’s concern regarding the frameshifting experiment. We pursued this experiment since Grayhack and colleagues (Wang et al., 2018) had clearly shown an increase in frameshifting in yeast lacking the EDF1 homolog, Mbf1. Our struggle with this experiment was in finding a reporter to use for a frameshifting experiment in mammalian cells. In our hands, poly(A) reporters are not suitable as they undergo so much slipping that there is no frame to report on. Iterated CGA codons are not problematic in mammalian cells, so these are not suitable either. We initially hoped that our loss in output (FLuc/RLuc) on the Xbp1u reporter reflected movement into a different frame and thus reflected a result consistent with Grayhack’s earlier work. However, we have now constructed Xbp1 reporters with FLuc in the +1 frame and the results are not as anticipated (that is, we do not observe +1 frameshifting of ribosomes in an ∆EDF1 background). We suspect the Xbp1u peptide stall is either too strong or the sequence itself is not amenable to frameshifting. We are left with no reporter to currently evaluate frameshifting in mammalian cells but assume, given the strong similarities between these proteins, that they function equivalently in these two systems. We have chosen to not include these biochemical data in the new version of the manuscript.

In our revised manuscript, we solved cryo-EM structures of Mbf1 bound to a collided ribosome (Figure 4), which enabled us to rationalize how Mbf1/EDF1 may preventing frameshifting on stalling sequences. Mbf1/EDF1 binds a conserved 40S interface at the mRNA entry channel. In this position, EDF1/Mbf1 uses a conserved KKW motif (KKY in Mbf1) with the aromatic residue positioned to interact with the mRNA in the entry channel. In collaboration with an α-helical segment that immediately follows this KKW motif, Mbf1/EDF1 clamps the mRNA in a headlock-like arrangement (Figure 4C) that is likely to prevent frameshifting.

5) Some of the claims for the role of EDF1 in recruiting ZNF598 need to be tuned down.

First, there are clearly strong previous studies about ZNF598 and its role in recruiting GIGYF2 to initiate translational repression of specific sub-classes of mRNAs (Morita et al., 2012; Garzia et al., 2019, Tollenaere., et al., 2019). Importantly, we do not rule out the possibility that semi-redundant mechanisms for the recruitment of the translational repression machinery exist, and that ZNF598 recruits GIGYF2 and EIF4E2 in other situations, or under different environmental perturbations.

Additionally, there is a recent preprint from the Weissman group (Hickey et al., 2019) – in this manuscript, the authors argue that ZNF598 is potentially important for recruiting GIGYF2 in the context of ribosomes collisions, though they acknowledge that they cannot exclude that other factors are also critical. In our manuscript, we believe that our data support a model wherein recruitment of GIGYF2•EIF4E2 to collided ribosomes to initiate translational repression is driven by EDF1, with no measurable contribution from ZNF598. We provide a more detailed explanation in our comments to

reviewer 1.

6) The authors need to explain why they used different approaches to inhibit the EDF1 function, as they went back and forth using siRNA and CRISPR knockout, as well as the drugs used to induce ribosome collisions.

We acknowledge that this was a bit confusing but summarize here. Our initial observation of the decrease in eS10 ubiquitylation (Figure 2G) and p38 phosphorylation (Figure 2—figure supplement 2H) was made upon partial depletion of EDF1 using siRNAs. All subsequent analyses of EDF1, including the validation of the eS10-ubiquitylation depletion phenotype (Figure 2H), polysome proteomics (Figure 6B-6C, Figure 6—figure supplement 1), stalling reporter assays (Figure 6E and Figure 6—figure supplement 2) and transcriptomics (Figure 7) were performed in the ∆EDF1 knockout background.

We also acknowledge that the use of different drugs was confusing and in this case underscore that we were limited by COVID-forced termination of lab work. In our initial manuscript, the activation of c-JUN in response to ribosome collisions (Figure 7A) was performed using a different translational elongation inhibitor, deoxynivalenol, as opposed to emetine. We have now replaced the deoxynivalenol titration with emetine titration – our results are consistent with those noted for deoxynivalenol – intermediate but not high doses of both deoxynivalenol and emetine induce ribosomal collisions and activate c-JUN. We have substituted this new Figure 7A in the manuscript.

We include the unabridged comments of the 3 referees for better explanation of the revisions required, and for those expected in follow up work.Reviewer #1:Ribosome quality control is an important mechanism responsible for resolving stalled ribosomes in eukaryotic cells. Important questions, particularly on the mechanisms of detection of the target mRNAs and the fate of the mRNA are unanswered. Particularly, if and how the cell blocks further rounds of translation of the mRNA which hosts the stalled ribosome remains to be revealed. In this manuscript Sinha, et al. report the discovery of Endothelial differentiation-related factor 1 (EDF1) as a key factor in regulation of RQC in response to ribosome stalling. The report that EDF1 is recruited to the stalled ribosome and this event is required for the ZNF598 mediated ubiquitination of the small ribosomal subunit proteins eS10 and uS10. Notably, they also show that EDF1 interacts with and recruits GIGYF2 and EIF4E2 proteins, which have previously been linked to miRNA-induced translational repression of target mRNAs, to the stalled ribosome.This is an interesting report which reveals new aspects of the RQC mechanism and is particularly important for its implication in a better understanding of the translational repression induced by RQC. An aspect that has been largely neglected in favor of mRNA decay induced by RQC. However, some of the findings are counterintuitive and need to be further explored/explained.- For instance, the majority of prior studies showed that 4EHP is restricted to very light fractions (e.g. Timpano and Uniacke, 2016 and Garzia et al., 2017). But in this study, 4EHP and GIGYF2 are heavy polysome fractions. Overall, considering the translational repressor function of these two proteins, it would make more sense to observe them in light fractions. Could the authors explain the possible reasons for this inconsistency?

Agreed. Our polysome proteomics and immunoblotting analyses (Figure 6) indicate that the recruitment of GIGYF2•EIF4E2 to sites of transcriptome-wide collisions is dependent on EDF1 and that these factors are together found preferentially in deep rather than light fractions. Our experiments, however, are distinct from earlier ones in using emetine to promote transcriptome-wide collisions rather than more subtle perturbations such as those that may occur during physioxia (Timpano and Uniacke, 2016) or basal growth conditions (Garzia et al., 2019).

We suggest that there could be several reasons to explain why GIGYF2 and EIF4E2 are not found in lighter fractions. First, while we do think EDF1 could be among the early responders of ribosomal collisions due to its high cellular abundance (Wisniewski et al., 2014), if collisions are transient, perhaps EDF1 is unable to engage the collided ribosomes efficiently at first, and therefore does not effectively recruit GIGYF2•EIF4E2. As collisions lead to terminal “dead-end” stalling of ribosomes, longer ribosomal queues are expected to accumulate in heavier fractions of polysomes, thus potentially allowing more efficient recruitment of EDF1 and subsequent recruitment of GIGYF2•EIF4E2. Such a situation is more likely to occur when the clearance machinery is overwhelmed as anticipated with cells experiencing abundant collisions.

We now discuss these possibilities in the Discussion section of our revised manuscript.

It is also interesting to know why GIGYF2/4EHP stays associated with a stalled ribosome in WT cells, long enough to be detected by polysome profiling. Isn't RQC-dependent mRNA decay supposed to be so fast that it degrades the target mRNA almost instantly? In that case, would not it be expected to see these proteins only in very heavy fractions?

While ribosome-mediated mRNA decay by Cue2 and Xrn1 has been shown to effectively decay problematic mRNAs with ribosome-pausing sequences in yeast (D’Orazio et al., 2019), there is currently little evidence for robust targeted degradation of NGD substrates in mammalian cells (Juskiewicz et al., 2017; Goldman et al., 2020). It is also possible that such mRNA decay pathways are overwhelmed under conditions that induce transcriptome-wide collisions. These possibilities may have contributed to a window that enabled us to enrich for EDF1, GIGYF2•EIF4E2 and other QC factors on heavier polysomal fractions of collided ribosomes.

- GIGYF2 and 4EHP recruitment to the stalled ribosome are claimed to be dependent on EDF1 but independent of ZNF598. ZNF598 has been demonstrated to interact with GIGYF2 via its PPGL motifs (Morita et al., 2012). Therefore, ZNF598 would have been the most obvious candidate for direct binding and recruitment of GIGYF2 and via that 4EHP to the stalled ribosome. In fact, the recent findings of Weissman's group (GIGYF2 and 4EHP Inhibit Translation Initiation of Defective Messenger RNAs to Assist Ribosome-Associated Quality Control) support that hypothesis. The findings in this manuscript are at variance with the latter findings, as they argue for a ZNF598-independent mechanism for the recruitment of GIGYF2 and 4EHP. The authors need to address this discrepancy. How does EDF1 do that? Does it also have a PPGL motif, which directly binds to the GYF motif on GIGYF2?

The reviewer is correct. Our finding that the recruitment of GIGYF2•EIF4E2 to sites of stalled ribosomes depends on EDF1, instead of ZNF598, was initially surprising. We are, however, confident in our data and in rationalizing the apparent discrepancies.

First: Our polysome proteomics experiment in ∆ZNF598 cells provided initial evidence that the recruitment of several quality control factors, including EDF1 and GIGYF2, to collided ribosomes was ZNF598-independent (Figure 2). Moreover, similar polysome proteomics experiments in ∆EDF1 cells provided strong evidence that the recruitment of GIGYF2•EIF4E2 to sites of stalled ribosomes is EDF1-dependent (Figure 6A-6D, Figures 6—figure supplement 1A-1D). These results were consistent with affinity purification experiments revealing GIGYF2•EIF4E2 as strong interacting partners of EDF1 (Figure 5A-5B).

Second: In experiments reporting on bulk translational output from polyA- (Figure 6E) and Xbp1u (Figure 6—figure supplement 2H) stalling reporters, we show that loss of EDF1 and GIGYF2•EIF4E2, but not ZNF598, lead to loss of translational repression. We further provide new evidence showing that re-expression of EDF1 rescues the loss of translational repression observed in ∆EDF1 cells (Figure 6—figure supplement 2A). Importantly, loss of ZNF598 had no effect on translational repression on either the polyA- or the Xbp1u stalling reporter (Figures 6E and Figure 6—figure supplement 1H). Together, our results are consistent only with a role for EDF1 in mediating GIGYF2•EIF4E2 repression in our system.

Third: Yes – it will be important to figure out the molecular interaction between EDF1 and GIGYF2. EDF1 does not have a PPPPGL motif like ZNF598, which could directly bind to the GYF motif on GIGYF2. We now have a structure of Mbf1 bound to the colliding ribosome (Figure 3 and Figure 4 of the revised manuscript) and we anticipate that these structures will help us identify mutations in EDF1 that block ribosome binding and GIGYF2 interaction.

The next issue is how to address the paper by Hickey et al., 2019.

Experiments reported in Figure 4 of Hickey et al., report on the effect of ZNF598, GIGYF2 and EIF4E2 on translational repression on non-stop and no-go reporters.

First, consistent with results reported in our manuscript, the authors report that CRISPRi-mediated depletion of GIGYF2 and EIF4E2 resulted in loss of translational repression on a polyA-mediated stalling reporter (as observed by an increase in BFP levels). Depletion of ZNF598 in Hickey et al., however, has no effect on translational repression of the polyA-reporter, consistent with our data showing that ZNF598 is not involved in recruiting the translational repression machinery to poly-A stalling reporters.

Second, the authors do see effects of ZNF598 depletion on a non-stop reporter (Figure 4A and 4C): however, these effects are partial and do not phenocopy the loss of translational repression observed with GIGYF2 and EIF4E2 depletion. This partial phenotype may either be due to incomplete depletion of ZNF598 by CRISPRi, or due to redundancy in factors that are capable of recruiting GIGYF2•EIF4E2 to collided ribosomes.

Third, the interaction studies reported by Hickey et al. are different from ours as they do not report on the ZNF598 interactome in the context of ribosomal collisions. The ZNF598-3XFLAG affinity purification reported in Hickey et al., 2019 (Figure S2) and in Garzia et al., (2019) were both performed under overexpression conditions and simply report on co-interactors of ZNF598 under basal growth conditions.

We now acknowledge in our revised manuscript that semi-redundant mechanisms for the recruitment of the translational repression machinery may exist, and that ZNF598 may recruit GIGYF2 and EIF4E2 in other situations, or under different environmental perturbations.

Reviewer #2:The manuscript by Sinha and colleagues used quantitative mass-spec approaches to identify EDF1 as a factor that is recruited to collided ribosomes. The authors then went on to explore the role of its recruitment in quality control and transcriptional response to ribotoxic stress. Human EDF1 and its homologues (called MBF1 in most organisms) have been extensively studied for their role in transcription in stress responses; they bind TATA-binding protein and bridge an interaction with transcription factors. Recently, however, yeast Mbf1 has been shown to likely associate with ribosomes through interactions with Asc1. The factor is also required to maintain reading frame of the ribosome during stalling. Furthermore, the archaeal homologue interacts with the 30S and 70S ribosome subunits.In this study, the authors expand on this ribosome-centric role of the human factor, and provide some insights into how these two seemingly unrelated functions of the factor might be coordinated by collided ribosomes. They found the factor to be required for recruiting the translation-initiation inhibitors GIGYF2/EIF4E2 to problematic mRNAs that stall the ribosome to inhibit initiation on them. Interestingly, the factor has been implicated in transcription coactivation of JUN-mediated response, and the authors in this paper show that this coactivation is likely to be mediated through ribosome collisions.Overall, I thought the paper provided some important details about the role of EDF1 in quality control and transcriptional regulation. I found the data on its recruitment to collided ribosomes and its requirement for RACK1 to be especially convincing. Having said that, for certain parts I felt the authors overinterpreted the data. This is especially true for the effect of the factor on Znf598 and frameshifting. Below I summarize my concerns that I feel need to be addressed before the paper can be published:1) I was curious as to why the authors did not carry out MS or western-blotting analyses on sucrose-gradient fractions of ribosomes treated with high concentration of emetine. Did I miss that data? Also does the factor remain on polysomes following RNAse treatment.

These are important points. The high concentration emetine is an important control that was lacking in our initial draft. We have now included the high-emetine treatment to monitor the distribution of EDF1 along the sucrose gradient under different concentration regimes of emetine (Figure 1I). We report that while little to no EDF1 was detected in polysomal fractions of untreated and high-dose emetine treated cells, EDF1 was strongly recruited to polysomes of cells treated with low-dose emetine (Figure 1I). For cost and instrumentation time reasons, we opted not to include the high-emetine samples in our polysome proteomics experiment.

As for the potential association with colliding ribosomes per se, we are unable to detect interaction between EDF1 and colliding ribosomes following digestion of lysates with RNaseA. While a minor fraction of EDF1 remains associated with nuclease-resistant disomes and trisomes following RNase A digestion, the majority of EDF1 shifts to monosomal and RNP fractions. In our revised manuscript, we include cryo-EM structures of Mbf1 binding to colliding yeast ribosomes. This structure reveals Mbf1 making extensive contacts with helices 16, 18, and 33 (Figure 3) of the 18S rRNA (Figure 3D-3E of revised manuscript). Details about these interactions are discussed in the revised manuscript – we suspect that some of these RNA binding interfaces between EDF1 and the colliding ribosome are sensitive to RNaseA digestion.

2) It is not clear to me how the data in Figure 3A report on frameshifting. Is Fluc out of frame with Rluc? Shouldn't EDF1 deletion promote more readthrough, unless fluc is in frame. In that case, a different reporter is needed. In yeast deletion of Mbf1 on its own is sufficient to promote frameshifting.

We acknowledge that the experiment in Figure 3A was unclear and difficult to interpret. We performed additional experiments which are inconsistent with our previous interpretation, and have now removed this figure (and section) as suggested by the reviewer. We have provided a more detailed discussion above (Essential revisions, point 4, second section).

3) I found the data on ribosomal protein ubiquitination not convincing. The changes in the levels of ubiquitinated ribosomal proteins seem small. If EDF1 indeed affects ZNF598 function, one would expect its deletion to have a similar effect on the stalling reporter.

We agree with this comment. The decrease in collision-dependent eS10 and uS10 ubiquitylation that we observe in the absence of EDF1 is modest (but reproducible).

Our initial observation of the reduced ubiquitylation phenotype (Figure 2G) with partial depletion of EDF1 (using siRNAs) prompted us to probe this effect further in ∆EDF1 cell lines using a targeted mass-spectrometry approach (Figure 2H) – these data confirmed that eS10 ubiquitylation at K138/139 was indeed affected in ∆EDF1 lines. Moreover, this phenotype was recapitulated with different clonal populations of ∆EDF1 under physiological conditions (UV-treatment) that induce ribosome collisions (Figure 2—figure supplement 2E). In each case, however, the reduction in ubiquitination is partial, and thus EDF1 is not essential for the recruitment of ZNF598.

We suspected that the decrease in eS10 ubiquitylation is due to reduced efficiency of ZNF598 recruitment or its stability on collided ribosomes in ∆EDF1 cells. Indeed, when we quantitated ZNF598 abundance on polysomes of ∆EDF1 cells in response to emetine treatment, we observed a modest (10-20%) decrease in ZNF598 protein levels (included now as Figure 2—figure supplement 2G), which is consistent with the modest decrease in eS10 ubiquitylation at K138/139 (Figure 2I).

In light of the reviewer’s comment and our new quantitation of the ZNF598 interaction, we have softened our conclusions based on this new perspective.

4) Does the immunoprecipitation shown in Figure 4 work for ribosome-bound factor; is the antibody capable of recognizing the factor when it associates with the ribosome? More important, does the BirA-fused protein associate with the collided ribosomes? Does it rescue the ∆EDF1 cells?

This is an important point. The polyclonal antibody against EDF1 was raised against an antigen comprising amino acids 98-148, which includes the C-terminal HTH domain. Since the HTH domain is required for ribosome association (Figure 3), we were concerned that the antibody would only recognize EDF1 and its associated binding partners off the ribosome; however, r-proteins and translational initiation/elongation factors accounted for a significant proportion of proteins robustly interacting with EDF1– these are indicated in Figure 5A (60S r-proteins indicated in blue, 40S r-protein in orange) and in Figure 5—source data 1. Based on these data, we conclude that ribosome-bound EDF1 is recognized by the antibody.

The complementary proximity ligation experiment with BirA*-tagged EDF1 was performed in the absence of emetine (for technical reasons). Since EDF1 does not associate with ribosomes in the absence of collisions, as anticipated, we saw no enrichment of r-proteins in our Bio-ID experiment (Figure 5B). We have not tested whether BirA*-tagged EDF1 sediments with colliding ribosomes or can rescue the translational repression phenotype in ∆EDF1 cells using our poly-A stalling reporter. However, we have performed additional experiments with EDF1-Strep II fusion variants which associate with collided ribosomes and rescue the defect in translational repression. We have provided a detailed discussion above (Essential revisions, point 4, first section).

5) Why did the authors keep changing between using siRNAs and ∆EDF1?

We addressed this point above and apologize for the confusion (Essential revisions, point 6).

6) Could the authors speculate on why the deletion of ZNF598 results in less translation of the reporter (Figure 2G)? What were the GFP fluorescence values normalized to? The data in Figure 5E shows opposite results. Deletion of the factor results in slight increase of GFP in this case.

While there are several possibilities, we speculate that the differences in relative GFP intensities in the ∆ZNF598 backgrounds between Figure 2G (in HCT116 cells) and Figure 6E (in HEK293-FlpIn-TRex cells) arose from cell-specific backgrounds with differences resulting from transient overexpression of the reporter and in overall differences in translational capacity and quality control processes. We prefer not to overinterpret these small changes, as this point is not critical for any of the main conclusions in this study.

Because of the semi-redundant nature of these experiments and the space constraints arising from new cryo-EM data, we removed Figure 2G from the original draft in our revised manuscript – we consider this change to be minor and to have no impact on any main conclusions in this study.

For all conditions, the mean GFP fluorescence intensities (obtained from 10000 individual events, 48 hours following reporter transfection) were normalized to the mean fluorescence intensity of the parental (WT) cell line.

Reviewer #3:The authors identified EDF1 as a novel protein recruited to collided ribosomes during translational distress with proteomics combined with sucrose gradient fractionation. They demonstrated that (1) RACK1, but ZNF598 is required for the recruitment of EDF1 to collided ribosomes. (2) ZNF598-mediated ubiquitylation of eS10 and uS10 and ZAKa-mediated phosphorylation of p38 by the emetine treatment is moderately reduced in the EDF1-KO cells. (3) EDF1 recruits GIGYF2•EIF4E2 to collided ribosomes and is required for the increase of JUN, ATF3 mRNAs by the emetine treatment. Based on these results, the authors propose that EDF1 binds collided ribosomes, recruits translational repressors GIGYF2/EIF4E2, and initiates a transcriptional response.Major comments:1) The reduction of ribosome stalling by K(AAA)20 and XBP1u in ZNF598 KO is dependent on EDF1 (Figure 3 and Figure 5E). In ribosome stalling by K(AAA)20, the ZNF598-independent function of EDF1 is dependent on GIGYF2•EIF4E2 (Figure 5E). It is necessary to confirm that EDF also functions in ribosome stalling on XBP1u mRNA in ZNF598-independent but GIGYF2•EIF4E2-independent manner.

We thank the reviewer for this suggestion. As discussed above (Essential revisions, point 1), we have now tested the role of EDF1 (and ZNF598) in translation repression using a *Renilla*-2A-3xFLAG-Xbp1u-2A-Firefly reporter and a corresponding version without the intervening stalling sequence. These data have now been included in Figure 6—figure supplement 2H-2I of the revised manuscript and nicely provide an additional handle on the role of EDF1 in promoting translational repression on the multiple ribosome-stalling sequences that we tested.

2) ZNF598-mediated ubiquitylation of eS10 and uS10 and ZAKa-mediated phosphorylation of p38 by the emetine treatment is moderately reduced in the EDF1-KO cells. However, EDF1 slightly represses ribosome stalling by K(AAA)20 and XBP1u in the presence of ZNF598 (Figure 3 and Figure 5E). This may be due to the reduced association of ZNF598 in the EDF1-KO cells. This possibility could be examined by the determination of the distribution of ZNF598 in the EDF1-KO cells.

We thank the reviewer for these suggestions. We have provided a detailed discussion above (Essential revisions, point 2).

3) EDF1 recruits GIGYF2•EIF4E2 to collided ribosomes and is required for the increase of JUN, ATF3 mRNAs by the emetine treatment (Figure 5D and Figure 6). However, it is unclear whether the increase of JUN, ATF3 mRNAs by the emetine treatment depends on the EDF1-mediated recruitment of GIGYF2•EIF4E2 to collided ribosomes.

This is an interesting point. As discussed above (Essential revisions, point 3), we find that EDF1 contributes to at least two arms of the ribotoxic stress response – first, to recruit GIGYF2•EIF4E2 to collided ribosomes and consequent translational repression on problematic mRNAs (Figure 6), and second to coordinate a transcriptional response to ribosomal collisions (Figure 7). Our studies do not establish whether these responses are directly coordinated, for example through GIGYF2•EIF4E2. Those will be interesting questions to pursue in subsequent studies, but we feel they are beyond the scope of this initial publication.

It should be examined whether the JUN, ATF3 mRNAs are increased by the emetine treatment in the condition that EDF1 is not recruited to the collided ribosomes, like the RACK1-KO background.

This point was addressed above (Essential revisions, point 3). While we think this is a very interesting question, we feel that the proposed experiments are beyond the scope of the current manuscript. Importantly, the new structural information that we now provide (Figure 3 and Figure 4) will guide the construction of ribosome-binding defective EDF1 variants to ask this question in a direct manner.